# Multiple myeloma associated long non-coding RNA *PLUM* confers chemoresistance by enhancing PRC2 mediated UPR pathway activation

Kamalakshi Deka[1], Jean-Michel Carter [1], Akash Bahai[1], Daniel Aron Ang [1], Nicholas Sim [1], Hooi Yan Chong[2], Guan Hwee Bernard Lee[1], Suet Mien Tan [1], Wee Joo Chng [2,3,4,5], Dennis Kappei [2,4,6] & Yinghui Li [1,7] ✉

Multiple myeloma (MM) is the second most common hematological malignancy that displays diverse genetic heterogeneity leading to treatment resistance. Recurrent mutations causing hyperactivation of the non-canonical NF-κB pathway are highly prevalent in relapsed, refractory MM patients, but the precise mechanisms driving chemoresistance are poorly understood. Here, we identify a long non-coding RNA termed *PLUM*, that is overexpressed in NF-κB mutant high-risk MM subtypes and patients who are refractory to VRd treatment regimen. Mechanistically, *PLUM* interacts with Polycomb Repressive Complex 2 to regulate its stability and histone methyltransferase activity, modulating the expression of tumor suppressor genes, *FOXO3* and *ZFP36*, to activate the unfolded protein response (UPR). Importantly, disruption of *PLUM*-EZH2 interaction using steric antisense oligonucleotides re-sensitizes myeloma cells to drug treatment in vivo, correlating with the loss of PRC2 stability and H3K27 trimethylation activity. These findings indicate that *PLUM* facilitates formation of PRC2 complex and enhances EZH2 activity, modulating the myeloma epigenome to mediate chemoresistance. Hence, targeting *PLUM*-EZH2 interactions may represent a clinically potent strategy for the treatment of relapsed, refractory MM.

Multiple myeloma (MM) is an aggressive B-cell malignancy that displays poor prognosis despite significant advances in novel treatment strategies[1–3]. The current treatment regimen comprises of Velcade® (bortezomib) with Revlimid® (lenalidomide) and dexamethasone [VRd], in combination with antibody treatment[4–8]. However, it remains incurable due to high genetic heterogeneity driving treatment resistance and relapse[9–12]. Genetic mutations leading to the dysregulation of oncogenic pathways have been implicated to contribute to the chemoresistance of MM patients[13]. One such aberrantly activated pathway is the non-canonical NF-κB (ncNF-κB) pathway, which we recently found to

[1]School of Biological Sciences (SBS), Nanyang Technological University (NTU), Singapore, Singapore. [2]Cancer Science Institute of Singapore, National University of Singapore (NUS), Singapore, Singapore. [3]Department of Medicine, Yong Loo Lin School of Medicine, NUS, Singapore, Singapore. [4]NUS Centre for Cancer Research, Centre for Translational Medicine, Singapore, Singapore. [5]Department of Hematology-Oncology, National University Cancer Institute of Singapore (NCIS), The National University Health System (NUHS), Singapore, Singapore. [6]Department of Biochemistry, Yong Loo Lin School of Medicine, NUS, Singapore, Singapore. [7]Institute of Molecular and Cell Biology (IMCB), Agency for Science, Technology and Research (A*STAR), Proteos, Singapore, Singapore. ✉e-mail: liyh@ntu.edu.sg

potentiate oncogenic transcription through reprogramming of the myeloma epigenome[14,15]. However, the mechanistic role of ncNF-κB signalling and its downstream targets contributing to the chemoresistance of MM remains unexplored.

Long non-coding RNAs (lncRNAs) are a part of the non-coding transcriptome that are longer than 200 nucleotides (nt), with weak or no-coding capacity[16,17]. Due to their specificity towards disease types and ability to control gene expression, lncRNAs are emerging as potential candidates for therapeutic targeting as well as biomarker discovery. Several lncRNAs have been reported to promote chemoresistance in MM by regulating various factors, including miRNAs and oncogenes, implicating their potential applications as biomarkers or drug targets for diagnosis, prognosis and therapy[18–20]. Although majority of these studies are still under pre-clinical investigation, the advancement in FDA approved RNA-based therapies such as antisense oligonucleotides (ASOs) for various diseases suggest the promising prospect of developing lncRNA based therapeutics to improve cancer treatment[21]. While there are currently no FDA approved lncRNA targeting drugs for cancer therapy, a number of clinical trials are ongoing to evaluate the use of lncRNAs as biomarkers for cancer detection and therapy, as well as oligonucleotide based therapeutics targeting oncogenic lncRNAs. Some reported lncRNAs which are in clinical trials showing successful targeting using oligonucleotide based therapeutics include lncRNA *TUG1* in glioblastoma, *H19* in pancreatic cancer, and non-coding mitochondrial RNA in several solid cancers[22–25] (database: http://clinicaltrials.gov). Other trials also include detecting the expression of lncRNAs like *CCAT1* and *HOTAIR* in CRC patients[25], *HOTAIR* in thyroid cancer[26] and in general using lncRNAs as biomarkers for the detection/prognosis of several other cancer types including lung cancer (NCT03830619), ovarian cancer (NCT03738319) and triple negative breast cancer (NCT02641847). Such developments pave the way towards the discovery and characterization of mechanisms driving the activation and cellular functions of cancer associated lncRNAs. Our previous study showed that dysregulated ncNF-κB signalling can alter the myeloma epigenome, in turn regulating gene expression critical for MM progression[14]. Hence, investigating whether this pathway can drive the expression of lncRNAs associated with myeloma progression and resistance to current chemotherapeutic drugs may provide avenues for the targeted detection and therapy of MM patients.

Emerging studies have revealed the diverse functions of lncRNAs as recruiters, guides and scaffolds for protein complexes, including chromatin modifiers to assist in epigenetic activities[27]. Among the lncRNA interacting chromatin modifiers, PRC2 is being extensively studied in relation to oncogenic lncRNAs across various cancers, including MM[28–30]. Enhancer of Zeste Homolog 2 (EZH2), the enzymatic subunit of Polycomb repressive complex 2 (PRC2), is one of the major chromatin regulator in MM that mediates histone H3 lysine 27 trimethylation (H3K27me3) to repress gene expression[31,32]. Its aberrant activity has been implicated in the pathogenesis and chemoresistance of several malignancies, including MM[33–36]. Additionally, the non-canonical functions of EZH2 via lncRNA interactions have been reported to mediate chemoresistance in MM[29,37]. Recently, PRC2 complex interaction with lncRNA *PVT1* have also been shown to repress genomic regions associated with apoptosis and tumour suppressor function in MM[38]. Besides EZH2, deregulation of the unfolded protein response (UPR) pathway is causally linked to the poor clinical outcome of MM. MM cells continuously secrete M-proteins, leading to persistent endoplasmic reticulum (ER) stress that makes them highly reliant on UPR for survival[39]. Thus, the extent of ER stress driven protein aggregate accumulation and UPR pathway activation determines the sensitivity of myeloma plasma cells to therapeutic regimens and their fate towards apoptosis or recovery. However, the mechanisms by which non-coding RNAs regulate UPR mediated chemoresistance in MM are not well understood.

In this study, we examine the role of lncRNAs regulated by ncNF-κB pathway in the chemoresistance of MM. Through integrated analysis of RNA-seq data from NF-κB/p52 knockdown (KD) MM cells[14] with the transcriptomic profiles of MM patients, we identify a NF-κB/p52-regulated lncRNA, *LINCO2362*, which is upregulated in NF-κB mutant, high-risk MM. This lncRNA, termed as *PLUM* (*PRC2 associated LncRNA regulating UPR in MM*), interacts with EZH2 to mediate PRC2 complex formation and its activity, promoting chemoresistance via activation of UPR pathway. Importantly, targeting the *PLUM*-EZH2 interaction using steric ASOs effectively disrupts *PLUM*-EZH2/PRC2 complex stability and enzymatic function, re-sensitizing myeloma cells to drug treatment. Our results demonstrate that *PLUM* alters the epigenetic functions of PRC2 by interacting with EZH2 to mediate chemoresistance in MM. Our data further suggest that *PLUM* may be a potential therapeutic target for high-risk MM patients displaying enhanced EZH2 levels and resistance to current treatments.

## Results

### *PLUM* is p52-regulated and upregulated in NF-κB+ high-risk myeloma subtypes

To understand the role of ncNF-κB regulated lncRNAs in MM progression, we performed RNA seq analysis of control versus NF-κB/p52 KD MM cells, which identified several p52-regulated non-coding transcripts showing negative correlation with the expression of p52 (blue circles in Fig. 1a). We next integrated differentially regulated transcripts with the transcriptomic profiles of patient samples from MMRF CoMMpass study (NCT01454297) and identified an uncharacterized lncRNA—*LINCO2362* (termed as *PLUM*) associated with NF-κB mutant, high-risk MM subtypes (Fig. 1a). The direct regulation of *PLUM* expression by NF-κB/p52 was further confirmed by CRISPR-mediated KD of NF-κB/p52 in two NF-κB mutant multiple myeloma cell lines (MMCLs), KMS11 and LP1, which resulted in the downregulation of *PLUM* expression (Supplementary Fig. 1a, b). Additionally, MMCLs carrying activating mutations in NF-κB pathway (NF-κB+) display elevated *PLUM* expression compared to non-mutant (NF-κB−) MMCLs (Supplementary Fig. 1c), correlating with NF-κB activity.

To determine the clinical relevance of *PLUM* in MM, we examined its expression levels in various MM patient subtypes and cancer cell lines (from Cancer Cell Line Encyclopedia—CCLE). Using MMRF CoMMpass dataset, *PLUM* was found to be overexpressed in patient samples harbouring high-risk, aggressive MM subtypes displaying hyperactivation of NF-κB pathway (subtyping data published in ref. 14) (Fig. 1b). Amongst the p52-regulated non-coding transcripts, *PLUM* is one of the top upregulated lncRNAs in the MAF *t*(14;16) and HRD low TP53 high-risk subtypes carrying NF-κB activation (NF-κB+) (Fig. 1b, c). Additionally, we analysed the transcriptomic data from MM patients undergoing VRd regimen therapy using CoMMpass metadata (Best response). Patients showing Complete Response (CR), Stringent Complete Response (SCR) or Very Good Partial Response (VGPR) were grouped as responsive and patients with Stable Disease (SD) or Progressive Disease (PD) were grouped as non-responsive. Interestingly, *PLUM* expression was found among the significantly upregulated genes in non-responsive (SD or PD) versus responsive (CR, SCR or VGPR) patients under differential testing (DESeq2, FDR < 0.1), with non-responsive patients showing ~ twofold increase in expression (Fig. 1d). *PLUM* is also selectively upregulated in MMCLs compared to other 51 cancer types and other p52-regulated lncRNAs like *NR2F2-AS1* and *MALT1-AS1* found in our earlier analysis, suggesting its aberrant expression could be linked to MM disease (Fig. 1e). These findings demonstrate the plausible association of *PLUM* overexpression with NF-κB activated, high-risk subtypes of MM and poorer treatment response.

We next characterized the expression and cellular localisation of *PLUM* in MM cells. 5′–3′RACE assays revealed 3 main isoforms of *PLUM* in NF-κB + MMCLs (KMS11 and LP1), whereby isoform 201 (Batch 3) represents approximately 40% of all isoforms (Fig. 1f and Supplementary Fig. 1d, e). Overexpression studies of *PLUM* using a HA-tagged construct verified this lncRNA does not code for any protein or small

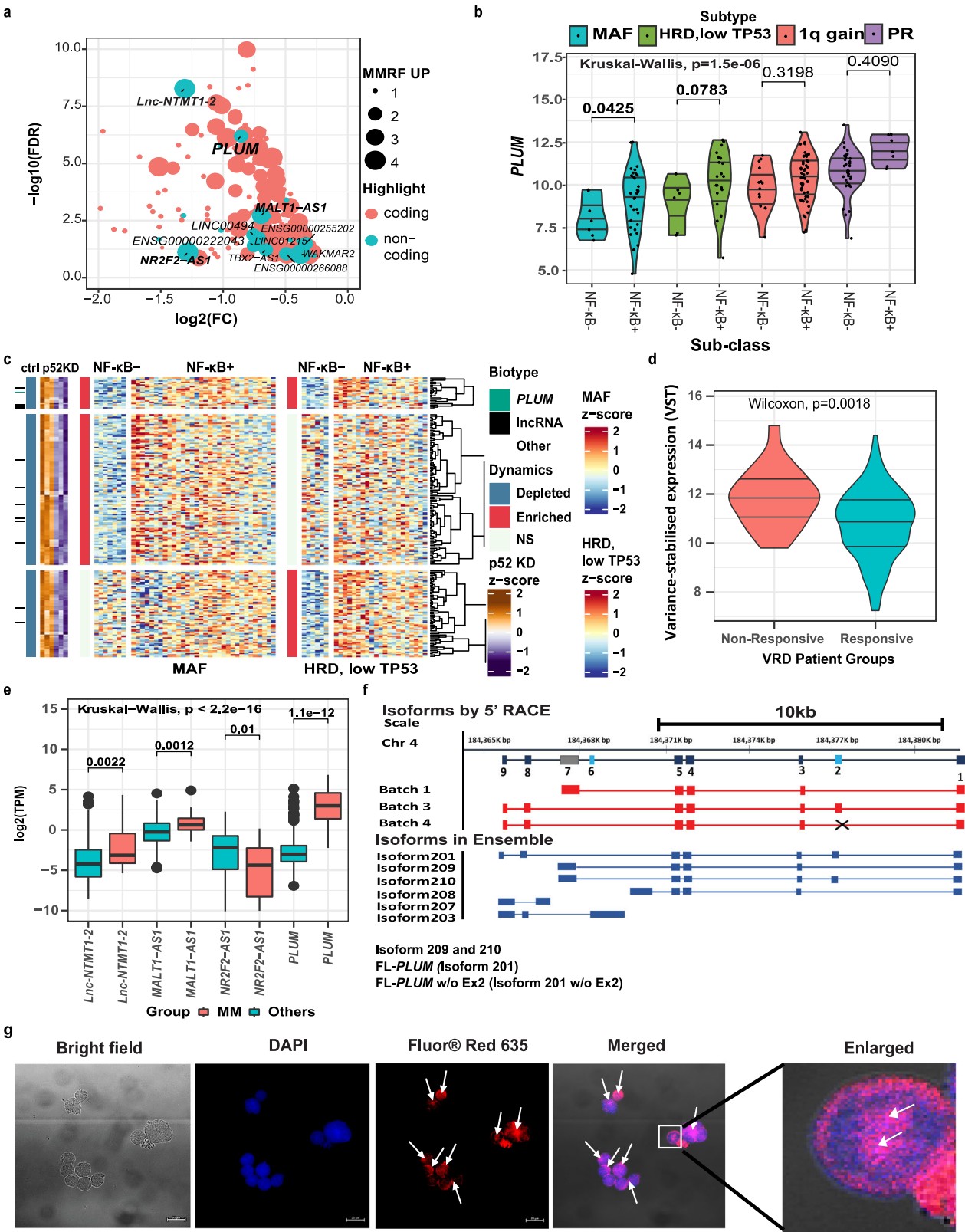

peptides (Supplementary Fig. 1f). RNA-FISH and sub-cellular fractionation studies further indicated *PLUM* is predominantly localised in the nucleus (Fig. 1g and Supplementary Fig. 1g–i).

### *PLUM* confers enhanced proliferation and VRd resistance in MM

To understand the functional relevance of *PLUM* in MM, we used both gain of function and loss of function approaches. Overexpression of

*PLUM* (FL-*PLUM* isoform 201) in both NF-κB + MMCLs (KMS11, LP1, MM.1.S) and NF-κB− MMCLs (XG7 and H929) resulted in their enhanced proliferation (Fig. 2a and Supplementary Fig. 2a–d), whereas shRNA KD of *PLUM* induced the apoptosis of NF-κB + MMCLs post 6 days lentiviral transduction (Fig. 2b and Supplementary Fig. 2e, f). Consistent with the oncogenic functions of *PLUM*, treatment with ASOs designed to degrade *PLUM* reversed the proliferative phenotype

**Fig. 1 | PLUM is a p52-regulated, oncogenic lncRNA enriched in high-risk MM subtypes carrying NF-κB activation. a** Plot showing the coding and non-coding genes downregulated upon p52-KD in KMS11 cells ($N = 3$; 2-sided Wald Test). Red dots: coding transcripts; Blue dots: non-coding transcripts. Size of the dots: Number of high-risk MM subtypes showing upregulation in NF-κB+ samples. **b** Violin plot showing the expression level of *PLUM* in patient samples divided into 4 aggressive subtypes and NF-κB activity (−/+): 1q gain ($N = 7/32$), HRD-low TP53 ($N = 6/21$), MAF ($N = 14/50$), and PR ($N = 33/7$). Pair wise and global *p*-values obtained using two-sided Wald (DESeq2) and Kruskal–Wallis (total $N = 170$), respectively. **c** Heat map representing gene expression dynamics of p52-regulated lncRNAs and other genes across MAF and HRD low TP53 patient samples split by NF-κB activity identified in DESeq2 (2-sided Wald, FDR < = 0.1) using RNA-seq data from[116]. Highlight: green-*PLUM* and black-other p52-regulated lncRNAs. Dynamics: blue−depleted, red−enriched and white−non-significant (ns) with relation to p52 KD and NFκB+/−. *PLUM* marked in green. $N = 3$. **d** Violin plot showing *PLUM* expression in VRd non-responsive ($N = 19$) and responsive patients ($N = 225$).

Pairwise-comparison for variance-stabilized transformed (VST) expression was determined by two-sided Wilcoxon rank-sum test (total $N = 244$); median and quartiles indicated with *p*-value = 0.018. **e** Expression level (log2 TPM) of top p52-regulated lncRNAs (*lnc-NTMT1, MALT1-AS1, NR2F2-AS1* and *PLUM*) in 51 pan-cancer cell types from CCLE datasets (Others, $N = 638$; MM, $N = 20$). Lower and upper hinges correspond to first and third quartiles. Central value corresponds to median. Lower and upper whiskers correspond to smallest and largest value. Outliers beyond 1.5× IQR indicated. Pair wise and global *p*-values obtained using two-sided Wilcoxon and Kruskal–Wallis, respectively. (*p*-values for *lnc-NTMT1-2*: 0.0022, *MALT1-AS1*: 0.0012, *NRF2-AS1*: 0.01, *PLUM*: 1.1e⁻12). **f** Line diagram showing isoforms of *PLUM* identified by 5′RACE (Red boxes: exons, red lines: introns) and 6 isoforms of *LINCO2362* available in Ensemble database (Blue boxes: exons; blue lines: introns). **g** Micrographs localization of *PLUM* using RNA-FISH probe (Fluor@-Red635) and nuclei stained with DAPI (Blue). Arrows: enriched spots for *PLUM*. Inset: colocalization of red color (Fluor@Red635 for *PLUM*) with blue color (DAPI for nuclei). Scale bar: 20 μm and $N = 3$ biological replicates.

of NF-κB + MMCLs (KMS11 and LP1) but not that of NF-κB− MMCLs (XG-7 and H929) (Supplementary Fig. 2g–i). There was no induction of apoptosis as well in NF-κB- MMCLs (XG-7 and H929) on treatment with *PLUM* targeting degradative ASOs (Supplementary Fig. 2j). These data confirmed the identification of a NF-κB/p52-regulated oncogenic lncRNA, *PLUM*, associated with myeloma progression.

Furthermore, consistent with the elevated expression of *PLUM* in VRd non-responsive patients, MMCLs carrying resistance to VRd components also displayed elevated *PLUM* expression compared to their parental sensitive lines (Supplementary Fig. 3a). Hence, we next validated its chemoresistance functions in both NF-κB+ and NF-κB− MMCLs against two of the main drugs, bortezomib (BTZ) and lenalidomide (Len). Overexpression of *PLUM* in NF-κB + MMCLs (KMS11, LP1 and MM.1.S) and NF-κB− MMCLs (XG7 and H9292) conferred resistance to both drugs, with an increase in IC$_{50}$ levels by ~two to threefold relative to vector control (Fig. 2c–f and Supplementary Fig. 3b). In contrast, degradative ASO-mediated *PLUM* repression enhanced the sensitivity of NF-κB + MMCLs (KMS11 and LP1) to both BTZ and Len treatment (Supplementary Fig. 3c, d). Meanwhile, NF-κB- MMCLs (XG-7 and H9292) showed no change in sensitivity to both drugs upon treatment with degradative ASOs targeting *PLUM* (Supplementary Fig. 3e, f), suggesting the dependency on *PLUM* overexpression in mediating the chemoresistance function of these MMCLs. Further in vivo validation studies revealed the enhanced tumour growth and poorer survival of *PLUM*-overexpressing (OE) subcutaneous tumour xenografts compared to their vector control (VC) counterparts. Moreover, *PLUM* OE xenografts showed increased resistance to BTZ treatment and worse survival than VC xenografts treated with BTZ (Fig. 2g–i). These observations indicate the significant role of *PLUM* in conferring oncogenicity and chemoresistance in MM.

### *PLUM* interacts with PRC2 complex protein, EZH2, to mediates its catalytic function

To understand how *PLUM* mediates tumorigenicity and chemoresistance in MM, we characterized its functional domains and interacting factor(s). Our exon deletion mutagenesis study showed mutants deficient in exon1 and/or exon7 lose their oncogenic effects in proliferation and BTZ resistance in MMCLs (Fig. 3a, b and Supplementary Fig. 4a–c). Thus, we performed RNA-protein pull down experiments using three *PLUM* templates (Full length, ΔExon1 and ΔExon7) with nuclear protein extract, followed by label-free quantitative (LFQ) mass spectrometry (MS) analysis to identify the associated RNA binding proteins (RBPs) (Fig. 3c and Supplementary Data 1). While the MS analysis did not display any differentially bound RBPs between FL and ΔExon1 *PLUM* (Fig. 3d), deletion of ΔExon7 revealed a distinct list of differentially bound RBPs. Notably, EZH2, the catalytic subunit of the PRC2 complex, was identified as a *PLUM* interactor (Fig. 3e). The interaction was further confirmed by RNA immunoprecipitation (RIP)

qPCR and microscopy-based co-localization studies using RNA FISH-IF (FISH for *PLUM* and IF for EZH2) (Fig. 3f, g and Supplementary Fig. 4d).

Next, we investigated the binding of two other stoichiometric factors forming PRC2 core complex: Zeste (SUZ) 12 and embryonic ectoderm development (EED) to *PLUM* by performing RNA-protein pull down assays on nuclear extracts from KMS11 and acquired BTZ resistant RPMI8226 (BTZ-R 8226) cell lines followed by immunoblotting. The data confirmed their binding to *PLUM* via exon7 for both cell lines (Fig. 3h and Supplementary Fig. 4e). However, we found that binding of EED to *PLUM* requires both exon1 and exon7. To understand the binding biochemistry of EED further, we performed RNA-EMSA using purified EED and biotinylated exon1/exon7 region of *PLUM*. Our data showed direct binding of EED to exon1 but not exon7, suggesting its binding to exon7 of *PLUM* could be indirectly mediated via EZH2 (Supplementary Fig. 4f).

Further, we explored how the *PLUM*-EZH2 complex is regulated. While sh-EZH2 KD did not affect *PLUM* levels (Supplementary Fig. 4g), KD of *PLUM* or overexpression of exon7 deletion mutant reduced the stability of EZH2, without affecting SUZ12 or EED levels (Fig. 3i). The impact of *PLUM* KD on EZH2 stability was further verified through cycloheximide (CHX) chase and MG132 treatment assays (Fig. 3j, k). Additionally, CHX chase assay using KMS11 cells expressing exogenous EZH2, and treated with degradation ASOs (Supplementary Fig. 4h), confirmed the regulation of EZH2 stability by *PLUM*.

Since CDK inhibitors are known targets of EZH2 that negatively regulate EZH2 activity[33,40], we examined whether there is a regulatory feedback mechanism underlying *PLUM* mediated EZH2 stability and activity. ASO-mediated *PLUM* degradation resulted in the upregulation of CDK inhibitors (p14/p15/p21), consequently leading to the inhibition of CDK1/2 mediated EZH2 phosphorylation. This resulted in the loss of the histone methyltransferase activity of EZH2, despite its exogenous expression (Fig. 3l). These observations implied *PLUM* interaction with EZH2 could be critical for its catalytic activity, which in turn represses the expression of CDK inhibitors.

### Disordered region mediated interaction of EZH2 with *PLUM* is crucial for PRC2 complex formation

Given the importance of *PLUM*-EZH2 interaction in the activity and stability of EZH2, we investigated the plausible role of this ribonucleoprotein (RNP) complex in PRC2 complex formation. Through co-immunoprecipitation (co-IP) assays using sh-scramble and sh2-*PLUM* transduced cells in the presence of MG132 (to protect EZH2 from degradation), we observed a diminished interaction of EZH2 with EED and SUZ12 following *PLUM* KD compared to sh-scramble controls in NF-κB+ MMCLs−KMS11, LP1 and BTZ-R 8226 (Fig. 4a and Supplementary Fig. 5a). Interestingly, absence of EZH2 inhibited the interaction of *PLUM* with EED and SUZ12 whilst loss of EED or SUZ12 expression had minimal impact on binding of *PLUM* to EZH2 (Fig. 4b–d and

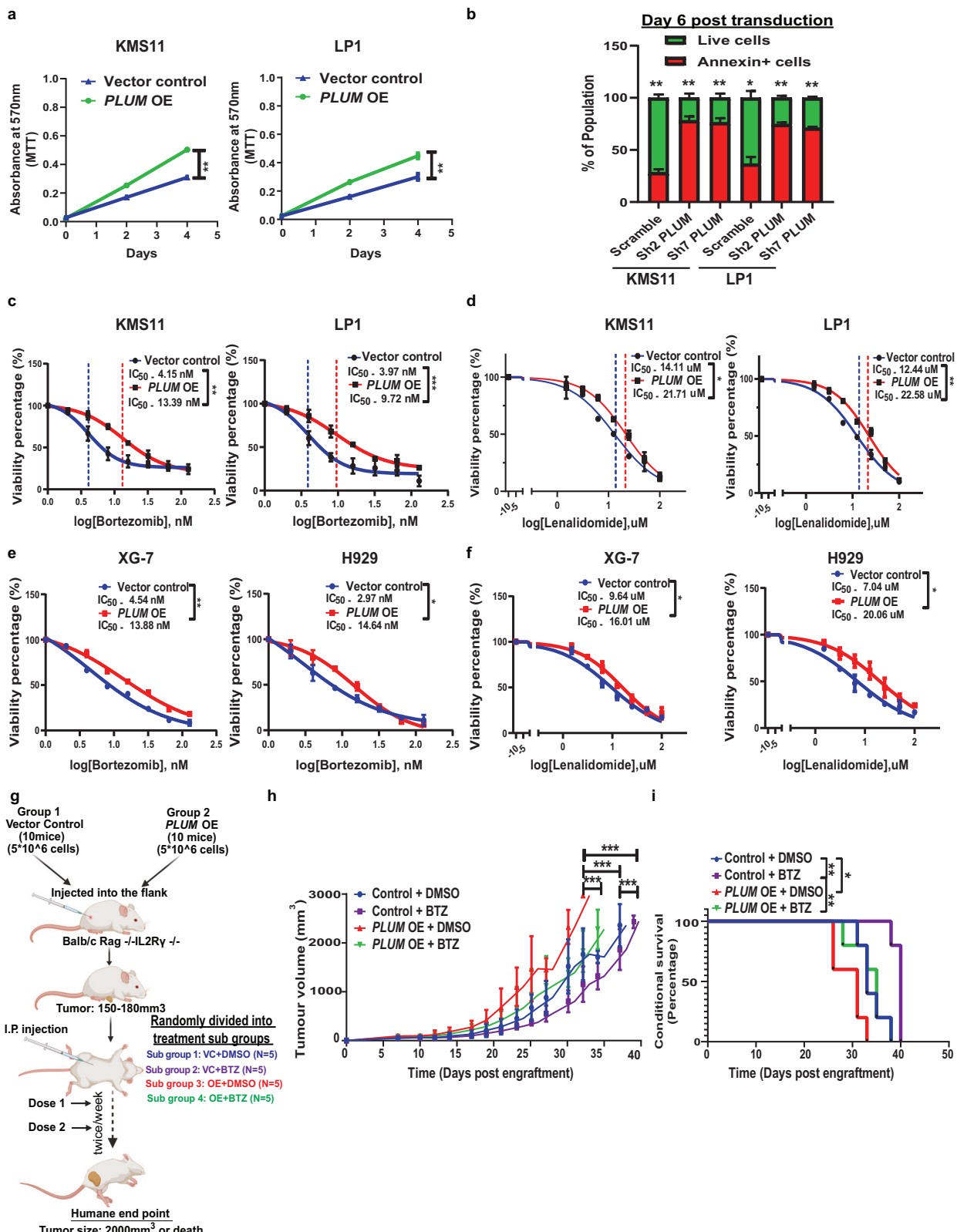

Supplementary Fig. 5b). These findings suggested the requirement for EZH2 to interact with *PLUM*, thereby facilitating its binding with EED and SUZ12 to form the PRC2 complex.

To examine the biochemistry of *PLUM*-EZH2 interaction, we performed in-silico docking using the remodelled structure of EZH2 and predicted tertiary structure of exon7 *PLUM* via HDOCK[41], thereby reconstructing the *PLUM*-EZH2 RNP complex (Fig. 4e). From the docking

data, the most significant interaction was observed with the C-terminal region of EZH2, 488–498 residues within the reported disordered region of EZH2 (479–515) and two additional structured regions (562–575 and 586–602) (Fig. 4e). Consistent with this, structural predictions of full-length *PLUM*-EZH2 complex using Alphafold3[42] similarly revealed interaction of the disordered region of EZH2 protein (491–497 residues) with atoms in the exon 7 of *PLUM* (Supplementary

**Fig. 2 | Oncogenic *PLUM* confers chemoresistance in MM. a** Graphs representing proliferation rate (mean ± SEM) of FL-*PLUM* overexpressed KMS11 and LP1 cells relative to vector control ($N = 3$, two-way ANOVA; $p$-values- KMS11: 0.0006 and LP1: 0.0097). **b** Box plot showing percentage of live and apoptotic cells (Annexin V staining) (mean ± SEM) in sh-scramble versus sh-*PLUM* MMCLs (KMS11 and LP1) at day 6 post transduction ($N = 3$, two-sided multiple *t*-test; $p$-values—KMS11-scramble: 0.001, KMS11-sh2: 0.001, KMS11-sh7: 0.001, LP1-scramble: 0.04, LP1-sh2: 0.001, LP1-sh7: 0.001). **c, d** Drug sensitivity $IC_{50}$ survival curve (mean percentage ± SEM) for Vector control (VC) and FL-*PLUM* overexpressed NF-κB+ mutant MMCLs (KMS11 and LP1) in response to BTZ treatment for 24 h and Len treatment for 4 days respectively ($N = 3$, two-way ANOVA; $p$-values for BTZ treatment—KMS11: 0.0006 and LP1: 0.0001, $p$-values for Len treatment—KMS11: 0.045 and LP1: 0.0027). **e, f** Drug sensitivity IC50 survival curve (mean percentage ± SEM) for vector control (VC) and FL-*PLUM* overexpressed NF-κB- non-mutant MMCLs (XG-7 and H929) in response to BTZ treatment for 24 h and Len treatment for 4 days, respectively

($N = 3$, two-way ANOVA; $p$-values for BTZ treatment—XG-7: 0.0013 and H929: 0.023), $p$-values for Len treatment—XG-7: 0.043 and H929: 0.045). **g** Graphical representation of the in vivo tumour xenograft and BTZ treatment experiments performed using mice engrafted with VC and *PLUM* OE KMS11 cells. Created in BioRender. Deka, K. (2025) https://BioRender.com/ptd9zro. **h** Tumour growth plot depicting average tumour volume (mean ± SEM) from mice engrafted with control (VC) and *PLUM* OE KMS11 cells followed by treatment with DMSO and BTZ as shown in this figure (**g**). ($N = 5$ mice/group; two sided multiple *t*-test; $p$-values—Control +DMSO versus Control + BTZ: 0.0001, Control + DMSO versus PLUM OE + DMSO: 0.0004, PLUM OE + DMSO versus PLUM OE + BTZ: 0.0001 and PLUM OE + DMSO versus Control + BTZ: 0.0002. **i** Kaplan–Meier survival plot of mice xenografts as mentioned in (2 h). Two-sided log-rank (Mantel–Cox) test; $p$-values—Control + DMSO versus Control + BTZ: 0.0045, Control + BTZ versus *PLUM* OE + BTZ: 0.005, Control + DMSO versus *PLUM* OE + DMSO: 0.036.

Fig. 5c and Supplementary File 3). Specifically, the prediction model showed interaction of EZH2 with 994, 1000, 1001, 1002, 1003, 1010, 1011 atoms on *PLUM,* which are part of exon7 (Supplementary Table 5). Previous studies have also indicated some potential RNA binding sites on mammalian PRC2 subunits, including residues 342–368 in an unstructured region of mouse EZH2 (phospho-mimic at residue 345), N-terminal residues from 32 to 42, and C terminal unstructured region from 489 to 494 in human EZH2 (in vitro binding data with G- quadruplex RNA)[43–46] (Fig. 4f). Additionally, earlier reports have suggested the disordered region of EZH2 is important for RNA recognition and interaction[47–52]. Hence, we proceeded to validate the region on EZH2 protein having maximum potential to interact with *PLUM*.

Having confirmed the role of *PLUM* on EZH2 phosphorylation (at T345 residue) and activity (Fig. 3l), we investigated the corresponding role of pEZH2(T345) in binding *PLUM*. The pull-down data revealed reduced binding of EZH2 to *PLUM* in $R_0$3306 (CDK1/2 inhibitor) treated cells compared to DMSO (vehicle control) in both KMS11 and BTZ-R 8226 cell lines (Fig. 4g and Supplementary Fig. 5d). Next, we performed alanine scanning mutagenesis within the short, structured region of EZH2 from 32 to 42 residues (HA-Mut1, HA-Mut2 and HA-Mut3) and employed NAAIRS (asparagine–alanine–alanine–isoleucine–arginine–serine)[53] sequence substitution to mutate the long unstructured region of EZH2 from 489 to 494 (HA-Mut4) (Fig. 4f). We found that HA-Mut2, HA-Mut3 and HA-Mut4 lost their ability to interact with *PLUM* compared to wild-type HA-EZH2 (Fig. 4h). The disordered region mutant, HA-Mut4 (PRKKKR498-494NAAIRS), showed most significant loss of its ability to interact with *PLUM*. This was further verified via RNA-EMSA using truncated Mut4-EZH2 (E317-PRKKR489-494NAAIRS-V522) protein, which showed reduced binding with biotinylated exon7 region of *PLUM* compared to WT EZH2 (E317–V522). We also performed RNA-EMSA for WT EZH2 (E317–V522) and biotinylated exon7 region of *PLUM* in the presence of 200× molar excess of unlabelled exon7 region of *PLUM*, which competitively attenuated the binding of EZH2 to labelled exon7 region of *PLUM*. These findings confirm the specificity of the binding between EZH2 and exon7 of *PLUM* (Supplementary Fig. 5e, f).

To explore whether the EZH2 mutants failing to interact with *PLUM* could still form the PRC2 complex with SUZ12 and EED, Co-IP assays were performed in HA-WT EZH2 and HA-mutant EZH2 OE cells. While WT HA-EZH2 and HA-Mut1 EZH2 showed interaction with SUZ12 and EED, this interaction was reduced for HA-Mut 2, HA-Mut 3 and HA-Mut 4 EZH2 (Fig. 4i). These findings suggest that the mutants unable to bind/interact with *PLUM* also lose their interaction with SUZ12 and EED, crucial for PRC2 complex formation. Collectively, our data indicate *PLUM* recruits PRC2 core complex factors by binding to EZH2 and suggest that this RNP complex is critical for enhancing PRC2-mediated histone trimethylation to promote myeloma functions.

## Disruption of *PLUM*-EZH2 interaction using steric ASOs leads to abrogation of drug resistance in MM

To explore whether the *PLUM*-EZH2 complex can be potentially targeted using RNA-based therapeutics like steric ASOs, we analysed the in-silico docked models of FL-*PLUM*-EZH2. Based on ITScorePP and ITScorePR[54], top three docked models were selected to identify the putative interacting regions on *PLUM* for targeting via steric ASOs (Fig. 5a–c). Ten steric ASOs (s-ASOs) were designed against all the interacting regions between EZH2 and *PLUM* (inclusive of all exons): s-ASO-g2 against exon 1, s-ASO-g3 against exon 9, s-ASO-g4 against exon 4 (Model 1-Fig. 5a); s-ASO-g6 against exon 4, s-ASO-g7 against exon 5, s-ASO-g8 against exon 5, s-ASO-g9 against exon 9 (Model 2–Fig. 5b); s-ASO-g11 against exon 6-exon 7 junction, s-ASO-g12 against exon 7 and s-ASO-g14 against exon 8 (Model 3–Fig. 5c). All our steric ASOs were modified with 2'-O-Methoxyethyl/locked nucleic acid (2'MOE-LNA+) chemistry (Fig. 5d). Furthermore, all the steric ASOs were tested alongside negative control (NC-ASO) for their activity to validate our in-silico docking data through RIP-qPCR using EZH2 antibody (Supplementary Fig. 6a and Fig. 5e). The results showed s-ASO-g11 and s-ASO-g12 targeting exon6-exon7 junction and exon7 respectively are most effective in disrupting the *PLUM*-EZH2 complex, whose treatment demonstrated maximum decline in fold enrichment for *PLUM* in EZH2 RIP-qPCR results compared to other s-ASOs (Fig. 5e and supplementary Fig. 6a). The targeting region for s-ASO-g11 and s-ASO-g12 matched the docked model 3 of *PLUM*-EZH2 interaction, which forms complementary pairing with fragments of exon 6-exon 7 junction and exon 7 of *PLUM* (Fig. 5c and supplementary Table 6). FL-*PLUM*-EZH2 model 3 could also recapitulate the Exon 7-*PLUM*-EZH2 docked model (Fig. 4e). For both the models, the residues between 489 and 494 in EZH2 interact with similar atoms on exon 7 of *PLUM* (Interface file provided in supplementary Data 2).

We next assessed the selected ASOs for their ability to impede PRC2 complex formation and downstream activity, including their impact on proliferation and drug resistance in MMCLs. Consistent with our RIP-qPCR results, s-ASO-g11 and s-ASO-g12 treatment attenuated PRC2 complex formation relative to NC-ASO in both NF-κB+ MMCLs, KMS11 and BTZ-R 8226, without affecting *PLUM* levels (Fig. 5f and Supplementary Fig. 6b, c). In contrast, *PLUM* expression was reduced by ~30% in s-ASO-g4 treated cells (Supplementary Fig. 6c), potentially affecting its reduced enrichment in RIP-qPCR experiments (Supplementary Fig. 6a). We therefore proceeded to examine the functional effects of s-ASO-g11 and s-ASO-g12 in KMS11 and BTZ-R 8226 cells.

Treatment of MM cells with s-ASO-g11 and s-ASO-g12 reduced EZH2 activation, resulting in lowered H3K27me3 levels, which was found to be comparable to treatment using known EZH2 inhibitor (Tazemetostat) (Fig. 5g and Supplementary Fig. 6d). However, treatment with s-ASOs did not affect the stability of EED and SUZ12 protein (Fig. 5g). In line with the oncogenic functions

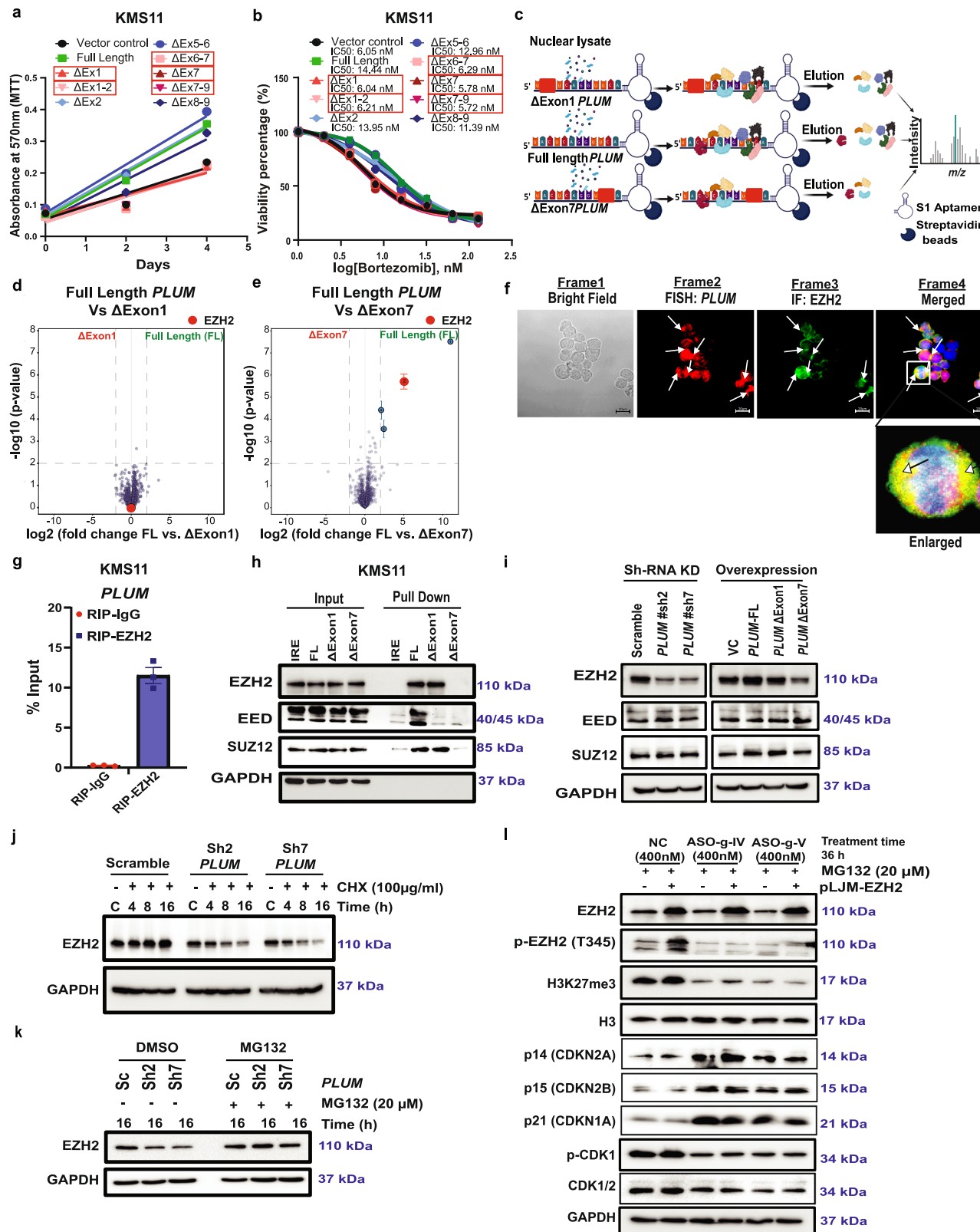

of *PLUM*-EZH2, s-ASO-g11 and s-ASO-g12 treatment reduced the proliferation of MM cells, whilst increasing the percentage of apoptotic cells (by ~10%) (Fig. 5h, i and Supplementary Fig. 6e, f). s-ASOs treated native and acquired resistant MMCLs also exhibited heightened sensitivity to both BTZ and Len (Fig. 5j, k). Interestingly, the effects of s-ASOs in resensitizing both native (KMS11) and acquired resistant MMCLs (BTZ-R 8226, Len-R

KMS11) to chemotherapeutic drugs were similar to that of EZH2 inhibitor (Tazemetostat) (Supplementary Fig. 6g, h). These findings suggest the potential use of such s-ASOs in mitigating the chemoresistance of MM, as an alternative to EZH2 inhibitors. Furthermore, our data collectively demonstrate the critical role of *PLUM*-EZH2 complex in mediating drug resistance in MM.

**Fig. 3 | PLUM interacts directly with EZH2 via exon 7 to regulate its activity and stability. a** Proliferative rate of 8 exon deletion mutants of *PLUM* overexpressed KMS11 cells compared to FL *PLUM* and vector control (VC). Red box: *PLUM* mutants with no proliferative phenotype compared to VC ($N = 2$ biological replicates). **b** Drug sensitivity $IC_{50}$ survival curve of 8 exon deletion mutants of *PLUM* over-expressed cells compared to FL-*PLUM* and VC post BTZ treatment (24 h). Red box: *PLUM* mutants sensitive to BTZ treatment compared to VC ($N = 2$ biological repli-cates). **c** Strategy used to perform RNA protein pull down assay followed by MS-LFQ analysis using FL-*PLUM*, ΔExon1 and ΔExon7 *PLUM* and nuclear lysate of KMS11 cells. Created in BioRender. Deka, K. (2025) https://BioRender.com/3kpz3l7. **d, e** Volcano plots showing differentially bound protein to FL-*PLUM* compared to ΔExon1 and ΔExon7 *PLUM*, respectively. Red dot: EZH2 protein. $N = 4$, MS analysis was done with fold change >2 and *p*-value < 0.01 determined by two-sided unpaired *t*-test. **f** Micrographs showing co-localization of EZH2 with *PLUM*. Frame1: Bright field, Frame2: RNA-FISH for *PLUM* (Fluor@Red), Frame3: IF for EZH2 (Green) and Frame4: merged frame for frames 2 and 3. Nuclei stained with DAPI (Blue). Inset/ arrows: magnified colocalization spot for EZH2 and *PLUM*. Scale bars: 20 μm; $N = 3$ biological replicates with 3 frames each. **g** RIP-qPCR validation of *PLUM* (mean ± SEM) with EZH2 and IgG antibody ($N = 3$, two-sided unpaired student's *t*-test; *p*-value: 0.0003). **h** Binding of PRC2 complex proteins (EZH2, EED and SUZ12) with FL-*PLUM*, ΔExon1 and ΔExon7 *PLUM* mutants. IRE transcript: negative control and GAPDH: loading control ($N = 2$ biological replicates). **i** Levels of EZH2, EED and SUZ12 in sh-scramble, sh-*PLUM* KD cells and FL-*PLUM*, *PLUM*-ΔExon1, *PLUM*-ΔExon7 mutants overexpressed cells ($N = 3$ biological replicates). **j** Expression levels of EZH2 in scramble, sh2-*PLUM* and sh7-*PLUM* KD cells at different time point of treatment with cycloheximide (100 μg/ml) ($N = 2$ biological replicates). **k** Levels of EZH2 protein in scramble, sh2-*PLUM* and sh7-*PLUM* KD cells at 16 h of treatment with MG132 (20 μM). DMSO: vehicle control ($N = 2$ biological replicates). **l** Levels of total EZH2, pEZH2-T345, H3K27me3, H3, CDKN2A, CDKN2B, CDKN2A, total CDK1/2, pCDK1 proteins in control and EZH2 overexpressed cells (+MG132) post treat-ment with degradative NC-ASO, ASO-gIV and ASO-gV for 36 h ($N = 3$ biological replicates).

## PLUM confers chemoresistance via activation of UPR pathway in MM

To unravel the molecular mechanisms mediating the functional effects of *PLUM*, we performed RNA-sequencing of short hairpin RNA (shRNA)-mediated *PLUM* KD and sh-scramble MM cells at day 3 post transduction (before the induction of apoptosis). Our analysis revealed 8153 differentially expressed genes (DEGs) upon *PLUM* KD (adjusted *p*-value <= 0.05), of which 4042 and 4111 were upregulated and down-regulated respectively (Fig. 6a). Several of them are involved in the regulation of metabolic processes, translational control, and ER stress pathways (Supplementary Fig. 7a, b). Notably, the UPR pathway, known to regulate the balance of apoptotic and recovery signals dur-ing cancer pathogenesis in various cancer types including MM, was enriched among downregulated DEGs upon *PLUM* KD (Fig. 6b)[55–59]. Conversely, the expression of UPR genes (*EDEM, GRP94, ATF4, Bip, CHOP*) in NF-κB+ versus NF-κB- MMCLs was significantly elevated (Supplementary Fig. 7c). Further analysis of the activation status of three master regulators of UPR and its downstream ATF6α target genes (*Bip, HERPUD1, GRP94, PDIA4*), using *PLUM* OE and sh-*PLUM* MMCLs, suggested that *PLUM* expression activates UPR pathway (Fig. 6c, d and Supplementary Fig. 7d). Additionally, s-ASO-g11 and s-ASO-g12 treated cells showed diminished activity of pIRE1α-sXBP1 and EIF2α axis of UPR pathway regulation (Supplementary Fig. 7e), implying the importance of *PLUM*-EZH2 interaction in driving che-moresistance in MM through UPR activation.

Hence, we postulated that *PLUM* promotes the proliferation and chemoresistance of MM via activation of UPR pathway. We validated this via shRNA KD of UPR master regulators in *PLUM* OE KMS11 cells, which attenuated the pro-proliferative and chemoresistance effects of *PLUM*. (Fig. 6e–g and Supplementary Fig. 7f–h). Additionally, shRNA KD of IRE1α in BTZ and Len resistant MMCLs, which displayed distinct induction of the pIRE1α-sXBP1 axis of UPR pathway (Fig. 6h) also re-sensitized both cell lines to BTZ and Len treatment, respectively (Supplementary Fig. 7i and Fig. 6i, j). However, in all three resistant cell lines, pEIF2α level remained low, which might be due to enhanced proliferative stress in the resistant cell lines. While previous reports have suggested the enhanced activation of eIF2α phosphatases in BTZ resistant conditions can cause MM cells to enter the quiescent phase[60], further studies will be needed to confirm the observed phenomenon. Overall, our findings highlight the crucial role of UPR pathway activa-tion in augmenting the oncogenic functions of *PLUM* in MM.

## PLUM augments myeloma progression and chemoresistance via PRC2 mediated hypermethylation of FOXO3/ZFP36

Next, to identify *PLUM* regulated targets of PRC2 complex that mediate chemoresistance in myeloma, we performed H3K27me3 ChIP-seq in sh-scramble versus sh-*PLUM* KD KMS11 cells. Notably, *PLUM* KD induced the global downregulation of H3K27me3 marks at the transcription start sites (TSS) of EZH2 bound genes (Fig. 7a). Further integration of the lost H3K27me3 peaks with the nearest upregulated DEGs from sh-*PLUM* RNA-seq data revealed several genes enriched in processes related to mRNA metabolism, cell cycle control and protein stability/apoptosis (Fig. 7b and Supplementary Fig. 8a). Since our earlier observations indicate *PLUM* mediates chemoresistance via activation of UPR pathway, (Fig. 6), we postulated that this may be regulated via the epigenetic functions of EZH2. To select putative *PLUM* regulated EZH2 targets involved in UPR-mediated chemoresis-tance, we focused on genomic loci exhibiting reduced H3K27me3 marks at the TSS of cancer associated genes. We identified two uncharacterized targets of EZH2, *FOXO3* and *ZFP36*, which have been previously implicated in the repression of UPR pathway in other cancer types[61,62]. Both targets display reduced H3K27me3 marks at their gene loci following *PLUM* KD (Fig. 7c), indicating they could be potentially regulated by *PLUM*-EZH2 complex.

In line with this, *PLUM* OE in KMS11 cells induced H3K27 tri-methylation at the TSS of *FOXO3* and *ZFP36*, which in turn repressed their protein expression (Supplementary Fig. 8b–d). Conversely, sh-*PLUM* KD or overexpression of *PLUM* ΔExon1 and ΔExon7 resulted in the reduced H3K27me3 enrichment of both targets, along with the induction of their protein expression (Supplementary Fig. 8d–f). These findings demonstrate that *PLUM* regulates *FOXO2* and *ZFP36* expres-sion by enhancing the H3K27 trimethylation activity of PRC2, plausibly via its interaction with PRC2 core complex factors, EZH2 and EED.

We next investigated whether both *PLUM*-EZH2 regulated targets are involved in the chemoresistance of MM. Here, we found that BTZ and Len resistant MMCLs displayed elevated H3K27 trimethylation of both targets, along with their reduced protein expression levels, compared to parental, sensitive cells (Fig. 7d–f). This is consistent with their elevated expression of *PLUM*. Notably, treatment of resistant MMCLs with s-ASO-g12 resulted in the loss of H3K27me3 enrichment at both gene targets, which correlated with their induced protein expression (Fig. 7g–i). These data validated the role of *PLUM*-EZH2 complex in regulating *FOXO2* and *ZFP36* expression in resistant MMCLs.

To further verify the role of *FOXO3* and *ZFP36* in modulating UPR-mediated chemoresistance in MM, we examined the activation of UPR pathway regulators following shRNA-mediated KD of both genes in parental KMS11 cells. *FOXO3* or *ZFP36* KD induced both the IRE1α-sXBP1 and eIF2α axis of UPR pathway, correlating with the enhanced proliferation and increased resistance of shRNA targeted cells to BTZ treatment relative to sh-scramble cells (Fig. 7j, k and Supple-mentary Fig. 8g–j). These findings are in line with our earlier findings on the regulation of UPR pathway driven chemoresistance by *PLUM* (Fig. 6).

We subsequently evaluated the therapeutic efficacy of our designed s-ASOs in vivo, using subcutaneous tumour xenografts of

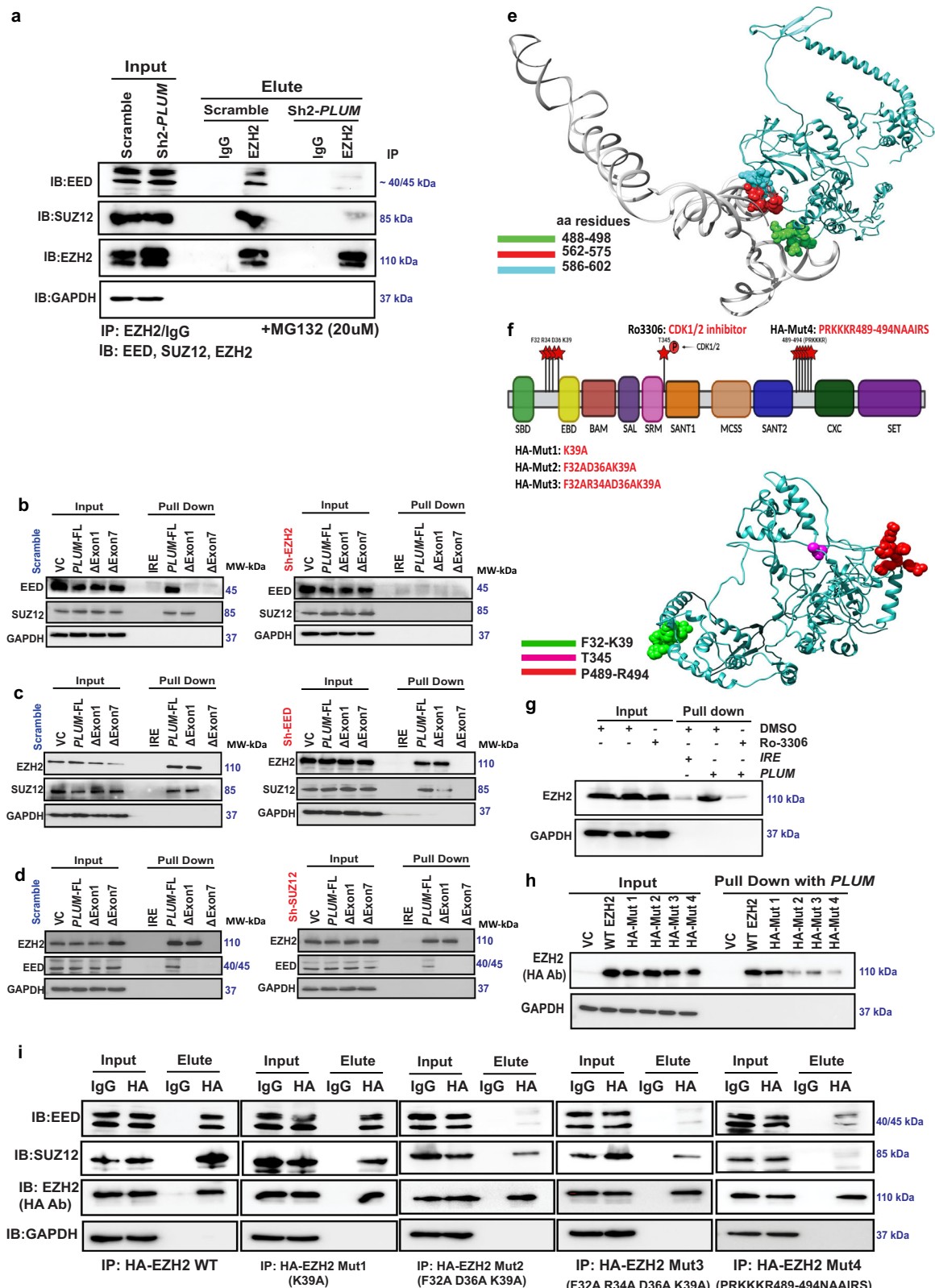

*PLUM* OE KMS11 cells. Consistent with our in vitro data, s-ASO-g12 treated xenografts displayed reduced tumour growth compared to ASO-NC treated xenografts. Furthermore, combination treatment with s-ASO-g12 and BTZ led to further reduction in tumour volume compared to s-ASO-g12 or BTZ treatment alone (Fig. 7l and Supplementary Fig. 9a, b). In accordance with tumour load, *PLUM* OE engrafted mice treated with combination of s-ASO-g12 and BTZ showed significantly

higher survival rate compared to s-ASO-g12, BTZ and ASO-NC treatment alone groups (Fig. 7m). These findings confirm the potential of targeting *PLUM*-EZH2 complex as a strategy to mitigate the therapeutic resistance of MM patients. Additionally, we reveal a regulatory mechanism for this RNP complex in MM, acting through the enhanced H3K27me3 activity of PRC2 complex, which in turn drives chemoresistance via the *FOXO3/ZFP36/*UPR pathway axis (Fig. 7n).

**Fig. 4 | *PLUM*-EZH2 interaction is important for PRC2 complex formation.**
**a** Levels of EED and SUZ12 protein co-immunoprecipitated by EZH2 antibody in scramble and sh2-*PLUM* KD KMS11 cells. (+MG132-20 μM). IgG: IP control; GAPDH: input protein loading control (*N* = 3 biological replicates). **b** Levels of EED and SUZ12 protein pulled down by FL-*PLUM*, ΔExon1 *PLUM*, ΔExon7 *PLUM* transcripts from nuclear lysate of scramble and sh-*EZH2* KD KMS11 cells. IRE: negative control and GAPDH: input protein loading control (*N* = 1). **c** Levels of EZH2 and SUZ12 protein pulled down by FL-*PLUM*, ΔExon1 *PLUM*, ΔExon7 *PLUM* transcripts from nuclear lysate of scramble and sh-*EED* KD KMS11 cells. IRE: negative control and GAPDH: input protein loading control (*N* = 1). **d** Expression levels of EZH2 and EED protein pulled by FL-*PLUM*, ΔExon1 *PLUM*, ΔExon7 *PLUM* transcripts from nuclear lysate of scramble and sh-*SUZ12* KD KMS11 cells. IRE: negative control and GAPDH: input protein loading control (*N* = 1). **e** Docked structure of exon7-*PLUM* and EZH2 protein. Interacting residues on EZH2 marked as Green: 488–498 aa; Red: 562–575 aa and Blue: 586–602 aa. **f** Graphical representation of the EZH2 protein with all the domains marked in different colors. (*) sign: reported residues having RNA-binding

role. Mutants generated: HA-Mut1: K39A, HA-Mut2: F32AD36AK39A, HA-Mut3:-F32AR34AD36AK39A and HA-Mut4: PRKKKR489-494NAAIRS. Tertiary structure of EZH2 protein with marked RNA-interacting residues in different color code. Green: F32-K39, Magenta: T345 (CDK1/2 phosphorylation site), Red: P489-R494 (disordered region). Created in BioRender. Deka, K. (2025) https://BioRender.com/f40k21z. **g** Levels of EZH2 protein pulled down with FL-*PLUM* in +/− CDK1 inhibitor (R$_0$3306) cells. DMSO: vehicle control and GAPDH: input protein loading control (*N* = 2 biological replicates). **h** Levels of wild type HA-EZH2 and HA-EZH2 mutants (HA-Mut1: K39A, HA-Mut2: F32AD36AK39A, HA-Mut3:F32AR34AD36AK39A and HA-Mut4: PRKKKR489-494NAAIRS) pulled down with FL-*PLUM*. IB: HA antibody and GAPDH: input protein loading control (*N* = 2 biological replicates). **i** Levels of EED, SUZ12 and EZH2 protein co-immunoprecipitated by HA antibody in wild type HA-EZH2 and HA-EZH2 mutants (HA-Mut1: K39A, HA-Mut2: F32AD36AK39A, HA-Mut3:F32AR34AD36AK39A and HA-Mut4: PRKKKR489-494NAAIRS) overexpressed KMS11 cells. IgG: IP control and GAPDH: input protein loading control (*N* = 2 biological replicates).

## Discussion

Chemoresistance in MM contributes to the increased relapse and poorer survival of patients, which further limits their treatment options. This multifaceted challenge underscores the need for better treatment approaches that explore targetable biomarkers associated with drug resistance. Here, we identified a myeloma-associated lncRNA, *PLUM*, that mediates chemoresistance through interacting with EZH2. We further revealed *PLUM* expression is enriched in NF-κB+ mutant high-risk MM subtypes and VRd non-responsive patient samples, implicating its potential as a biomarker or therapeutic target for treatment resistant MM.

Elevated EZH2 levels in aggressive MM subtypes have been correlated with the dysregulated expression of PRC2-interacting lncRNAs[29,63,64]. Yet detailed mechanistic studies deciphering the role of lncRNA-EZH2/PRC2 complex(es) in myeloma progression and chemoresistance are lacking. We report biochemically how *PLUM*-EZH2 interaction leads to enhanced PRC2 activity, promoting chemoresistance via the activation of UPR pathway, in addition to several targeted therapy-based studies confirming the involvement of cross talk between UPR pathway regulators (IRE1α, eIF2α, ATF6α) in driving MM pathogenesis and chemoresistance[58,59,65–70]. Our results offer an insight into the probable molecular cascade of events involved upstream of UPR pathway activation via lncRNA-mediated augmentation of EZH2 activity in MM. Though previous studies have exploited UPR pathway markers (IRE1-XBP1/PERK-eIF2α) as a treatment target in MM both in vitro and in vivo, our findings on the EZH2 mediated regulation of UPR pathway have uncovered two additional EZH2 regulated genes, *FOXO3* and *ZFP36*, that activate UPR pathway via *PLUM*-EZH2 interaction to drive the therapeutic resistance of MM.

Since EZH2 is a non-canonical RBP, its consensus RNA-binding site remains controversial to date. However, in vitro studies demonstrate its higher affinity for G-quadruplex RNA, suggesting its RNA binding capability depends on the nature of the RNA[44]. The phosphorylation of EZH2 at T345 residue and its intrinsically disordered region have earlier been shown to facilitate interaction with RNA[43,44,71]. Consistent with this, our study demonstrates these sites are critical for binding of EZH2 to exon7 of *PLUM*, which contains several G-quadruplex sequences in its secondary structure.

Recently, PRC2 complex with mutant EZH2 (PRKKKR494–499NAAIRS) has been shown to be deficient in both RNA binding and H3K27me3 activity[44,46,71]. Interestingly, our data indicates the specificity of this region for *PLUM*-EZH2 interaction, in turn regulating PRC2 complex formation and H3K27me3 activity (Fig. 4h, i). Furthermore, the disruption of *PLUM*-EZH2 complex via steric ASOs suggest that *PLUM* may act as a bridge (either directly assisting in chromatin interaction or indirectly assisting in PRC2 complex formation) between PRC2 and chromatin in MMCLs. Our findings therefore suggest that *PLUM* is required for PRC2 complex formation and its

localization to specific target loci for repression, critical for driving chemoresistance in MM.

The tumorigenic and chemoresistance functions of EZH2 in MM have prompted several pre-clinical studies on the use of EZH2 catalytic inhibitors as a treatment option[36,72–74]. Though these initial pre-clinical results on the use of EZH2 inhibitors in MM are encouraging, some reports also demonstrate that MM cells can develop resistance to such inhibitors like tazemetostat (EPZ-6438) through epigenetic mechanisms[75]. In other haematological malignancies like DLBCL, these cancer cells have also been found to acquire genetic mutations to overcome the sensitivity to EZH2 inhibitors[76]. Additionally, unlike other cancers, EZH2 expression in MM is mostly regulated by epigenetic mechanisms rather than gain-of-function mutations[34,35,77–79]. Thus, exploring EZH2 as a potential therapeutic target demands detailed investigation of its regulatory factors involved in the functional dynamics of MM. Several clinical trials are also ongoing involving EZH2 inhibitors in haematological malignancies (NCT02220842, NCT03460977, NCT02395601, NCT02900651), but it appears that the drug(s) are not evaluated in MM patients yet[80]. Hence, our data on characterization of the myeloma-associated expression of *PLUM* and its role in enhancing EZH2 stability and activity towards chemoresistance suggest its promising potential and future development as a biomarker and/or therapeutic target for MM. Moreover, the similar effects of *PLUM*-EZH2 targeting s-ASOs with that of EZH2 inhibitor (tazemetostat) in the destabilization of EZH2 activity and abrogation of chemoresistance support the possible use of *PLUM* as an alternative target to overcome the resistance mechanisms associated with known EZH2 catalytic inhibitors in MM. Herein, given the unknown role of *PLUM* in normal physiological processes and key roles of EZH2/PRC2 complex in normal immunoglobulin VDJ recombination in pre-B-cells and formation of germinal centre (GC) in B cells[81,82], our strategy of using mixmer ASOs to interrupt specific PLUM-EZH2 interaction and its downstream oncogenic functions could be more beneficial compared to using global inhibitors of EZH2 like Tazemetostat. However, a competitive evaluation of these steric ASOs with available EZH2 inhibitors requires further studies.

Collectively, our study confirmed a epigenetically regulated lncRNA, *PLUM*, which interacts with EZH2, facilitating PRC2 complex formation and epigenetic functions to promote UPR pathway mediated chemoresistance in MM. Additionally, we exploited the use of mixmer ASOs as a potential therapeutic tool to target specific lncRNA-EZH2 interactions, providing a unique strategy to disrupt the oncogenic functions of PRC2-associated RNPs.

## Methods

This research complies with all relevant ethical regulations. Consent and approval for the use of clinical samples was obtained from the Nanyang Technological University Institutional Review Board,

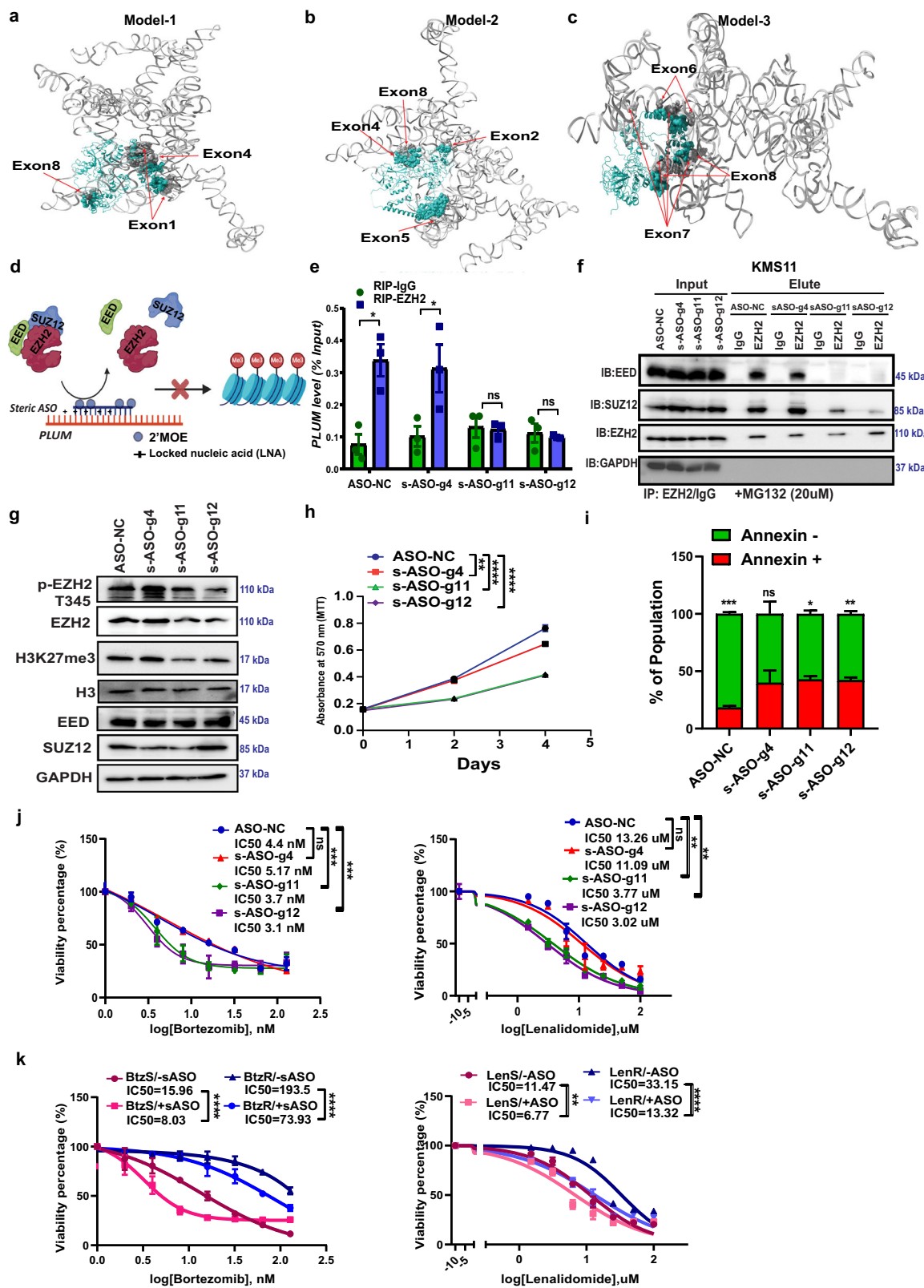

Singapore under the protocol, Investigating the role of oncogenic transcription factors, lncRNAs, enhancers and enhancer-regulated genes in cancer development and chemoresistance, in accordance with the Human Biomedical Research Act requirements. Animal studies were approved by Institutional Animal Care and Use Committee of Nanyang Technological University Singapore (NTU-ARF; AUP: A21070). Tumour size/burden was monitored every 2–3 days. The maximal tumour volume approved by the IACUC was 2000 mm³. Once this limit is exceeded, the mice were immediately euthanised.

## Antibodies and reagents

**Antibodies for western blotting.** NFKB2 (1:1000; 3017, CST), p-EZH2 (1:1000; PA5114574, Thermo), EZH2 (1:1000; 491043, Thermo), EED (1:1000; PA534430, Thermo), SUZ12 (1:1000; 3737S, CST), p-IRE1α

**Fig. 5 | Steric ASO-mediated disruption of *PLUM*-EZH2 interaction abrogates drug resistance in MM. a–c** Top three docked model of FL-*PLUM* and EZH2 with interacting residues highlighted in space-filling format. Red arrows: Interacting exons of *PLUM* with EZH2 for all three models. **d** Hypothetical working model of steric ASOs (LNA + 2′MOE modified) inhibiting H3K27me3 activity through hindering the interaction between *PLUM*-EZH2. Created in BioRender. Deka, K. (2025) https://BioRender.com/p7si5d6. **e** RIP-qPCR validation of *PLUM* with EZH2 and IgG antibody in KMS11 treated with NC-ASO and steric ASOs (*N* = 3, mean % input ± SEM is plotted, *p*-values determined by two-sided unpaired student's *t*-test; *p*-value−NC: 0.011, s-ASO-g4: 0.048, s-ASO-g11 and s-ASO-g12 *p*-value non-significant). **f** Levels of EED, SUZ12 and EZH2 co-immunoprecipitated by EZH2 antibody in NC-ASO and steric ASOs treated KMS11 cells. (+MG132-20 μM); IgG: IP control (*N* = 3 biological replicates). **g** Levels of EZH2, p-EZH2 T345, H3K27me3, H3, EED and SUZ12 proteins in NC-ASO and steric ASOs treated KMS11 cells (*N* = 5 biological replicates). **h** Proliferation rate (mean ± SEM) of KMS11 cells treated with NC-ASO and steric ASOs till day4 (*N* = 3, *p*-values determined by two-way ANOVA; *p*-values−NC Vs s-

ASO-g4: 0.005, NC Vs s-ASO-g11: 0.0005 and NC Vs s-ASO-g12: 0.0005). **i** The percentage (mean ± SEM) of annexin+ cells and annexin−cells post treatment with steric ASOs (*N* = 4, *p*-values determined by two-sided multiple *t*-test; *p*-values - NC-ASO: 0.0001, s-ASO-g4: non-significant, s-ASO-g11: 0.01, s-ASO-g12: 0.0039). **j** Drug sensitivity IC$_{50}$ survival curve (mean ± SEM) for steric ASOs treated KMS11 cells in response to BTZ treatment for 24 h and Len treatment for 4 days, respectively (*N* = 3, *p*-values determined by two-way ANOVA; BTZ treatment *p*-values−NC-ASO Vs s-ASO-g4: non-significant, NC-ASO versus s-ASO-g11: 0.0001, NC-ASO versus s-ASO-g12: 0.0001 and Len treatment *p*-values−NC-ASO Vs s-ASO-g4: non-significant, NC-ASO Vs s-ASO-g11: 0.0007, NC-ASO versus s-ASO-g12: 0.003). **k** Drug sensitivity IC$_{50}$ survival curve (mean ± SEM) for steric ASOs treated BTZ S/BTZ R cells in response to BTZ treatment for 24 h and Len S/Len R cells in response to Len treatment for 4 days respectively (*N* = 3, *p*-values determined by two-way ANOVA; BTZ S/−ASO versus BTZ S/+ASO: 0.0001, BTZ R/−ASO versus BTZ R/+ASO; 0.0001, LenS/−ASO versus LenS/+ASO: 0.005, LenR/−ASO versus LenR/+ASO: 0.0001).

(1:1000; ab243665, Abcam), IRE1α (1:500; sc-390960, Santacruz), sXBP1 (1:500; sc-133132, santacruz), p-eIF2α (1:1000; 9721, CST), eIF2α (1:500; sc-133132, santacruz), ATF6α (1:500; sc-166659, santacruz), GAPDH (1:10000; sc-32233, santacruz), HA antibody (1:1000; sc-7392, Santacruz), pCDK1/2 (1:1000; 4539S, CST), CDK1/2 (1:500; sc-53219, santacruz), H3K27me3 (1:1000; 9733S, CST), H3 (1:1000; sc-517576, santacruz), FOXO3 (1:500; sc-48348, santacruz), ZFP36 (1:500; sc-374305, santacruz), anti-rabbit IgG-HRP conjugated secondary antibody (1:10000; sc-2357, santacruz) and anti-mouse IgG-HRP conjugated secondary antibody (1:10000; sc-516102, santacruz).

**Antibodies for ChIP.** H3K27me3 (9733S; CST); 4 μg/8 × 10$^6$ cells, EZH2 (491043; Life technologies); 8 μg/8 × 10$^6$ cells, HA Ab (901503; Genomax); 4 μg/8 × 10$^6$ cells, Anti-rabbit IgG (7074S; CST); 4 μg/8 × 10$^6$ cells.

**The inhibitors used.** Bortezomib for cell culture use (5.04314.0001CN; Merck Sigma), Bortezomib for in vivo use (HY-10227; MedChem Express), Lenalidomide for cell culture and in vivo use (HY-A0003; MedChem Express), Ro-3306−CDK1/2 inhibitor (HY-12529; MedChem Express), MG132 (ab141003; Abcam), Cyclohexamide (C1988; Sigma).

## Cell culture

Human MMCLs, including KMS11, U266, RPMI8226, JJN3, XG7 and H929, were gifts from Prof. Leif Bergsagel (Mayo Clinic, Scottsdale, AZ, USA). The MM1.S cell line was obtained from ATCC, while LP1 and MOLP8 were obtained from the German Collection of Microorganisms and Cell Cultures. BTZ-resistant (BTZ-R RPMI8226), lenalidomide-resistant (Len-R KMS11) and their parent sensitive cells were obtained from Dr. W.J. Chng (Cancer Science Institute of Singapore, Singapore)[83,84]. Isogenic lenalidomide-resistant (LenR KMS11) cell line was established in the presence of escalating doses of lenalidomide over an extended period. Whole-exome sequencing was performed to examine for genetic alterations in LenR cells[85]. All the cells were authenticated by short tandem repeat profiling. Cell lines were tested to be mycoplasma-free using Mycoplasma PCR Detection Kit (G238; Abm) prior to experiments. MM cell lines were cultured in RPMI 1640 media with 2.05 mM ʟ-Glutamine (SH30027.01; Hyclone). XG7 was supplemented with 2 nM IL-6 (200-06-20UG; PeproTech) while H929 was supplemented with 0.05 mM β-mercaptoethanol (444203; Sigma). HEK 293 T cells were cultured in Dulbecco's modified Eagle's medium (DMEM) with 4 mM glucose (high glucose) (SH30243.01; Hyclone) and passaged with 0.25% Trypsin/EDTA (25200072; Gibco). All cells were supplemented with 10% Fetal Bovine Serum (F7524; Sigma) and cultured in a humidified 5% CO$_2$ incubator at 37 °C.

## Western blotting

Cell lysates were prepared using ice-cold RIPA lysis buffer (10 mM TrisHCl, pH 8.0, 1 mM EDTA, 0.5 mM EGTA, 1% Triton ×-100, 0.1%

Sodium Deoxycholate, 0.1% SDS, 140 mM NaCl, 0.5 mM DTT, 0.3 mM NaVO3) with 1 × protease inhibitor cocktail (5056489001; Merck). 30−50 μg of total protein was resolved on 8–14% acrylamide gel (as per experimental requirement) under reducing conditions and transferred to 0.2 μm polyvinylidene difluoride membrane in Tris-glycine buffer. Membrane was blocked for 1 h in 5% non-fat milk, in Tris-buffered saline-tween 20 (TBST) buffer/5% BSA in TBST. Blocked membrane was washed 3 times with TBST buffer. Probing with primary antibody (ab) was conducted for 2 h at room temperature (RT)/overnight at 4 °C, followed by 3× washing with TBST for 10 min each. Incubation with HRP conjugated secondary Ab was performed for 1 h at RT, followed by 3× washing with TBST for 10 min each, and further developed using Clarity Western ECL substrate (1705062; BioRad). Images were taken with Chemidoc XRS imaging system (BioRad) and quantified using GelQuant.NET software provided by biochemlabsolutions.com (CA, USA).

## Gene expression analysis

For gene expression analysis, cells were harvested for RNA using TRIzol reagent (15596026; Invitrogen) following manufacturer's instructions. 500 ng−1 μg of RNA was used to prepare cDNA using TOYOBO ReverTra Ace qPCR RT master mix kit (FSQ-301; TOYOBO) as per manufacturer's protocol. Quantitative polymerase chain reaction (qPCR) was performed using SYBR green chemistry (Biorad system) using primers as mentioned in Supplementary Table 2. GAPDH was taken as internal control.

## Lentiviral production and transduction

Two million 293T cells (maintained in DMEM supplemented with 10% FBS) were seeded in a poly-L-lysine-coated 100 mm dish 1 day prior to transfection so that it reaches a confluency of ~70% at the day of transfection. Cells were then transiently transfected using the four-plasmid system: 2.5 μg pRSV-Rev, 6.5 μg pMDLg/pRRE, 3.6 μg pCMV-VSVG and 10 μg of plasmid of interest (CRISPR constructs/PLKO.1-shRNA constructs/LeGO-lnc *PLUM* overexpression constructs) and topped up to final volume of 500 μl with distilled H2O. The plasmid mix was then added drop wise to 500 μl of 2× HBS buffer; pH 7.07 (280 mM NaCl, 50 mM HEPES, 1.5 mM Na2HPO4, 10 mM KCl, 12 mM Dextrose) and incubated for 20 min at RT followed by adding to the cells dropwise. After 6−8 h of incubation, medium was discarded, and cells were washed 3 times with PBS followed by replenishment with 8 ml of fresh medium (70% RPMI and 30% DMEM). The supernatant containing lentiviral particles was collected at 24 h post-transfection and passed through a 0.45 μm syringe filter before being stored at −80 °C.

For transduction, spin infection method was adopted from[86]. In brief, 5 × 10$^5$ of target cells suspended in 500 μl of complete medium were mixed with 2× volume of lentivirus. Polybrene

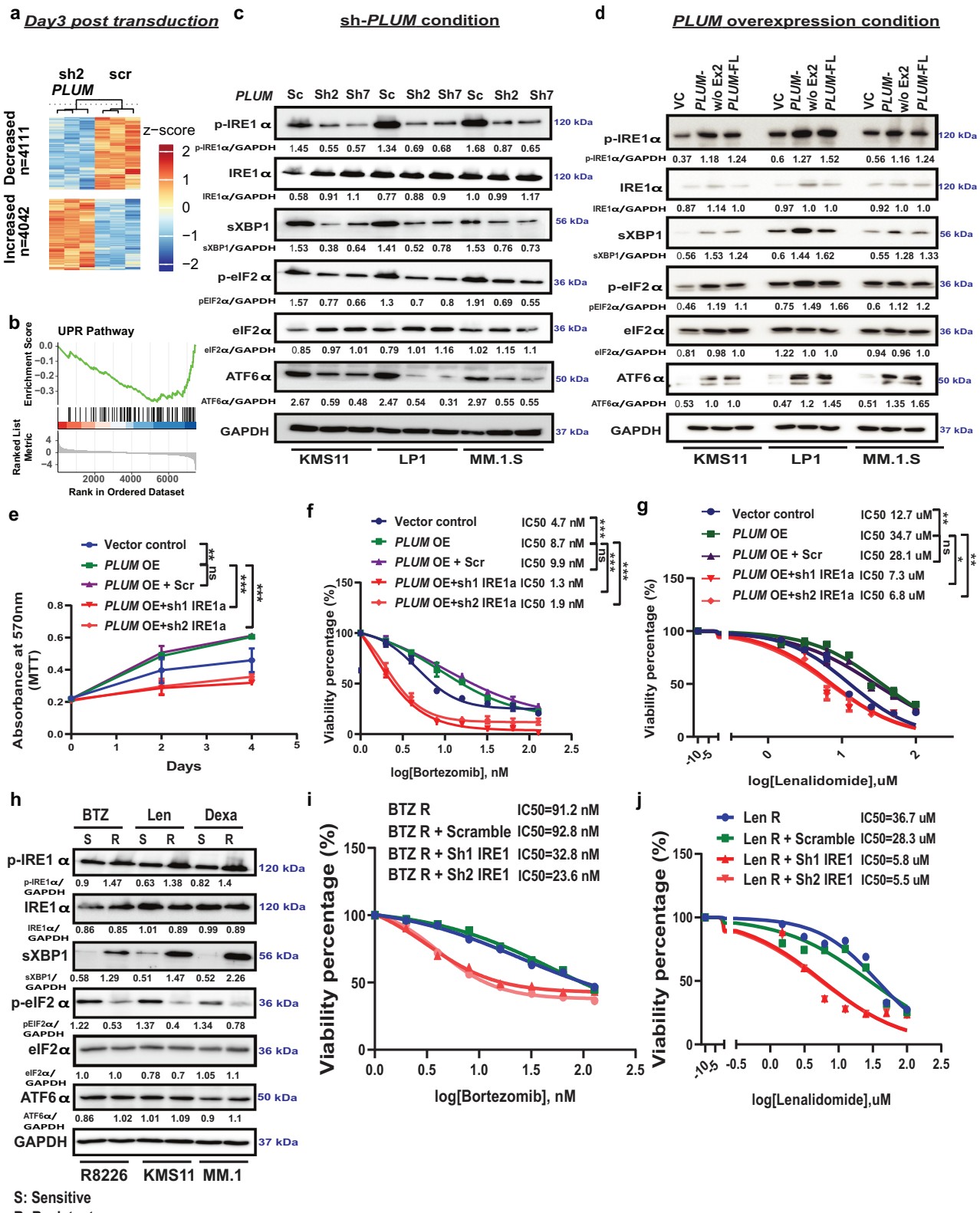

was added to a final concentration of 8 µg/ml. The cells were then centrifuged at $400 \times g$ for 45 min at RT and gently resuspended in the same tube to be seeded in a 6-well plate. After 24 h, virus was removed, and cells were resuspended in fresh medium. After another 24 h, cells were selected using specific drug pressure for 24–72 h followed by recovery of 3–7 days (as per different plasmid system used).

**CRISPR-Cas9 deletion of super-enhancer (SE) region**

For *PLUM* SE deletion, gRNAs were designed using the Broad Institute GPP sgRNA Designer/CRISPick[87] to target the SE region proximal to *PLUM*. Forward guide (g1) sequence was cloned into TLCV2 lentiviral vector (Addgene plasmid #99376, a gift from Kristen Brennand) and reverse guide (g2) sequence was cloned into pHIV-dTomato lentiviral vector (Addgene plasmid # 21374) as previously described[88]. The

**Fig. 6 | *PLUM* confers chemoresistance via activation of UPR pathway. a** Heat map representing the differential gene expression (DEGs) for RNA sequencing 3 days post transduction in sh-scramble and sh2-*PLUM* KD KMS11 cells ($n = 3$; 2-sided Wald test; adjusted *p*-value <= 0.05). **b** GSEA for UPR signatures using down DEGs from *PLUM* KD RNA-seq in KMS11 cells. Enrichment Score and Ranked List Metric are depicted. NES: −1.68; FDR: 0.00353. **c** Levels of UPR master regulators (p-IRE1α, sXBP-1, p-eIF2α and ATF6α) in sh-scramble and *PLUM* KD NF-κB + MMCLs (KMS11, LP1, MM.1.S). Average quantified values marked below each blot ($N = 3$ biological replicates and quantification was done using GelQuant.NET). **d** Level of UPR master regulators in *PLUM* overexpressed cells relative to vector control in NF-κB+ mutant MMCLs (KMS11, LP1, MM.1.S). Average quantified values marked below each blot ($N = 3$ biological replicates and quantification was done using Gel-Quant.NET). **e** Proliferation rate (mean OD ± SEM) of *PLUM* overexpressed KMS11 cells transduced with sh-scramble and sh1/sh2 IRE1α ($N = 3$, *p*-values determined by two-way ANOVA; *p*-values−VC versus *PLUM* OE: 0.0008, *PLUM* OE versus *PLUM* OE + scramble: non-significant, *PLUM* OE versus *PLUM* OE + sh1 IRE1α:

0.0002, *PLUM* OE versus *PLUM* OE + sh2 IRE1α: 0.0003). **f, g** Drug sensitivity IC$_{50}$ curve (mean ± SEM) for *PLUM* overexpressed KMS11 cells transduced with sh-scramble and sh1/sh2 IRE1α in response to BTZ treatment for 24 h and Len treatment for 4 days respectively ($N = 3$, *p*-values were determined by two-way ANOVA; BTZ treatment: VC versus *PLUM* OE: 0.0001, *PLUM* OE versus *PLUM* OE + scramble: non-significant, *PLUM* OE versus *PLUM* OE+sh1 IRE1α: 0.0001, *PLUM* OE versus *PLUM* OE + sh2 IRE1α: 0.0001; Len treatment: VC versus *PLUM* OE: 0.001, *PLUM* OE versus *PLUM* OE + scramble: non-significant, *PLUM* OE versus *PLUM* OE + sh1 IRE1α: 0.0001, *PLUM* OE versus *PLUM* OE + sh2 IRE1α: 0.0001). **h** Levels of UPR master regulators in BTZ (RPMI8226), Len (KMS11) and Dexa (MM.1 R) acquired resistant cell lines versus parental sensitive cells. Average quantified values marked below each blot. ($N = 2$ biological replicates and quantification was done using Gel-Quant.NET). **i, j** Drug sensitivity IC$_{50}$ curve for sh-scramble versus sh-IRE1α KD cells in BTZ-R 8226 and Len-R KMS11 cells in response to BTZ treatment for 24 h and Len treatment for 4 days, respectively ($N = 2$ biological replicates).

following oligos were used to generate the TLCV2-*PLUM*-g1 and dTOM-*PLUM*-g3 plasmid to make a CRISPR deletion of ~3 kb within the SE region. TLCV2-*PLUM*-g1F: CACCGAACACAGGAGGGATCACAA; TLCV2-*PLUM*-g1R: AAACTTGTGATCCCTCCTGTGTTC; dTOM-*PLUM*-g3F: CACCGTTATGACCTAGAAAGCCTAG; and dTOM-*PLUM*-g3R: AAACCTAGGCTTTCTAGGTCATAAC.

Lentiviral production and transduction of target cells was performed using protocol mentioned in above method. Cells were first transduced with lentiGuide-Hygro-dTomato g3 construct and selected with 300 ng/ml hygromycin for 10 days. The cells containing lentiGuide-Hygro-dTomato plasmids were then transduced with the TLCV2-g1 plasmid and selected with 0.5 μg/ml puromycin for 24 h followed by recovery of 3 days.

To assess super enhancer deletion efficiency of the pool of sorted cells, genomic DNA was extracted, and genotyping was performed using primers flanking the genomic target region. A shorter PCR product is expected for deleted alleles compared to the uncut control (Supplementary Fig. 1c). The PCR product for the shorter band was sequenced using Sanger sequencing to confirm the deletion of the region (Supplementary Fig. 1b). Primers used are listed in Supplementary Table 1.

### Short hairpin RNA-expressing plasmid construction
All the shRNA sequences (Supplementary Table 1) were designed manually according to the TRC design rules: http://www.broadinstitute.org/science/projects/rnai-consortium/trc-shrna-design-process and cloned into PLKO.1 plasmid (Addgene plasmid #8453). Scramble RNA sequence was used as control. Cloning was confirmed by double digestion and sequencing. Transduction was performed using lentiviral system. Knock down efficiency for *PLUM* was measured using qPCR (primer sequence in Supplementary Table 2) and for other proteins by immunoblotting.

### Rapid amplification of cDNA ends (RACE) assay
Total RNA was isolated from MMCLs (KMS11, LP1 and JJN3) using TRIzol reagent (15596026; Invitrogen) according to the manufacturer's protocol. 5′ and 3′ RACE was performed according using SMARTer kit as per manufacturer's protocol (SMARTer). In brief, pair of *PLUM*-specific primers targeting specific exons of the isoforms (predicted on the Ensembl database) were designed. 5′/3′ RACE PCR products were then ligated into the pRACE vector, and multiple clones were sequenced with M13F primer and aligned using the Unipro Ugene Alignment Editor to identify the major isoforms of *PLUM* expressed in MMCLs.

### RNA fluorescence in-situ hybridisation (RNA-FISH)
Custom Stellaris™ RNA FISH Probes were designed against *PLUM* with Fluor® Red 635 Dye by utilising the Stellaris RNA FISH Probe Designer (LGC, Biosearch Technologies, Petaluma, CA) available online

at www.biosearchtech.com/stellarisdesigner. The myeloma cells were hybridised with the *PLUM* Stellaris RNA FISH probe set labelled with Fluor® Red 635 Dye (Biosearch Technologies), following the manufacturer's instructions available online at www.biosearchtech.com/stellarisprotocols with some modifications. In brief, $3 \times 10^6$ cells were collected in tube, fixed with 3.7% formaldehyde for 10–15 min and permeabilised in 0.5% Triton ×-100 in PBS for 15 min. Cells were then washed with PBS and incubated for 5 min in Stellaris RNA FISH buffer A on ice. Hybridization of FISH probes was carried out overnight at 50 °C in 100 μL of Stellaris FISH hybridization buffer containing probe (500 nM) in a dark humidified chamber. In the next day, after washing and incubation for 30 min with FISH buffer A, cells were counterstained with DAPI (5 ng/ml in wash buffer A) for 10 min in dark. After another wash with FISH buffer B, cells were resuspended in a small drop of vectashield mounting medium to prepare the glass slide for imaging. All the steps were performed in 1.5 ml tube and centrifugation was performed at $200 \times g$ for 2 min. Imaging was performed using Carl Zeiss confocal microscope with an objective magnification of 63×.

### Immunofluorescence (IF)
Post RNA-FISH hybridization step, cells were washed and incubated in wash buffer A and wash buffer as mentioned in the above methods. After that, cells were blocked with 5% BSA in PBST for 20 min in dark. After blocking, washing was performed 2 times with PBS and cells were incubated with 500 μL of primary antibody (EZH2 Ab used: 1:500 in 5% BSA) for 1 h in dark. 1° Ab was then removed and cells were washed 2 times with PBST. Secondary antibody (GFP conjugated: 1:500 in 5% BSA) incubation was performed for 1 h in dark followed by washing 3 times with PBST. In the third wash, DAPI (5 ng/ml) was added and incubated for 10 min in dark. Thereafter, mounting, slide preparation and imaging were performed as per above method.

### Site-directed mutagenesis (SDM)
Mutagenesis was performed using Q5 site directed mutagenesis kit (E0554; NEB) following manufacturer's protocol. In brief, mutagenic primers were designed using NEBase Changer tool, with required mutation in the forward primer for all the constructs. Exponential amplification was performed using Q5 hot start high fidelity 2× master mix and 1 μl of PCR product was then taken for Kinase, Ligase, and Dpn1 (KLD) treatment, using KLD enzyme mix for 5 min, at RT. Following treatment, 5 μl of the KLD mix was transformed in NEB 5-alpha competent *Escherichia coli* cells (C2987S; NEB). Positive colonies were selected for plasmid isolation, and mutants were confirmed by sequencing, using vector specific forward and reverse primers.

### Probe preparation and RNA-protein pull down assay
To generate template for in vitro transcription, two primers were designed for each target transcript carrying a T7-tag in the forward

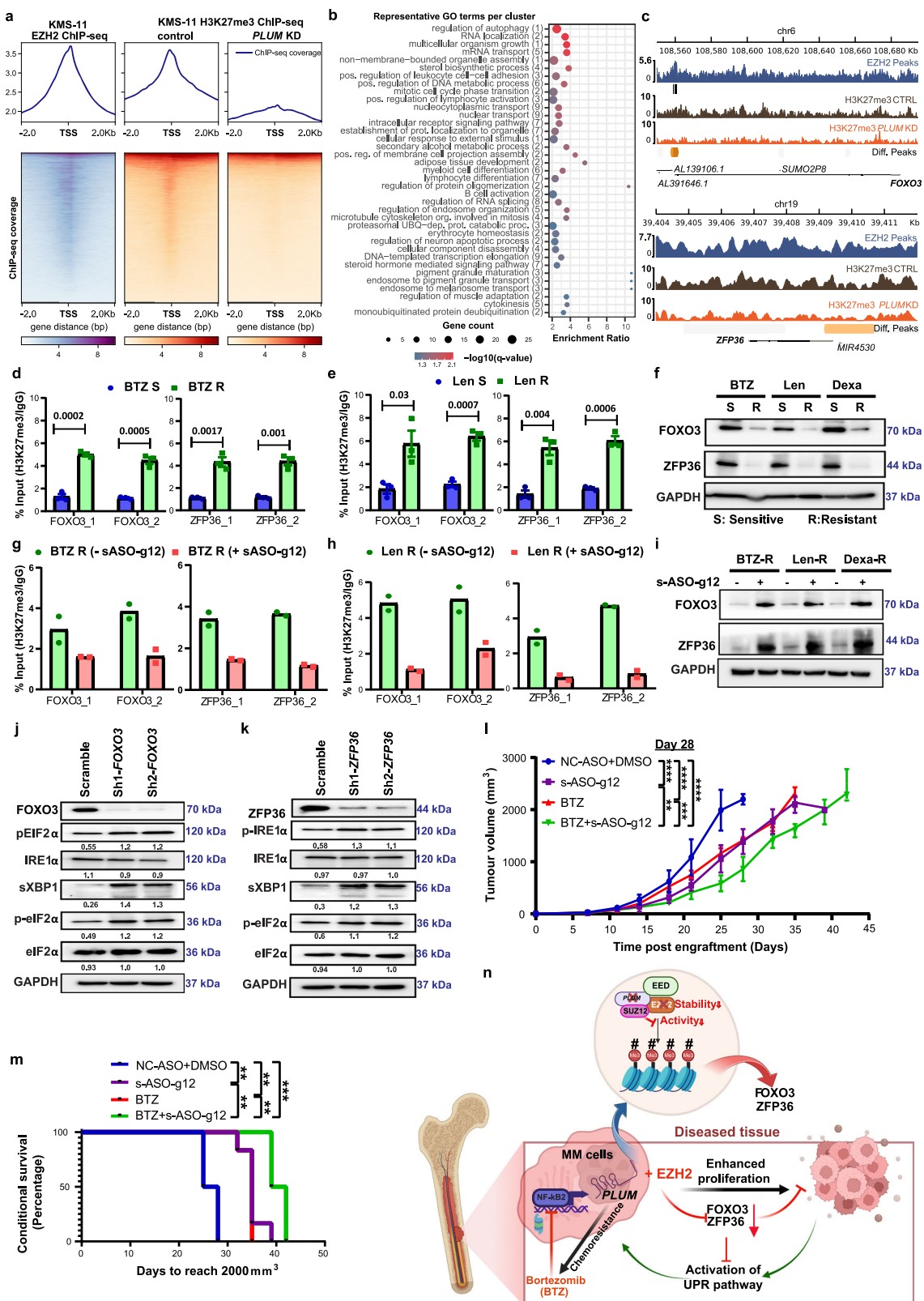

primer (5′-CGTTAATACGACTCACTATAGG target sequence-3′) and an aptamer sequence in the reverse primer (5′-CATGGCCCGGCCCGC-GACTATCTTACGCACTTGCATGATTCTGGTCGGTCCCATGGATCC target sequence (reverse complement)−3′[89]. The transcripts were amplified using Q5 polymerase (M0491L; NEB) with touch-up PCR program. Amplified PCR product was purified using ethanol purification method followed by in vitro transcription as follows: 6 μl 10×

transcription buffer, 8 μl 25 nM NTP mix (RO481; Life technologies) and 1 μl of T7 RNA polymerase (200 U/μl)(EP0113, Life technologies) were mixed with 1 μg of PCR product topped up the total volume of 60 μl. Incubation was performed at 37 °C for 4–6 h followed by purification using ethanol purification method and the RNA templates were proceeded for pull down. 25 μg of biotinylated RNA was bound to 50 μl of MyOne C1 beads (65001; Invitrogen) in RNA binding buffer (100 mM

**Fig. 7 | Hypermethylation of *FOXO3* and *ZFP36* via *PLUM*-mediated EZH2 activity promotes myeloma progression and chemoresistance. a** EZH2 and H3K27me3 binding at transcription start sites (TSS) along with loci displaying differential H3K27me3 marks in sh-scramble and sh2-*PLUM* KMS11 cells (*N* = 3 biological replicates). **b** Enriched biological processes GO terms associated with significantly upregulated genes displaying significant loss of H3K27me3 at their TSSs following *PLUM*-KD. **c** Genome browser visualization for the *FOXO3* and *ZFP36* loci. Tracks: EZH2 ChIP-seq (blue); H3K27me3 ChIP-seq: Control (brown) and *PLUM* KD (orange). **d, e** ChIP-qPCR validation (mean ± SEM) for *FOXO3* and *ZFP36* TSS region with H3K27me3 antibody in BTZ-S versus BTZ-R cells and Len-S versus Len-R cells, respectively (*N* = 3 biological replicates, two-sided unpaired student's *t*-test; *p*-values mentioned in the plot). **f** Levels of FOXO3 and ZFP36 protein in BTZ-R, Len-R, Dexa-R and their parental sensitive cell lines (*N* = 3 biological replicates). **g, h** ChIP qPCR validation for *FOXO3* and *ZFP36* TSS region with H3K27me3 antibody in BTZ resistant and Len resistant cells, respectively, treated with and without s-ASO-g12 for 24 h (*N* = 2 biological replicates). **i** Levels of FOXO3 and ZFP36 protein in BTZ-R, Len-R, Dexa-R cell lines treated with and without steric s-ASO-g12 for 24 h (*N* = 2 biological replicates). **j, k** Levels of UPR master regulators in sh-scramble versus sh-

*FOXO3* and sh-*ZFP36* cells, respectively. Average quantified values marked below each blot (*N* = 3 biological replicates and quantification done using GelQuant.NET). **l** Tumour growth plot depicting average tumour volume (mean ± SEM) from mice engrafted with *PLUM* OE KMS11 cells under different treatment groups (DMSO + NC-ASO; s-ASO-g12; BTZ; s-ASO-g12 + BTZ) (*N* = 6 mice per condition; two-sided multiple *t*-test; *p*-values−NC-ASO + DMSO versus s-ASO-g12: 0.0009, NC-ASO + DMSO versus BTZ: <0.0001, NC-ASO + DMSO versus BTZ + sASO-g12: <0.0001, s-ASO-g12 versus BTZ + s-ASO-g12: 0.005, BTZ versus BTZ + s-ASO-g12: 0.001). **m** Kaplan−Meier conditional survival plot of mice xenografts as mentioned in (**l**). *N* = 6 mice per condition; two-sided log-rank (Mantel−Cox) test; *p*-values−NC-ASO + DMSO versus s-ASO-g12: 0.0015, BTZ versus BTZ+s-ASO-g12: 0.0013, NC-ASO + DMSO versus BTZ: 0.0015, NC-ASO + DMSO versus BTZ+s-ASO-g12: 0.0009. **n** Graphical representation of the working model: *PLUM*-EZH2 interaction drives formation of PRC2 complex altering stability and activity of EZH2 to mediate repression of tumour suppressor genes (*FOXO3* and *ZFP36*), in turn inducing UPR pathway and confers chemoresistance in MM. Created in BioRender. Deka, K. (2025) https://BioRender.com/x1ek3vo.

---

NaCl, 10 mM MgCl2, 50 mM HEPES, ph7.4, 0.5% Igegal CA-530) for 30 min at 4 °C on a rotator. Beads were washed 3 times with RNA wash buffer (250 mM NaCl, 10 mM MgCl2, 50 mM HEPES, ph7.4, 0.5% Igegal CA-530) before incubation with 500 µg of nuclear protein extract, cOmplete protease inhibitor, 40 units of RNase inhibitor (AM2696; Life technologies) and 20 µg yeast tRNA (M0491L, Life technologies) for 1 h at 4 °C on a rotator. After washing with wash buffer 3 times, bound proteins were eluted with 25 µl 2× laemmli buffer by heating at 95 °C for 5 min.

## Mass spectrometry (Label Free Quantification−LFQ) and data analysis

RNA pull-down samples were separated on a 4−12% NuPAGE Bis-Tris precast gel (Thermo Fisher Scientific) for 10 min at 170 V in 1× MOPS buffer. The gel was fixed using the Colloidal Blue Staining Kit (Thermo Fisher Scientific) and processed as a single fraction. For in-gel digestion, samples were destained in destaining buffer (25 mM ammonium bicarbonate; 50% ethanol), reduced in 10 mM DTT for 1 h at 56 °C followed by alkylation with 55 mM iodoacetamide (Sigma) for 45 min in the dark. Tryptic digest was performed in 50 mM ammonium bicarbonate buffer with 2 µg trypsin (Promega) at 37 °C overnight. Peptides were desalted on StageTips and analysed by nanoflow liquid chromatography on an EASY-nLC 1200 system coupled to a Q Exactive HF mass spectrometer (Thermo Fisher Scientific). Peptides were separated on a C18-reversed phase column (25 cm long, 75 µm inner diameter) packed in-house with ReproSil-Pur C18-AQ 1.9 µm resin (Dr Maisch). The column was mounted on an Easy Flex Nano Source and temperature controlled by a column oven (Sonation) at 40 °C. A 105-min gradient from 2 to 40% acetonitrile in 0.5% formic acid at a flow of 225 nl/min was used. Spray voltage was set to 2.2 kV. The Q Exactive HF was operated with a TOP20 MS/MS spectra acquisition method per MS full scan. MS scans were conducted with 60,000 at a maximum injection time of 20 ms and MS/MS scans with 15,000 resolutions at a maximum injection time of 50 ms. The raw files were processed with MaxQuant version 1.5.2.8[90] with preset standard settings for label-free quantitation using the MaxLFQ algorithm[91]. Carbamidomethylation was set as fixed modification, while methionine oxidation and protein N-acetylation were considered as variable modifications. Search results were filtered with a false discovery rate of 0.01. Known contaminants, proteins groups only identified by site, and reverse hits of the MaxQuant results were removed and only proteins with LFQ intensities in four replicates of at least one condition of each pair-wise comparison were kept. Missing LFQ intensities were randomly imputed from a normal distribution around the lowest 5% of LFQ intensities based on the means of three iterations. Proteins that were enriched with a fold change >4 and *p*-value < 0.01 on the full-length *PLUM*

compared to either ΔExon1 or ΔExon7 constructs was considered as candidate RBP(s).

## RNA immunoprecipitation (RIP)−qPCR

Cells were cross linked with 37% formaldehyde (to a final concentration of 1%) in medium and incubated for 15 min at RT. Cross-linking was neutralized by adding 2 M glycine (to a final concentration of 0.2 M) for 5 min at RT. Crosslinked cells were washed twice with TBS and proceeded for nuclear fraction isolation using nuclear isolation buffer (1.28 M sucrose; 40 mM Tris-HCl, pH 7.5; 20 mM MgCl2; 4% Triton ×-100) with frequent mixing for 20 min. Nuclei pellet was collected by centrifugation at 2,500 × *g* for 15 min, which was resuspended in freshly prepared RIP buffer (150 mM KCl; 25 mM Tris, pH 7.4; 5 mM EDTA; 0.5 mM DTT; 0.5% NP-40; 100 U/mL RNAase inhibitor SUPERase•In (added fresh each time); Protease inhibitors (added fresh each time). Resuspended nuclei were split into two fractions: control and target, followed by shearing 3−5 rounds of low power sonication using Bioruptor (Diagenode) for 3 cycles (30 s ON and 30 s OFF). The lysate was subsequently cleared by centrifugation at 14,000 rpm for 10 min and proceeded for RIP. 10 µg of Ab was added to 1 mg of supernatant and incubated for 6 h at 4 °C in a rotator. Subsequently, protein A/G beads (40 µl) was added and incubated for another 1 h at 4 °C. Beads were pelleted down at 2500 rpm for 30 s and washed 3 times with RIP buffer. After second wash, 10% of the beads was kept for SDS-PAGE analysis to confirm the IP and remaining beads were used to isolate the co-precipitated RNA using TRIzol reagent following manufacturer's protocol. DNase treated and purified RNA was then reverse transcribed, and qPCR was performed using target RNA specific primer pairs.

## Co-immunoprecipitation (Co-IP)

For Co-IP, nuclear fraction was extracted using the protocol mentioned in method 15. 500 µg of nuclear extract was then incubated with 5 µg of primary Ab of interest or its corresponding IgG control diluted in IP lysis buffer at 1 mg/ml concentration in a 1.5 ml tube. Incubation was performed overnight at 4 °C in a rotator, followed by addition of 50 µl of homogenized Protein-G Dynabeads (10004D; Life Technologies) per each sample. The mixture was incubated in a rotator for 30 min at RT. Washing was done twice with wash buffer I (50 mM Tris, pH 7.5; 500 mM NaCl; 1 mM EDTA; 0.5% NP-40; 0.5% Triton ×-100, 0.05% Tween 20) and twice with wash buffer II (50 mM Tris, pH 7.5; 150 mM NaCl; 1 mM EDTA; 0.5% NP-40; 0.5% Triton ×-100, 0.05% Tween 20), 5 min each at 4 °C. After the last wash, beads were transferred to a fresh tube and immunoprecipitated protein was eluted using 35 µl of 2× laemmli buffer by heating at 99 °C for 10 min.

### Annexin V/PI apoptosis assay

Cells were harvested and washed twice with cold PBS. Pellet was resuspended in 1× binding buffer containing Annexin V-Alexa Fluor™ 350 (A23202; Invitrogen) and Propidium Iodide (PI, P4864; Merck) following the manufacturer's instructions. The samples were analysed by flow cytometry (BD LSRFortessa™ ×-20) to gate for apoptotic and live cells. Statistical analysis was done using the FlowJo v10.2 software.

### MTT assay for IC$_{50}$ survival curve

Cell growth inhibition and cell viability was determined by MTT assay. In brief, 50,000 cells were seeded/well in a 96-well plate one day prior to treatment with serial dilution of drug concentration (dug volume 10 μl/well). Following drug treatment, 10 μl of 0.5 mg/ml MTT reagent was added to each well after 24 h of treatment with BTZ and 4 days of treatment with lenalidomide and incubated for 2 h in CO$_2$ incubator. Then, 150 μl of DMSO was added into each well, mixed well and incubated further in a 37 °C incubator until all the water-insoluble formazan crystals were fully dissolved. Optical density (OD) was measured with a plate reader at a wavelength of 570 nm with a reference filter of 650 nm. The cytotoxicity was expressed as relative viability with the percentage of cell survival in the negative control (without drug treatment) taken as 100%. Relative viability = [( experimental absorbance−background absorbance)/(absorbance of untreated control−background absorbance)] × 100%. The half maximal inhibitory concentration (IC$_{50}$) values of each drug were calculated using the survival curves, which plotted fractional cell viability against logarithm of drug dose, and IC$_{50}$ values were calculated by Prisms software (GraphPad Software).

### RNA and EZH2 structure prediction for in-silico docking

For modelling the full-length *PLUM* and exon7 of *PLUM*, several tools were evaluated: 3dRNA[92], RosettaFOLD2NA[93], RNAcomposer[94], DeepFoldRNA[95], trRosettaRNA[96] and DRfold[97]. Based on RNA scoring functions (rsRNASP and cgRNASP), the *PLUM*-FL structure modelled by 3dRNA and exon 7 structure modelled by RNAComposer were selected for docking studies.

PRC2/EZH2 complex structures were retrieved from the RCSB Protein Data Bank (PDB-5HYN). Due to lack of the complete structure of EZH2 from PDB id 5HYN (165 of the 746 residues were absent), we used both homology modelling and ab initio methods to predict the complete structure. SWISS-PROT was used for homology modelling[98], while ab initio tools included AlphaFold 2[99], RosettaFold2[100], Phyre2[101] and I-TASSER[102]. Among these, the remodelled structure of EZH2 obtained using I-TASSER was chosen for docking studies based on its model quality and completeness.

For in-silico docking, we used HDock[103] to dock *PLUM*-exon7 and *PLUM*-FL RNA with the modelled EZH2 protein structure. The docking scores were evaluated using HDock's knowledge-based iterative scoring functions ITScorePP and ITScorePR. The top 3 docked structures for each RNA (*PLUM*-FL and *PLUM*-exon7) were selected for further analysis and in vitro validation experiments.

### RNA-electrophoretic mobility shift assay (RNA-EMSA)

RNA-EMSA was performed using LightShift® Chemiluminescent RNA EMSA Kit (Thermo Scientific; 20158) as per manufacturer's instructions. Briefly, 4 μg of purified proteins of interest were incubated with 1 nM of biotinylated RNAs (synthesised using services of GenScript, Singapore) for 30 min at RT, then loaded on 8% native polyacrylamide gel in 0.5× TBE. RNA was then transferred to nitrocellulose membrane, followed by UV cross-linking at 120 mJ/cm$^2$ for 1 min. After proper blocking and washing of the cross-linked membranes, biotin-labelled RNA-protein complex was detected by chemiluminescence using Streptavidin–Horseradish Peroxidase Conjugate. Labelled random RNA sequence was used as negative control (NC) for binding specificity

and unlabelled RNA was used as competitive binder to confirm the binding specificity.

### RNA-seq library construction, data processing and analysis

The TRIzol-isolated total RNA samples were quantitated by Qubit RNA BR Assay (Q10210; Thermo Scientific) and assessed for their quality by Agilent bioanalyzer run using RNA 6000 Nano Kit to determine the RIN scores for the samples. The RNA samples were then treated with the RNase-free, Turbo DNase I (AM2238; Thermo Scientific), followed by poly (A) mRNA enrichment using NEBNext® Poly(A) mRNA Magnetic Isolation Module (E7490S; NEB) according to manufacturer's instructions. Subsequently, the RNA library was prepared using the NEBNext® Ultra™ II Directional RNA Library Prep Kit (E7760S; NEB) according to manufacturer's recommendation. All libraries underwent 8 cycles of PCR with the recommended conditions. NEBNext oligos from NEB were ligated to allow for multiplexing of samples. After library preparation, the samples were quantitated and run on the Bioanalyzer using DNA High Sensitivity DNA Analysis kit to determine library molarities. Libraries were pooled, which was then diluted to a final concentration of 5 nM. Pooled and diluted library was sent for next-generation RNA-seq by NovogeneAIT Genomics Singapore Pte Ltd. Library was loaded on a HiSeqX and sequencing was performed at an output of 110G per lane. Each sample was prepared in biological triplicates to ensure the reliability of RNA-seq data. Reads were mapped to GRCh38 using STAR (2.7.4a) and counted using featureCounts (v2.0.1) against the GENCODE v42 annotation prior to differential testing with DESeq2 (v1.34.0).

### Chromatin immunoprecipitation-Quantitative PCR (ChIP-qPCR) and ChIP-seq library preparation

ChIP-qPCR and ChIP-seq were performed as per protocol mentioned in refs. [104],[105]. In brief, a total of 15 × 10$^6$ were crosslinked with 1% formaldehyde for 12 min at RT. Crosslinking was stopped by incubating the reaction with 0.125 M glycine. Cells were spinned down and washed once with cold DPBS. Washed cell pellet was resuspended in 2 ml SDS buffer (100 mM NaCl, 50 mM Tris.Cl, pH8, 5 mM EDTA, 0.5% SDS, 1× protease inhibitor) and incubated for 10 min on ice. Subsequently, cells were centrifuged (1200 rpm, 6 min, 4 °C), and the nuclear pellet was resuspended in 1.5 ml ChIP buffer (2 parts of SDS lysis buffer, 1 part of Triton buffer [100 mM NaCl, 10 mM Tris.Cl, pH8, 5 mM EDTA, 5% Triton-×100] and 1× protease inhibitor) and subjected to 20 cycles of high power (30 s ON and 30 s OFF) sonication to obtain a fragment size of 200–500 bp, which was confirmed by running on 2% agarose gel. The mixture was centrifuged (12,000 × g, 1 min, 4 °C), and the supernatant was collected. Sheared chromatin was then precleared with 40 μl of BSA-blocked Protein G agarose beads (50% slurry) with continuous rocking at 4 °C for 2 h. For immunoprecipitation (IP), appropriate amount of primary antibody (8–10 μg) was added to the precleared chromatin topped up to 1 ml reaction with IP buffer. IP reaction was incubated at 4 °C overnight in a rotator. Following incubation, 60 μl of blocked protein G agarose beads were added to each IP reaction and incubated at 4 °C for 6 h. Beads were captured and washed 4 times (5 min each) in sequence with Mixed Micelle buffer (150 mM NaCl, 20 mM Tris.Cl, pH8, 5 mM EDTA, 5.2% w/v sucrose, 1% Triton-×100, 0.2% SDS), Buffer 500 (0.1% sodium deoxycholine, 1 mM EDTA, 50 mM HEPES, pH7.5, 500 mM NaCl, 1% Triton-X100), LiCl buffer (0.5% sodium deoxycholine, 1 mM EDTA, 250 mM LiCl, 0.5% NP-40, 10 mM Tris.Cl, pH8) and TE buffer (10 mM Tris.Cl, pH7.4, 1 mM EDTA). Immunoprecipitated complexes were eluted by adding 200 μl elution/ decrosslink buffer (10% SDS and 0.084 g NaHCO$_3$ to a final volume of 10 ml with H$_2$O) to the beads containing 20 μg RNase A and 10 μl of 5 M NaCl. The reaction was incubated at 65 °C with agitation for 1 h followed by overnight incubation without agitation. DNA was extracted using QIAquick PCR purification kit (Qiagen). Locus-specific chromatin immunoprecipitation-quantitative PCR (ChIP-qPCR) was performed

with SsoAdvanced Universal SYBR Green Supermix (1725272; BioRad) using primers listed in Supplementary Table 3.

For ChIP-seq library preparation, immunoprecipitated DNA was quantified using Qubit™ 1× dsDNA High Sensitivity (HS) and Broad Range (BR) Assay Kits (Q33231; Life Technologies). Subsequently, library was prepared using NEBNext® Ultra™ II DNA Library Prep Kit for Illumina® (E7645S; NEB and E7103S; NEB) according to manufacturer's recommendations. After library preparation, the samples were quantitated and run on the Bioanalyzer using DNA High Sensitivity DNA Analysis kit (5067-4626; Agilent) to determine library molarities. Libraries were pooled, which was then diluted to a final concentration of 5 nM. Pooled and diluted libraries were sent for next-generation RNA-seq by NovogeneAIT Genomics Singapore Pte Ltd. Library was loaded on a HiSeqX and sequencing was performed at an output of 110G per lane. Each sample was prepared in biological triplicates to ensure the reliability of ChIP-seq data.

### ChIP-seq data processing and analysis
All ChIP-seq data was processed through the ENCODE ChIP-seq pipeline (GRCh38; in tf or histone mode) (https://github.com/ENCODE-DCC/chip-seq-pipeline2) as mentioned in ref. 104. ChIPseqSpikeInFree normalization was applied to the *PLUM* KD H3K27me3 experiment to enable detection of global changes in histone modifications[106]. After normalization, ChIP-seq coverage profiles and genome tracks were generated using deeptools and pygenometracks respectively[107,108]. For differential analyses, counts around each TSS (2 kb upstream and 500 bp downstream) were obtained using featureCounts v2.0.1 and normalized using ChIPseqSpikeInFree (v1.2.4) scaling factors prior to DESeq2 (v1.34.0) differential testing[109,110]. H3K27me3 levels associated with each TSS were matched with differential gene expression on the basis of shared gene identifiers and visualized in R (ggplot2)[111].

### CoMMpass and CCLE dataset analysis
Indexing, subtyping and differential testing of CoMMpass dataset IA14a were performed as detailed previously in ref. 14. Pre-normalised CCLE expression data (SRP186687) was retrieved from LncExpDB and annotated using LncBook 2.0 prior to visualization in R (ggplot2).

### GO analysis
GO annotation was conducted using clusterProfiler[112]. GO term summarization was performed by semantic clustering (Jiang) using GOSemSim in combination with hierarchical clustering (hclust)[113]. GSEA was conducted against the ranked gene list (FDR <= 0.1) using the GSEA function and the HALLMARK gene sets retrieved from MSIGdb[114].

### Steric antisense oligonucleotide (ASO) designing and treatment
Designing of steric ASOs (mixmers) was done using both 2nd and 3rd generation mixed backbone ASO modifications as they do not activate RNase H activity but produce their biological effects by using steric hindrance for binding of bulky protein complexes. ASOs were either 2′-O-Methoxyethyl (2′MOE)/locked nucleic acid at adenosine residue (LNAa) or 2′MOE/locked nucleic acid residue at cytosine residue (LNAc) modified. The designed ASOs were synthesized using services of GenScript, Singapore. For in vitro treatment, Lipofectamine™ RNAi-MAX Transfection Reagent (13778075; Invitrogen) was used. For in vivo treatment, ASO was administered via intra-tumoral (i.t.) injection.

### Mouse xenograft experiment
6–10 weeks old Balb/c RAG −/− IL2Rγ −/− mice (both male and female) were used to perform in-vivo mouse xenograft experiment to assess the role of *PLUM* and s-ASO on tumour growth and BTZ drug response as per detailed protocol mentioned in ref. 105. Briefly, for BTZ drug response studies: $5 \times 10^6$ of vector control and *PLUM* overexpressed (OE) cells were resuspended in DPBS containing 50% Matrigel (BD) and injected subcutaneously into the flank of the mice (10 mice/group). Upon reaching the tumour size of ~150 mm³, mice from each group of 10 were randomly divided into two sub-groups of 5 mice each to be treated by vehicle (DMSO) and BTZ (1 mg/kg of body weight): Sub-group 1-VC with DMSO ($N = 5$); Sub-group 2−VC with BTZ ($N = 5$); Sub-group 3−*PLUM* OE with DMSO ($N = 5$); Sub-group 4−*PLUM* OE with BTZ ($N = 5$). BTZ was administered intraperitoneally up to 5 doses every 3–4 days twice weekly and tumor sizes were measured using an electronic digital caliper. Tumor volume (mm³) was calculated as $\frac{1}{2} \times L \times W^2$. For survival evaluation, mice were euthanized when the tumor size reached 2000 mm³ (humane end point).

### ASO and BTZ combination drug response studies
$5 \times 10^6$ of *PLUM* overexpressed (OE) cells were resuspended in DPBS containing 50% Matrigel (BD) and injected subcutaneously into the flank of the mice (24 mice). Upon reaching the tumour size of ~150 mm³, mice were randomly divided into four treatment groups: Treatment group 1−DMSO with NC-ASO ($N = 6$); Treatment group 2−BTZ (1 mg/kg) ($N = 6$); Treatment group 3−s-ASO-g12 (15 µg) ($N = 6$); Treatment-group 4−BTZ with s-ASO-g12 ($N = 6$). BTZ was administered intraperitoneally (I.P.) and s-ASO was administered intra-tumorally (i.t.) up to 5 doses every 3–4 days twice weekly. Tumor sizes were measured using an electronic digital caliper and tumor volume (mm³) was calculated as $\frac{1}{2} \times L \times W^2$. For survival evaluation, mice were euthanized when the tumor reached 2000 mm³ (humane end point). BTZ was injected intra-peritoneally (I.P.) at a concentration of 10 mg/kg of body weight diluted in PBS (total volume of 200 µl) and sASO was injected intra-tumorally (i.t) at a total concentration of 15 µg diluted in PBS (total volume of 100 µl).

All the animal studies were approved by Institutional Animal Care and Use Committee of Nanyang Technological University, Singapore (NTU-ARF; AUP: A21070). The mice were housed in temperatures of 21–25 °C, relative humidity (RH) of 55–60% and pressure (Pa) of 5–8 with a 12 h light-dark cycle. Carbon-dioxide inhalation for 6 min followed by cervical dislocation was used as the method of euthanasia. Animal welfare monitoring was routinely done by NTU- Institutional Animal Care and Use Committee (IACUC), which operates under NACLAR guidelines and conducts Post-Approval Monitoring (PAM) to ensure compliance with approved Animal Use Protocols (AUPs).

### Statistics and reproducibility
All experiments which are statistically validated are representative of at least 2–3 independent experiments. All MTT experiments were done in triplicate wells and normalized as indicated in figure legends. Experimental replicate numbers are designated as "*N*" in the figure legends of each figure. Data are presented as mean ± SEM. Unpaired student *t*-tests/two-way ANOVAs were performed to calculate the *p*-values unless otherwise indicated using Graph Pad Prism 5.0 (Graph-Pad software, La Jolla, CA, USA). For in vivo studies, tumor measurement, treatment and analysis were performed in a blinded manner. Tumour volume significance was determined by two-sided multiple *t*-test and survival analysis was computed by the Kaplan−Meier method with statistical significance being determined by two-sided log-rank (Mantel−Cox) test. Animals were randomized, with each group receiving mice with similar tumor size or similar body weight. As for in vitro studies, randomization and blinding of cell lines was not possible; all cell lines were treated identically without prior designation. All statistical tests were performed with the assumption of similar variance for all test groups. No inclusion/exclusion criteria were pre−decided in any of the experiments. Fold change, *p*-values and adjusted *p*-values (IHW) were calculated in analysis. *p*-values < 0.05 are considered statistically significant.

### Reporting summary
Further information on research design is available in the Nature Portfolio Reporting Summary linked to this article.

## Data availability

The raw mass spectrometry data generated in this study have been deposited to the ProteomeXchange Consortium via the PRIDE partner repository with the dataset identifier PXD054586. The analysed mass spectrometry data is available in supplementary Data 1. The raw ChIP-seq and RNA-seq data generated in this study have been deposited in the GEO (Gene Expression Omnibus) database under accession code GSE274152. The publicly available p52 KD RNA-seq data used in this study are available in the GEO database under accession code GSE230293[14]. The publicly available MMRF CoMMpass data used in this study can be accessed from the MMRF Researcher Gateway[115]. The remaining data are available within the Article, Supplementary Information or Source Data file. Source data are provided with the paper. Source data are provided with this paper.

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

## Acknowledgements

This research is supported by the Singapore Ministry of Health's National Medical Research Council under its Open Fund Individual Research Grant (NMRC Project Number: MOH-001710), Ministry of Education, Singapore, under its Academic Research Fund Tier 1 (Project Numbers: RT18/23 and RG33/20) and the National Research Foundation (NRF) Singapore, under its Singapore NRF Fellowship (Project Number: NRF-NRFF2018-04). D.A.A. is funded by the PhD scholarship from Nanyang Technological University. We appreciate the assistance from members of Y.L.'s lab who participated in this work and thank Aloysius Teo Kai Soon for assisting D.A.A. in the RACE assays. We also thank the NTU Protein Production Platform (www.proteins.sg) for the cloning, expression tests and purification of WT-EZH2 and mut4-EZH2 constructs. Any opinions, findings and conclusions or recommendations expressed in this material are those of the authors and do not reflect the views of the Ministry of Education, Singapore.

## Author contributions

K.D. and Y.L. conceptualized the study; K.D., D.A.A., N.S. and H.Y.C. performed the experiments; K.D., JM.C., D.A.A., A.B., H.Y.C. and D.K. conducted formal analyses; G.H.B.L. and S.M.T. provided technical support and advice; Y.L. provided resources; W.J.C. provided expertise for analysis of clinical data and resistant/sensitive MMCLs; K.D. and Y.L.

were responsible for manuscript writing with input from D.K. and W.J.C.; and Y.L. provided supervision.

## Competing interests

Yinghui Li and Kamalakshi Deka are co-inventors of a provisional patent application - 10202501192X (Title: Design of antisense oligonucleotides targeting EZH2 activity as a new treatment modality for drug resistant cancers) for the in-silico docking models of *PLUM*-EZH2 complex, design and functional validation of steric ASOs described in this paper. The other authors declare no competing interests.
