## [Transparent Peer Review file · Nature Communications]

Multiple myeloma associated long non-coding RNA PLUM confers chemoresistance by enhancing PRC2 mediated UPR pathway activation

Corresponding Author: Dr Yinghui Li

Version 0:

Reviewer comments:

Reviewer #1

(Remarks to the Author)

This study by Deka and colleagues identifies PLUM as a lncRNA that is regulated by alternative NF- κ B activation in multiple myeloma. The authors find that PLUM over expression induces resistance to commonly used drugs to treat myeloma. They go on to show by mass spec that the functional spliceoform of PLUM associates with EZH2, and that PLUM-EZH2 interactions stabilize EZH2 expression. Small molecules that disrupt binding of PLUM and EZH2 were developed by the authors and these drugs were found to be toxic in combination with proteasome inhibitors or immunomodulatory drugs.

Overall, I found the manuscript interesting, well-written, and visually appealing. In particular, the novel biology surrounding PLUM expression is intriguing. However, I remain unconvinced about the therapeutic relevance of targeting PLUM-EZH2 interactions, given the existence of effective EZH2 inhibitors in myeloma.

Major comments

1. While the authors effectively demonstrate the role of PLUM in stabilizing EZH2, a direct comparison to the clinically available EZH2 inhibitor, tazemetostat, is missing. This comparison would provide valuable insights into the potential advantages of targeting PLUM. It would be helpful to discuss the potential mechanisms by which PLUM-targeting therapies might overcome resistance mechanisms associated with EZH2 inhibitors.
2. The current RNA-seq analysis, using a p-value of 0.1, seems overly permissive. A more stringent FDR cutoff of 0.01 would likely yield a more focused and interpretable gene list. This would allow for a more robust pathway analysis and identification of key downstream targets of PLUM.

Minor comments

1. The authors should provide a brief explanation for prioritizing PLUM over lnc-NTMT1-2, which was the most significantly changed lncRNA following p52 KD.
2. The reference cited on line 106 appears to be irrelevant to the context of myeloma and PLUM. Please verify and correct the reference. This same line/reference is being used to support the identification of patients from the MMRF CoMMpass cohort "displaying hyperactivation of NF- κ B". This needs to be better detailed.
3. The criteria used to select responsive and non-responsive patients from the CoMMpass cohort should be clearly defined. This information is essential for interpreting the clinical relevance of the findings.
4. Related to Figure 2, it would be valuable to assess the impact of PLUM KD on cell viability in MM cell lines without NF- κ B mutations or known low NF- κ B activity. This would help to determine the extent to which PLUM's effects are dependent on NF- κ B signaling.
5. The viability curves in Fig 2 c-f demonstrate that PLUM oe can to some level rescue toxicity from PIs and IMiDs. Do you think that this effect is specific to these drugs or is this just a general pro-survival effect of PLUM? What would happen if you used a generally cytotoxic drug (etoposide?) with PLUM oe?

6. In figure 4a, why is EZH2 expression not reduced following PLUM KD?

Reviewer #2

(Remarks to the Author)

The article by Deka et al "Multiple myeloma associated lncRNA PLUM confers chemoresistance by enhancing PRC2 mediated UPR pathway activation" to be considered for publication in Nature Communication concerns the epigenetic regulation via lncRNA and interaction to the PRC2 complex. Findings of new targets for precision medicine including also within the epigenome is indeed imperative as new treatment options to reduce side effects and severe pain for these patients. Deka et al presents a robust paper with a good number of relevant experiments and validation. However, the framing of the paper claiming PLUM to be a novel target for MM resistance seems largely overexaggerated. A similar response concerning the alterations in histone methylation and gene expression will likely be achieved by the use of an EZH2 inhibitor (EZH2i). Is not the resistance induced by EZH2 mainly and PLUM is just a passenger? On this note for clinical relevance, several EZH2 inhibitors have already been tried both in a pre-clinical setting and in clinical trials for multiple myeloma with little success, not because of toxicity but rather lack of an expected response. Therefore, the frame and focus of lncRNAs becoming clinically relevant in treatment regimens should be altered. EZH2 inhibition has already been shown in combinatorial regimes to sensitize myeloma cells to e.g. BTZ treatment and authors therefore need to convince the reader how inhibition of PLUM would promote sensitivity to e.g. BTZ treatment over EZH2i. As such the conclusions lack, at least in part, clinical relevance and might be better suited to be a functional paper rather than framing it as a pre-clinical evaluation of a potential drug target for MM.

General comment:

The results should be presented in the frame of novel partners interacting with the PRC2 complex and put in this context with proper background and references to already published papers in this area. At present the clinical relevance seems overexaggerated and should be tuned down. A good number of relevant experiments and validation are presented in the paper. However, many of the experiments are performed with only one cell line or a limited number of replicates. A suggestion would be that the authors include a few more MM cell lines to confirm that this is not only a cell specific response or an artifact of 1 cell line. Also, many controls and uncropped WB are missing including loading controls and total H3 and GAPDH, as well as molecular weight indication. There is also limited statistical information in the figures and figure legends and no analysis for data normality seems to have been done.

Major concerns of statements in text and Figures:

Line 42: The authors do not specify any lncRNAs that are currently being evaluated as drug targets in a clinical setting. The papers cited here refer to pre-clinical evaluations and does not at all reflect the current clinical evaluation of lncRNAs. This part of the paper should either be referenced correctly to original papers or omitted.

Line 60: There are recent papers describing the recruiting function of lncRNAs for PRC2 in MM in particular. These results should be cited these to get a proper overview of the field of interacting partners to the PRC2 complex.

Line 65: The authors suggests that the lncRNA PLUM might drive chemoresistance but also EZH2. What would be the advantage of targeting the lncRNA over targeting EZH2 directly? If anything the lncRNAs in general have many targets that are not acting on EZH2 or the UPR pathway genes.

Line 106: The text refers wrongly to Figure 1a.

Line 113: In what way can the authors conclude that overexpression of PLUM compared to other cancers has any impact on MM development? The authors have not demonstrated any functional association between PLUM and MM pathogenesis, but rather characterized the expression in different molecular groups and the differential expression in other cancers, none of which suggests pathogenesis or tumor development. This statement should be underbuilt by data or modified.

Line 132: The authors claim that PLUM is novel to NF-kb/p52 activation, but when inhibiting PLUM the results show indication of increased proliferation in both NF-kb+ and NFkb- cell lines. There is also induction of apoptosis in 3 out of the 4 cell lines, so in what way can the authors argue and claim this to be NF-kb activation-specific or are the results independent of NF-kb activation? Did the authors try to use ASOs on the NF-kb- cell lines and if so - what was the results? (see comment Fig 2 Suppl).

Line 143: In the method section the authors state that 10 mice/group was used, however, in the figures n = 5 only, meaning that half of the population was excluded. In the paper there is no reasoning for exclusion and in addition, 50% loss is quite extensive in an in vivo experiment. The authors should increase the number of mice in their pool of the experiment or validate their experiments in another model.

Line 166 (fig 3h i): Like the other WBs the authors did not supply the uncropped versions of the WBs making assessments difficult. Also, EZH2 and SUZ12 have similar MW and as EED have 3 isoforms and similar MW as GAPDH, can the authors confirm if the WB in figure 3h was done on the same membrane? If not additional a loading control e.g. GAPDH is required.

Line 178: Have the authors evaluated if the same results can be achieved by using an EZH2 inhibitor? The argument made prior that this is potential novel targeting needs to be certified in a way that it outperforms EZH2 inhibition. The use of EZH2i has already been evaluated in MM clinical trials with very limited successful outcomes.

Fig 1a: The article includes functional studies showing that PLUM interacts with the catalytic subunit EZH2 of the PRC2 to regulate histone methyltransferase activity and thus modulate the expression of tumour suppressor genes to activate the unfolded protein response (UPR). Why did the authors specifically select PLUM above other lncRNAs? In figure 1a it also looks like PLUM has a negative log₂FC suggesting that it is downregulated as compared to control? Figure 1a is overall very difficult to understand and should be clarified.

Figure 1c: These differences referred to in the text (line 110) are not significant. This should be omitted or clarified. What does the results really show?

Fig 1a Suppl.: The knockdown of the p52 looks very minor and not at all convincing on the western blot. In general evaluation of western blots are very difficult. The authors should provide uncropped images of the WB and add molecular weight indication.

Fig 1c Suppl: No statistics is presented. This should be added for proper evaluation.

Fig 2 Suppl: Did the authors use ASOs on the NF- κ B- cell lines and if so what was the results?

Fig 3: Exon deletion mutations: It is very difficult to see those lines to make a conclusion. The authors state that mutant deficient in exon 7 lose their oncogenic effects and BTZ resistance but binds EZH2, while the full length and exon 1 deficient does not. In what way does this drive resistance?

Fig 3 f and g: The authors decided only to use one cell line here and only 2 replicates yet have statistics on the data! This should be clarified and corrected.

Fig 4: Was the co-immunoprecipitation (co-IP) only done once or in triplicates?

Fig 4g: This is only shown in one, very specific MM cell line which limits the impact of the functional experiment and validation. This is also true for many of the other experiments. The authors should include a few more MM cell lines to confirm that this is not only an artifact of 1 cell line.

Fig 4i: Results should be shown whether this causes reduction of H3K27me3 either globally or site specific.

Fig 5g: This figure does not show any reduction of H3K27me3 so how does the authors know that this treatment reduced EZH2 activation?

Fig 5h and i: The changes here look very limited, what statistical test was used here? The results should also be validated in other MM cell lines.

Line 260 and Fig 5: Does PLUM or EZH2 contribute to the resistance here? With lacking information that PLUMi outperforms EZH2i in a clinical setting this statement becomes rather overexaggerated.

Fig 6a: 9400 genes seem utterly a large number as this would almost affect 50% of the genome. Perhaps a cut/off is needed?

Fig 6c and d: This WB has to be explained better, what are the differences here?

Reviewer #3

(Remarks to the Author)

Reviewer #4

(Remarks to the Author)

I restrict my comments to the in-silico modelling work in this manuscript.

1. To produce molecular models for the PLUMB-EZH2 RNA-protein complex, the authors first generate models for the two independent components using a variety of structure prediction tools, filter the results for plausibility on the basis of scoring metrics, then use docking (HDock) to generate alternative models for the complexes. The issue with this approach is that a critical section of EZH2 (residues 480-515), is observed to be disordered in the available crystal structure (5HYN). This includes the highly basic (and so highly positively charged) RKKKR patch (residues 490-994) that unsurprisingly is highly

implicated in RNA binding. The fact that the region is observed (in the context of the protein component alone) to be dynamic and disordered argues that it is interaction with the RNA component that is required to bring about its structural organisation. There is therefore a chicken-and-egg problem here: the protein-RNA docking method used here requires a complete structural model for each component, but an accurate model for the complete protein component requires a prediction of the RNA-protein interaction. With the recent release of AlphaFold3, the authors have an opportunity to evaluate the prediction of the complete complex ab initio, overcoming this issue, and this should be done.

2. The in-silico structure prediction work falls into two sections. In the first it is assumed (on the basis of significant experimental data) that the key interaction is between EZH2 and exon7=PLUMB, and so molecular models for this complex are generated (with the issues described above) in order to predict key RNA-interacting residues in EZH2. In the second section the assumption that exon7 is involved is put aside, and models for the complex of full-length PLUM with EZH2 are generated, in order to predict key protein-interacting regions in PLUMB. What is missing though is any presentation or discussion of the correlation between these two pieces of work. The authors must demonstrate that the structure predictions for FL-PLUMB-EZH2 recapitulate (in the exon-7 region) the structure prediction for exon-7 PLUMB-EZH2; so that e.g., prediction of key RNA-interacting regions of the protein are the same. Of course, if they do then one can question the value of the “exon 7-only” work; while if they disagree, this is a significant issue that must be addressed.

3. In any case I find the “exon-7 only” modelling section weaker, in that it does not drive any subsequent experimental work or provide significant additional insights. The prediction that the RKKKR patch is involved in RNA recognition is hardly surprising. What could have been more interesting was the prediction of two other RNA-recognising regions (562-575 and 586-602), but I do not see this discussed or followed up in any way. As an aside, the authors should clarify their wording on lines 199-200: the statement “the C-terminal disordered region of EZH2, ranging from aa residue 488 to 602 (Figure 4e)” could be misleading, as the crystal structure only shows residues 479-515 as disordered (depending slightly on which protein chain is being discussed). This means that these two extra patches correspond to well-structured regions of the protein, and so, ultimately, might be more attractive targets for the future development of therapeutic modalities.

4. The “FL-PLUMB” modelling section is much stronger, in that it results in three alternative candidate models for the protein-RNA complex being generated that then compete for validity through the subsequent ASO studies. The authors assert that the results of these experiments align most closely with what would be expected if their “model 3” is correct (lines 240-242) however I cannot see the full evidence for this – Figures 5a-c are not insightful. To support their contention, the authors should show - probably in one-dimensional form – what regions of PLUMB are predicted to be interacting directly with the protein in each of the three models put forward, and what regions of PLUMB each of the ASOs target.

Reviewer #5

(Remarks to the Author)

The manuscript by Deka et al. describes a new mechanism of chemoresistance in multiple myeloma (MM) involving a lncRNA named PLUM. PLUM regulates PRC2, leading to activation of the UPR and chemoresistance in MM. The manuscript presents a complete mechanistic story backed up by compelling evidence linking RNA, cell, and cancer biology, with well-chosen robust in vitro systems and metrics and substantiation of the findings in appropriate in vivo settings. However, it needs more discussion of previous research on the IRE1-XBP1 pathway in MM pathogenesis. Overall, the manuscript presents a significant advancement in understanding MM chemoresistance mechanisms and is suitable for publication in Nature Communications after addressing the comments below.

Main comment:

The IRE1-XBP1 pathway has been extensively studied for more than a decade as a contributor/driver of MM pathogenesis, and several studies show that blocking it produces desirable therapeutic outcomes in animal models (see, for example, PMID 17418411, 21081713, 24812669, 38748820, 31371506). Moreover, a recent comprehensive review of the UPR in disease highlights its many roles in cancer, including MM (PMID 39501044). These works should be cited and discussed accordingly.

Minor comments and suggestions for improvements:

1. The figures present an enormous amount of data to the reader. For that reason, the panels have been minimized considerably, which sometimes makes it challenging to read them and fully appreciate the data. Consider enlarging the panels and fonts.
2. The heat map in Fig. 1c is difficult to read; consider enlarging or presenting a crop with the most significant finding and place the complete heat map in the supplement.
3. The micrographs in Fig. 1g and 3f are hard to interpret as provided. Consider enlarging the images and providing close-ups (crops) in insets, showing representative data.
4. In Fig. 3e, what are the identities of the two data points depicted in blue that also show significant upregulation?
5. Fig. 3j/k suggests PLUM regulates EZH2 stability. Could the authors discuss or speculate how PLUM may protect EZH2 from degradation?
6. Can the unstructured region 32-42 of EZH2 be depicted in Fig. 4 to better situate the reader spatially on the structural model? The data in Fig. 4h suggest this region is essential.
7. Consider softening the claim that s-ASO-g4 has an effect; the effect is modest or non-existent (Fig. 5).
8. In Fig. 6h, did the authors inadvertently reverse the blot for P-eIF2a? These data are not consistent with the rest of the UPR markers.
9. In Fig. 6c,d, PLUM KD reduces P-eIF2a while OE enhances it. In Fig. 7j,k, the KD of FOXO3 and ZPF36 also enhance P-eIF2a, which is in line with the authors' interpretation that FOXO3 and ZPF36 suppress UPR activity (not just IRE1).

Accordingly, these findings and the possible roles of the other two UPR sensors, PERK and ATF6, should be discussed more thoroughly.

10. Are the blots shown for ATF6 for the full-length protein? Commercial ATF6 antibodies are unreliable, and if the authors wish to show potential ATF6 involvement, it would be best to look at downstream transcriptional targets (e.g., BiP, HERPUD1, GRP94, PDIA4).

11. The manuscript would benefit from quantification of the immunoblots, especially those in Fig.6 and 7, as they provide the mechanistic links between PLUM/PRC2 and the regulation of the UPR, potentially via FOXO3 and ZFP36.

Reviewer #6

(Remarks to the Author)

Version 1:

Reviewer comments:

Reviewer #1

(Remarks to the Author)

The authors have satisfactorily address all of my concerns.

Reviewer #2

(Remarks to the Author)

General major comments:

Could the authors clarify, for example in the abstract (lines 32–33: “Hence, targeting PLUM–EZH2 interactions may represent a clinically potent strategy for the treatment of relapsed, refractory MM”) and in lines 104–106 (“Our data further suggest that PLUM may be a potential therapeutic target for high-risk MM patients displaying enhanced EZH2 levels and resistance to current treatments”), how targeting PLUM–EZH2 interactions would yield a distinct clinical advantage compared to directly inhibiting EZH2 alone? Prior studies, including some cited by the authors, have demonstrated that EZH2 inhibition can sensitize multiple myeloma (MM) cells to proteasome inhibitors, primarily through reductions in EZH2 activity, loss of H3K27me3, and de-repression of tumor suppressors. If PLUM’s therapeutic relevance hinges on modulating these same downstream effects, it remains unclear what additional benefit PLUM targeting would offer. Furthermore, since EZH2 inhibitors are already FDA-approved and generally well-tolerated, the potential therapeutic gain from targeting a lncRNA like PLUM—which may have pleiotropic roles—must be carefully justified. Could the authors elaborate on whether PLUM targeting is expected to achieve mechanistically distinct or superior outcomes compared to existing EZH2 inhibitors?

Major comments:

1. Inconsistencies and Concerns Regarding Western Blot (WB) Source Data and Replicates

Several issues were observed in the Western blot data that should be clarified:

In Supplementary Figure 1a, the provided source data shows contrasting image qualities compared to the published figure. Notably, GAPDH in replicate 1 of LP-1 appears different between the figure and the source data. Additionally, replicate 3 of KMS11 shows an inverted isoform pattern (less isoform 1, more isoform 2), inconsistent with the other replicates. This dataset should be repeated for clarity and consistency.

The GAPDH blots for replicate 1–2 in KMS11 and those in LP-1 (replicates 2–3) differ significantly in appearance or cropping compared to their respective KD blots. Can the authors confirm that these are the correct corresponding membranes?

In several instances (e.g., Figures 3H, 3I, 3K, 3L, 4H, Supplementary Figures 4E, 4H, 5A, 5B), GAPDH blots appear inconsistent or missing altogether. For example:

Figure 3H: Replicate 1 suggests EZH2 and EED were run on the same membrane, but SUZ12 and GAPDH appear on a different one.

Figure 3I: Replicate 1 shows a GAPDH with a visible ladder not present in the other protein blots.

General comment: Total H3 should not be considered a loading control but rather to ensure that the effects seen are not due to a degradation of total histone H3.

Figure 3L: Replicates show varying results for H3K27me3 and pEZH2, but loading controls are inconsistent or unclear.

Figure 4H: Input and pull-down appear to be from different membranes, yet a single GAPDH is shown.

Supplementary Figure 5B appears to be missing GAPDH controls for sh-RNA SUZ12 replicates, and the number of replicates for EED (n=3) differs from EZH2 and SUZ12 (n=2).

Clarification is needed on whether membranes were stripped/reprobed or if these are distinct blots. If stripped, it should be noted; if distinct, each membrane should have its own corresponding loading control.

For Supplementary Figures 1F, 4H, and possibly others, the uncropped WB source data is missing and should be included.

2. Questions About Normalization and Replication in RT-qPCR and Functional Assays

In Supplementary Figure 1G, the normalization across cytoplasmic and nuclear fractions differs between cell lines (KMS11 and LP-1), despite supposedly identical reference transcripts (7SK and cyto-C). However, MM.1S shows the same values for both compartments. Please clarify.

The figure legends claim biological duplicates were used, but based on the source data (Supplementary Figures 1G, 1C, 2A), it appears only a single technical replicate was used for many assays. Can the authors confirm whether qPCR assays were performed in technical triplicates for each gene and sample?

Supplementary Figure 6C is unclear—were these results from technical triplicates?

For Figure 7C, are data presented from biological duplicates with only one technical replicate, or from full technical triplicates?

3. Statistical and Data Presentation Concerns

The authors refer to statistically significant upregulation in Figure 1D (lines 133–135), though this is not supported by the visual or statistical output. Similarly, Supplementary Figure 7C lacks statistical testing, despite claims of significance. Apoptosis assays in Figure 5I appear to be based on only two biological replicates, which limits statistical confidence. Can the authors justify the conclusions based on this sample size?

General concern: Is statistical testing based on $n=2$ reliable across multiple figures (e.g., Figures 3H, 3I, 5I, 6C–D)? Please explain the rationale for using such low replicate numbers and whether any statistical adjustments were made.

The issue with only two replicates and statistical assessments:

No estimate of variability: With only two replicates, you can't accurately estimate variance (standard deviation), which is essential for most statistical tests like t-tests or ANOVAs.

No degrees of freedom: In a t-test, degrees of freedom ($n_1 + n_2 - 2$) become very small (e.g., $df = 2$), leading to: Wide confidence intervals. Low statistical power. High risk of both Type I (false positive) and Type II (false negative) errors.

Outlier sensitivity: One outlier can completely skew your result because there's no buffer from additional data.

4. Experimental Design and Data Interpretation Issues

Figure 1E: The differences between MM and other cancers are difficult to distinguish. Consider improving contrast or data representation.

Supplementary Figure 2B: The source data lack column headers, making interpretation of the dataset difficult.

Supplementary Figure 3B and Figures 2C–F: There are inconsistencies in how the data are processed—for example, OnM is normalized to 100 in some cases (KMS11) but not others (LP-1). Please clarify the normalization method and rationale.

The logic of Figure 5E vs Supplementary Figure 6A is unclear. One shows ASO treatment vs IgG, while the other should be treated vs untreated. Are these from different experiments?

Figure 5G: H3K27me3 is run on two membranes, but histone H3 loading control appears on only one. Why were the blots not run on the same membrane?

5. Interpretation of Functional Outcomes and Therapeutic Relevance

The claim that PLUMi reduces toxicity compared to other drugs is questionable. The authors report >8000 DEGs following shRNA-mediated PLUM knockdown, which could indicate extensive off-target effects. How do the authors reconcile this with the suggestion of reduced toxicity?

It is also unclear why PLUM overexpression affects proliferation but only induces treatment resistance in MM cell lines with NF- κ B mutations. Can the authors provide mechanistic insights or supporting evidence?

6. Animal Study Clarifications

The manuscript states that 20 mice were used (10 vector controls, 10 treated with DMSO or BTZ), implying 5 mice per group. However, the source data lists 6 mice in the IncOE + DMSO group and 4 in the IncOE + BTZ group. Please clarify this discrepancy and update the source data as needed.

Reviewer #3

(Remarks to the Author)

Reviewer #4

(Remarks to the Author)

The authors have addressed my questions, comments and suggestions on their original manuscript in a comprehensive and positive manner in their rebuttal letter; though it's slightly disappointing that the proportion of this that has actually been incorporated into the revised manuscript and supplementary material seems to be the bare minimum. I have no new issues to raise.

Reviewer #5

(Remarks to the Author)

I want to thank the Authors for addressing each one of the points I raised in my initial review of the work. All comments have been addressed satisfactorily.

However, to improve the manuscript, I would like to suggest the authors consider the following:

1. Content: Please consider discussing the P-eIF2a results presented in Fig. 6 in the manuscript. The lowered P-eIF2a levels in resistant cells are unexpected given the high levels of IRE1 activity (measured by XBP1 splicing). The apparent discrepancy is explained in the rebuttal, but it would benefit the reader to include it in the manuscript. I also encourage the authors to consider adding their qPCR data on ATF6 targets (in response to one of my comments) as a supplementary figure, as it strengthens their conclusions.

2. Writing: Line 400, consider the following edit: "...chemoresistance via the activation of the UPR pathway, in addition to several targeted therapy..."

3. Figures: Revise the Western Blots in Figs. 6c and 6d as they appear distorted (squished); maybe there is a problem with document format conversions.

Reviewer #6

(Remarks to the Author)

Version 2:

Reviewer comments:

Reviewer #2

(Remarks to the Author)

The authors have addressed most of my questions, comments and suggestions. I have no new issues to raise.

Reviewer #3

(Remarks to the Author)

Reviewer #5

(Remarks to the Author)

I thank the authors for addressing my suggestions and recommendations satisfactorily. I have no further comments.

Reviewer #6

(Remarks to the Author)

Revisions and Responses to Reviewers' comments

Reviewer #1, expertise in NF- κ B signalling, MM, omics and models (Remarks to the Author):

This study by Deka and colleagues identifies PLUM as a lncRNA that is regulated by alternative NF- κ B activation in multiple myeloma. The authors find that PLUM over expression induces resistance to commonly used drugs to treat myeloma. They go on to show by mass spec that the functional spliceoform of PLUM associates with EZH2, and that PLUM-EZH2 interactions stabilize EZH2 expression. Small molecules that disrupt binding of PLUM and EZH2 were developed by the authors and these drugs were found to be toxic in combination with proteasome inhibitors or immunomodulatory drugs.

Overall, I found the manuscript interesting, well-written, and visually appealing. In particular, the novel biology surrounding PLUM expression is intriguing. However, I remain unconvinced about the therapeutic relevance of targeting PLUM-EZH2 interactions, given the existence of effective EZH2 inhibitors in myeloma.

Major comments

Comment A1

While the authors effectively demonstrate the role of PLUM in stabilizing EZH2, a direct comparison to the clinically available EZH2 inhibitor, tazemetostat, is missing. This comparison would provide valuable insights into the potential advantages of targeting PLUM. It would be helpful to discuss the potential mechanisms by which PLUM-targeting therapies might overcome resistance mechanisms associated with EZH2 inhibitors.

Response A1

Thank you for the experimental suggestion and insightful comment. To address this, we have treated two MM cell lines (KMS11 and BTZ resistant RPMI8226) with EZH2 inhibitor (tazemetostat) and compared the effects of EZH2 inhibition with that of our steric-ASOs (ASO-NC, s-ASO-g4, s-ASO-g11 and s-ASO-g12) treatment on the stability and activity of EZH2 through immunoblotting for p-EZH2, EZH2, H3K27me3 and H3 protein levels (supplementary figure 6d). We also performed drug sensitivity assays in the presence of tazemetostat and compared its efficiency with our steric ASOs in abrogating resistance to BTZ and Len treatment in KMS11, BTZ resistant RPMI8226 and Len resistant KMS11 (Supplementary figure 6g and 6h). Here, we observed that the effect of steric ASO treatment is comparable to that of tazemetostat in terms of reducing EZH2 stability/activity as well as sensitizing the myeloma cells to BTZ and Len treatment.

A

(Supplementary figure 6d- KMS11 and BTZ-R 8226)

B

(Supplementary figure 6g-KMS11)

C

(Supplementary figure 6h-BTZ-R 8226 and Len-R KMS11)

Figure 1: A. Immunoblots showing protein expression levels of p-EZH2/EZH2, H3K27me3/H3 and GAPDH in steric ASOs (0.5 μM) and EZH2 inhibitor (Tazemetostat) (0.5μM) treated cell lines (KMS11 and BTZ-R RPMI8226). ASO-NC and DMSO are used as negative control for s-ASO and EZH2 inhibitor treatment respectively. GAPDH is used as loading control for all the samples and treatment was administered for 3days. (N=3 biological replicates). B. Drug sensitivity IC₅₀ survival curve for steric ASOs and EZH2i (Tazemetostat) treated KMS11 cells in response to BTZ treatment for 24 h and Len treatment for 4 days respectively (N=3, mean ± SEM is plotted, and statistical significance was determined by two-way ANOVA; **p<0.005, ***p<0.0005, ns: non-significant). C. Drug sensitivity

IC50 survival curve for steric ASOs and EZH2i (Tazemetostat) treated BTZ resistant R8226 cell line in response to BTZ treatment for 24 h and Len resistant KMS11 cell line in response to Len treatment for 4 days respectively (N=3 mean \pm SEM is plotted, and statistical significance was determined by two-way ANOVA; *p<0.05, ns: non-significant).

To address the second part of the comment, we have revised our discussion to include the potential use of *PLUM*-targeted therapies in overcoming the resistance mechanisms linked to EZH2 inhibitors in the following paragraph: “Our characterization of the myeloma-associated expression of *PLUM* and its role in enhancing EZH2 stability and activity towards chemoresistance suggest its promising potential and future development as a biomarker and/or therapeutic target for MM. Moreover, the similar effects of *PLUM*-EZH2 targeting s-ASOs with that of EZH2 inhibitor (tazemetostat) in the destabilization of EZH2 activity and abrogation of chemoresistance support the possible use of *PLUM* as an alternative target to overcome the resistance mechanisms associated with known EZH2 catalytic inhibitors in MM. Herein, given the unknown role of both *PLUM* and EZH2/PRC2 complex in other physiological processes, we explored an alternative strategy using mixmer ASOs to interrupt *PLUM*-EZH2 interaction and its downstream oncogenic functions efficiently, instead of using inhibitory or degradative ASOs against *PLUM* and EZH2. The specificity of our mixmer ASOs to *PLUM* further minimizes off-target effects, reduces toxicity, and enhances affinity, offering a promising therapeutic approach compared to EZH2 inhibitors. However, a competitive evaluation of these steric ASOs with available EZH2 inhibitors requires further studies.” (line 430 to 443 in the revised manuscript)

Comment A2

The current RNA-seq analysis, using a p-value of 0.1, seems overly permissive. A more stringent FDR cutoff of 0.01 would likely yield a more focused and interpretable gene list. This would allow for a more robust pathway analysis and identification of key downstream targets of PLUM.

Response A2

We thank the reviewer for highlighting that a more balanced statistical approach in our RNA-seq analysis could yield more robust results. In response to this concern, we have revised our analysis by implementing the Independent Hypothesis Weighting (IHW) method with an adjusted p-value cutoff of 0.05, aligning with standard RNA-seq analysis practices. This approach provides substantially more statistical power and better control of the false discovery rate (FDR) than the original Benjamini-Hochberg approach. These changes bolstered the results of downstream analyses such as the UPR Gene Set Enrichment Analysis which showed increased enrichment and significance (Figure 6a, 6b).

A Day 3 post transduction

B

C

Supplementary Figure 7

Figure 2: A. Heat map representing the differential gene expression (DEGs) for RNA sequencing 3 days post transduction in sh-scramble and sh-*PLUM* KD KMS11 cells. (adjusted p-value ≤ 0.05). **B.** Gene Set Enrichment Analysis (GSEA) for UPR signatures showing association with significantly downregulated genes detected during *PLUM* KD in KMS11 cells. NES: -1.68; FDR: 0.00353. **C.** Enriched biological processes among upregulated DEGs upon *PLUM* knockdown in NF- κ B+ KMS11 cell line. Enriched biological processes among downregulated DEGs upon *PLUM* knockdown in NF- κ B+ KMS11 cell line.

Minor comments:

Comment A3

The authors should provide a brief explanation for prioritizing PLUM over Lnc-NTMT1-2, which was the most significantly changed lncRNA following p52 KD.

Response A3

Thank you for the suggestion. Although *lnc-NTMT1-2* was the most significantly regulated lncRNA following p52 KD, *PLUM* expression was most highly enriched in MM cell lines compared to other 51 cancer types from the CCLE dataset (Figure 1e). In contrast, *lnc-NTMT1-2* expression was not selectively enriched in MM cell lines relative to other cancer types. As our study focused on potential p52-regulated lncRNAs linked to MM progression, we prioritized our investigation of *PLUM* functions over *lnc-NTMT1-2*. Additionally, there is no significant difference in *lnc-NTMT1-2* expression levels between VRd responsive and non-responsive MM patients.

As suggested, we have also rephrased our statement in main text as “*PLUM* is also selectively upregulated in MMCLs compared to other 51 cancer types and other p52 regulated lncRNAs like *NR2F2-ASI* and *MALT1-AS* found in our earlier analysis, suggesting its aberrant expression could be linked to MM disease (Figure 1e).” (line 135 to 137 in the revised manuscript)

Comment A4

The reference cited on line 106 appears to be irrelevant to the context of myeloma and PLUM. Please verify and correct the reference. This same line/reference is being used to support the identification of patients from the MMRF CoMMpass cohort “displaying hyperactivation of NF-κB”. This needs to be better detailed.

Response A4

Thank you for the insightful comment. The reference cited here is related to our previous publication where we have performed the subtyping of patient samples from CoMMpass dataset based on various high-risk carrying mutations in MM. The same dataset was used to check the expression profile of *PLUM* in different high-risk subtypes of MM. As suggested, we have rephrased our statement as: “Using MMRF CoMMpass dataset, *PLUM* is found to be overexpressed in patient samples harbouring high-risk, aggressive MM subtypes displaying hyperactivation of NF-κB pathway (subtyping data published in ref [14]) (Figure 1b). (line 124 to 126 in the revised manuscript)

Comment A5

The criteria used to select responsive and non-responsive patients from the CoMMpass cohort should be clearly defined. This information is essential for interpreting the clinical relevance of the findings.

Response A5

Responsive and non-responsive patients were defined on the basis of the CoMMpass metadata (Best Response). Patients showing Complete Response (CR), Stringent Complete Response (SCR) or Very Good Partial Response (VGPR) were grouped as Responsive. Patients with Stable Disease (SD) or Progressive Disease (PD) were grouped as non-responsive. Partial Response cases were excluded. We have clarified these criteria in the main text at lines 127-131.

Comment A6

Related to Figure 2, it would be valuable to assess the impact of PLUM KD on cell viability in MM cell lines without NF- κ B mutations or known low NF- κ B activity. This would help to determine the extent to which PLUM's effects are dependent on NF- κ B signaling.

Response A6

Thank you for your insightful comment and suggestion. Since the endogenous expression of *PLUM* is already found to be 5-10 fold lower in NF- κ B- cells (XG-7, H929 and MOLP8) compared to NF- κ B+ cells (Supplementary figure 1c), so instead of KD studies, we performed exogenous overexpression of *PLUM* in NF- κ B- cells (XG-7 and H929) which showed enhanced proliferation and enhanced cell viability relative to vector control cells (VC) upon treatment with both BTZ and Len (Supp figure 2c, 2d and figure 2e, 2f).

Figure 3: A. Fold change in expression level of *PLUM* in MMCLs (KMS11 and LP1) post Lenti V2. CRISPR mediated KD of p52 compared to empty vector transduced cells (N=3, mean fold change \pm

SEM is plotted, and statistical significance was determined using unpaired student's t test; *** $p < 0.0005$). **B.** Proliferation rate of NF- κ B- MMCLs (XG-7 and H929) overexpressed with FL *PLUM* relative to vector control till the duration of 4 days post recovery (N=3, mean \pm SEM is plotted, and statistical significance was determined by two-way ANOVA; *** $p < 0.0005$). **C.** Drug sensitivity IC50 survival curve for vector control (VC) and FL-*PLUM* OE NF- κ B- non-mutant MMCLs (XG-7 and H929) in response to Bortezomib (BTZ) treatment for 24h (N=3, mean viability percentage \pm SEM is plotted, and statistical significance was determined by two-way ANOVA; * $p < 0.05$, ** $p < 0.005$, *** $p < 0.0005$). **D.** Drug sensitivity IC50 survival curve for vector control (VC) and FL-*PLUM* OE NF- κ B- non-mutant MMCLs (XG-7 and H929) in response to Lenalidomide (Len) treatment for 4 days (N=3, mean viability percentage \pm SEM is plotted, and statistical significance was determined by two-way ANOVA; * $p < 0.05$, ** $p < 0.005$, *** $p < 0.0005$).

Additionally, as suggested, we have performed another experiment: degradative ASO treatment in NF- κ B- cells (XG7, H929) followed by proliferation assay, drug sensitivity assay and annexin V staining. Our data showed no significant differences between ASO-NC and ASO-gIV, ASO-gV treated NF- κ B- MMCLs, suggesting the dependency of *PLUM* expression on the presence of activated NF- κ B signalling.

(Supplementary figure 3e, 3f)

A

B

(Supplementary figure 2i)

C

D

(Supplementary figure 2j)

Figure 4: A, B: Drug sensitivity IC₅₀ survival curve for NC-ASO and *PLUM* degradative ASO (ASO-gIV and ASO-gV) treated NF- κ B- MMCLs (XG-7 and H929) in response to BTZ treatment for 24h and Len treatment for 4 days (N=3, mean viability percentage \pm SEM is plotted, and statistical significance was determined by two-way ANOVA; ns: non-significant). **C.** Proliferation rate of NF- κ B- MMCLs (XG-7 and H929) treated with negative control (NC) ASO and *PLUM* degradative ASOs (ASO-gIV and ASO-gV) for 36 h (N=3, OD mean \pm SEM is plotted, and statistical significance was determined by two-way ANOVA; ns: non-significant). **D.** Percentage of live and apoptotic cells (Annexin V staining) in NC-ASO and degradative ASO-gIV, g-V treated NF- κ B- MMCLs (XG-7 and H929) day 6 post treatment. (N=3, percentage mean \pm SEM is plotted, and statistical significance was determined by multiple t-test).

Comment A7

The viability curves in Fig 2 c-f demonstrate that *PLUM* oe can to some level rescue toxicity from PIs and IMiDs. Do you think that this effect is specific to these drugs or is this just a general pro-survival effect of *PLUM*? What would happen if you used a generally cytotoxic drug (etoposide?) with *PLUM* oe?

Response A7

Thank you for your insightful question and suggestion. Previously, we did not check the effect of *PLUM* on any other cytotoxic drugs other than BTZ and Len. As suggested, we have now included the following experiment: Drug curve assay for etoposide treatment in VC and *PLUM*-OE MMCLs (KMS11 and LP1). Our results indicated the viability and IC50 values of *PLUM* overexpressed MMCLs compared to vector control (VC) cells upon treatment with etoposide are not significantly different. Hence, it could be possible *PLUM* might be affecting drugs which act on pathways associated with ER stress mediated cell proliferation. Unlike BTZ (proteasome inhibitor), lenalidomide is a known immunomodulatory drug (IMiD) but is also reported to exert ER stress via accumulation of intracellular immunoglobulin proteins which in turn enhances bortezomib induced apoptosis in MM [1]. Thus, our data suggests that *PLUM* mediated resistance effects may be specific to these drugs and not just a general pro-survival effect.

Figure 5: Drug sensitivity IC50 survival curve for Vector control (VC) and FL-*PLUM* OE NF- κ B⁺ mutant MMCLs (KMS11 and LP1) in response to Etoposide (Eto) treatment for 48h (N=3). mean \pm SEM is plotted, and statistical significance was determined by two-way ANOVA; ns: non-significant.

Comment A8

In figure 4a, why is EZH2 expression not reduced following *PLUM* KD?

Response A8

Thank you for the question.

To protect the EZH2 protein from degradation, we performed all our Co-IP experiments for sh-*PLUM* and s-ASO treatment in the presence of a proteasome inhibitor - MG132 (20 μ M). For more clarification, we have revised our text as follows: “Through co-immunoprecipitation (co-IP) assays using scramble and sh2-*PLUM* transduced cells in the presence of MG132 (to protect EZH2 from degradation), we observed a diminished interaction of EZH2 with EED and SUZ12 following *PLUM* KD compared to sh-scramble controls in NF- κ B⁺ MMCLs - KMS11, LP1

and BTZ-R 8226 (Figure 4a and Supplementary figure 5a).” (line 223 to 227 in the revised manuscript)

In accordance, a similar description is provided in the figure and figure legend as well.

Reviewer #2, expertise in lncRNA-mediated chromatin interactions in MM and EZH2 (Remarks to the Author):

The article by Deka et al “Multiple myeloma associated lncRNA PLUM confers chemoresistance by enhancing PRC2 mediated UPR pathway activation” to be considered for publication in Nature Communication concerns the epigenetic regulation via lncRNA and interaction to the PRC2 complex. Findings of new targets for precision medicine including also within the epigenome is indeed imperative as new treatment options to reduce side effects and severe pain for these patients. Deka et al presents a robust paper with a good number of relevant experiments and validation. However, the framing of the paper claiming PLUM to be a novel target for MM resistance seems largely overexaggerated. A similar response concerning the alterations in histone methylation and gene expression will likely be achieved by the use of an EZH2 inhibitor (EZH2i). Is not the resistance induced by EZH2 mainly and PLUM is just a passenger? On this note for clinical relevance, several EZH2 inhibitors have already been tried both in a pre-clinical setting and in clinical trials for multiple myeloma with little success, not because of toxicity but rather lack of an expected response. Therefore, the frame and focus of lncRNAs becoming clinically relevant in treatment regimens should be altered. EZH2 inhibition has already been shown in combinatorial regimes to sensitize myeloma cells to e.g. BTZ treatment and authors therefore need to convince the reader how inhibition of PLUM would promote sensitivity to e.g. BTZ treatment over EZH2i. As such the conclusions lack, at least in part, clinical relevance and might be better suited to be a functional paper rather than framing it as a pre-clinical evaluation of a potential drug target for MM.

General comment:

Comment B1

The results should be presented in the frame of novel partners interacting with the PRC2 complex and put in this context with proper background and references to already published papers in this area. At present the clinical relevance seems overexaggerated and should be tuned down. A good number of relevant experiments and validation are presented in the paper. However, many of the experiments are performed with only one cell line or a limited number of replicates. A suggestion would be that the authors include a few more MM cell lines to confirm that this is not only a cell specific response or an artifact of 1 cell line. Also, many controls and uncropped WB are missing including loading controls and total H3 and GAPDH, as well as molecular weight indication. There is also limited statistical information in the figures and figure legends and no analysis for data normality seems to have been done.

Response B1

Thank you for your insightful comments and suggestion. We agree that the main focus of this study is to characterize and elucidate the molecular mechanisms and biological roles of a newly identified myeloma-associated lncRNA - *PLUM* in MM progression and drug resistance via interaction with the chromatin modifier EZH2/PRC2 complex. We would also like to clarify that our manuscript is not intended as a pre-clinical study of a potential drug target for MM. Additionally, the steric ASOs designed by us are only tested in context to our biochemical findings to evaluate if they can reverse the effect of *PLUM*-EZH2 interactions in promoting proliferation and drug resistance in MM. However, a competitive evaluation of our designed ASOs with current EZH2i is beyond the scope of our current work and can be explored in future work as a separate study.

As suggested, we have rephrased our text as “Our characterization of the myeloma-associated expression of *PLUM* and its role in enhancing EZH2 stability and activity towards chemoresistance suggest its promising potential and future development as a biomarker and/or therapeutic target for MM. Moreover, the similar effects of *PLUM*-EZH2 targeting s-ASOs with that of EZH2 inhibitor (tazemetostat) in the destabilization of EZH2 activity and abrogation of chemoresistance support the possible use of *PLUM* as an alternative target to overcome the resistance mechanisms associated with known EZH2 catalytic inhibitors in MM. Herein, given the unknown role of both *PLUM* and EZH2/PRC2 complex in other physiological processes, we explored an alternative strategy using mixmer ASOs to interrupt *PLUM*-EZH2 interaction and its downstream oncogenic functions efficiently, instead of using inhibitory or degradative ASOs against *PLUM* and EZH2. The specificity of our mixmer ASOs to *PLUM* further minimizes off-target effects, reduces toxicity, and enhances affinity, offering a promising therapeutic approach compared to EZH2 inhibitors. However, a competitive evaluation of these steric ASOs with available EZH2 inhibitors requires further studies.” (line 430 to 443 in the revised manuscript)

To address the reviewer’s concern regarding biological replicates and observations from 1 cell line, we would like to clarify that majority of our experiments were performed in at least two MM cell lines and the data shown are representative of 2-3 biological replicates. In addition, we have repeated multiple experiments in new MM cell lines in our revised manuscript. We have included details of the number of cell lines and biological replicates for all experiments performed as follows:

- For functional assays:
 - i. Proliferation assay: Already performed in three NF-κB mutant MMCLs (KMS11, LP1 and MM.1.S) and two NF-κB non-mutant MMCLs (XG-7 and H9292). (N=3) (Figure 2a, 2b, supplementary figure 2b, 2d).
 - ii. Drug sensitivity assay: Already performed in three NF-κB mutant MMCLs (KMS11, LP1 and MM.1.S) and two NF-κB non-mutant MMCLs (XG-7 and H9292) and were done in triplicates (N=3) (Figure 2c-2f, supplementary figure 3b- 3f).
 - iii. Annexin V staining: Already performed in three NF-κB mutant MMCLs (KMS11, LP1 and MM.1.S) and two NF-κB non-mutant MMCLs (XG-7 and H9292) and were done in triplicates (N=3) (Figure 2c and supplementary figure

2f). Additionally, we performed Annexin V staining in degradative ASO treated KMS11 cells in our revised manuscript. (N=3; supplementary figure 3j)

- For *PLUM* mutants' validation:
 - i. Proliferation and BTZ drug assay already shown in KMS11 (N=2) (Figure 3a, 3b). We have added one more MM cell line (LP1) in the revised manuscript (N=2). (Supplementary figure 4b, 4c).
- RIP-qPCR validation of *PLUM* binding to EZH2:
 - i. Already shown in one MMCL (KMS11) (N=3) (Figure 3g). We have added RIP-qPCR data for another cell line (LP1) (N=2)(Supplementary figure 4d) in our revised manuscript.
- For UPR pathway studies:
 - i. We have already used three NF- κ B mutant MMCLs (KMS11, LP1 and MM.1.S), BTZ S/R cell line (RPMI8226), Len R/S cell line (KMS11) and Dexa S/R cell line (MM.1.S/R), all of which were done in 3 biological replicates (Figure 6c, 6d, 6h)
- Degradative ASO treatment
 - i. We have already used two NF- κ B mutant MMCLs (KMS11, LP1) (N=3). In the revised manuscripts, we included two more non-mutant MMCLs (XG-7 and H929) (N=3) (supplementary 2h, supplementary 3c-3f).
- Co-IP experiment
 - i. We have already used one MMCL (KMS11) and were done in three biological replicates (Figure 4a). In the revised manuscript, we have added two more cell lines (LP1 and BTZ-R RPMI8226). N=2 (Supplementary figure 5a)
- steric ASO treatment and Co-IP and functional assays
 - i. We have already used one MMCL (KMS11) for Co-IP, drug assay, proliferation and AnnexinV assay (N=3). BTZ S/R RPMI8226 and LenS/R KMS11 cell lines were also used for drug assays (N=3). (Figure 5f-5k). In the revised manuscript, we have added Co-IP assay and functional assays for acquired BTZ-R RPMI8226 cell line as well (N=3) (Supplementary figure 6b, 6e, 6f).
- For target validation by ChIP-qPCR
 - i. We have already used two MMCLs (KMS11, LP1), BTZ S/R RPMI8226 cell line and Len S/R KMS11 cell line (N=3). (Figure 7d-6i; supplementary figure 7b, 7c, 7e, 7f)
 - ii. For target (FOXO3 and ZFP36) KD and functional assays: we have used one MMCL (KMS11) (Figure 7j, 7k; supplementary figure 7g-7j) (N=3)

Below are the results for the additional experiments performed in our revised manuscript.

Figure 6: **A.** Graphs representing the proliferative rate of 8 exon deletion mutants of *PLUM* (Δ Ex1, Δ Ex1-2, Δ Ex2, Δ Ex5-6, Δ Ex6-7, Δ Ex7, Δ Ex7-9, Δ Ex8-9) overexpressed MMCLs compared to FL *PLUM* and vector control (VC) overexpressed LP1 cell line till day 4 post transduction and recovery of 48h. Red box: Denotes *PLUM* mutants whose overexpression showed no enhanced proliferative phenotype compared to VC (N=2, OD mean \pm SEM is plotted). **B.** Drug sensitivity IC₅₀ survival curve of 8 exon deletion mutants of *PLUM* (Δ Ex1, Δ Ex1-2, Δ Ex2, Δ Ex5-6, Δ Ex6-7, Δ Ex7, Δ Ex7-9, Δ Ex8-9) overexpressed MMCLs LP1 cells compared to FL *PLUM* and vector control (VC) post 24 h of treatment with BTZ. Red box: Denotes *PLUM* mutants whose overexpression failed to provide resistance to BTZ treatment compared to VC (N=2, mean viability percentage \pm SEM is plotted).

- RIP-qPCR validation of *PLUM* binding to EZH2:

LP1 (Supplementary Figure 4d)

Figure 7: RIP-qPCR validation of *PLUM* with EZH2 antibody compared to IgG antibody in LP1 cell line. (N=2, mean %input \pm SEM is plotted, and statistical significance was determined by unpaired student's t test; *p<0.05, **p<0.005).

- Degradative ASO treatment

A

(Supplementary Figure 2i)

B

(Supplementary Figure 3e, 3f)

Figure 8: A. Proliferation rate of NF- κ B- MMCLs (XG-7 and H929) treated with negative control (NC) ASO and *PLUM* degradative ASOs (Aso-gIV and ASO-gV) for 36 h (N=3, OD mean \pm SEM is plotted,

and statistical significance was determined by two-way ANOVA; * $p < 0.05$, *** $p < 0.0005$). **B.** Drug sensitivity IC50 survival curve for NC-ASO and *PLUM* degradative ASO (ASO-gIV and ASO-gV) treated NF- κ B- MMCLs (XG-7 and H929) in response to BTZ treatment for 24h and Len treatment for 4 days (N=3, mean viability percentage \pm SEM is plotted, and statistical significance was determined by two-way ANOVA; * $p < 0.05$, ** $p < 0.005$, *** $p < 0.0005$).

- For Co-IP experiment,

LP1 and BTZ-R 8226 cell line (Supplementary figure 5a)

Figure 9: Immunoblots showing levels of EED and SUZ12 protein co-immunoprecipitated (Co-IP) by EZH2 antibody in scramble and sh2-*PLUM* KD MMCLs (LP1 and acquired BTZ resistant RPMI8226 cell line) post day2 after selection. Experiment was performed in the presence of MG132 (20 μ M) for all samples. IgG was taken as IP control, GAPDH was used as input protein loading and N=2 biological replicates for LP1 and BRZ R-R8226).

- For steric ASO Co-IP and functional assays

BTZ-R 8226 (Supplementary figure 6b)

Figure 10: Immunoblots showing levels of EED and SUZ12 protein co-immunoprecipitated (Co-IP) by EZH2 antibody in NC-ASO, s-ASO-g4, s-ASO-g11 and s-ASO-g12 treated acquired BTZ resistant RPMI8226 cell line after 3 days of treatment. Experiment was performed in the presence of MG132 (20 μ M) for all samples. IgG was taken as IP control, GAPDH was used as input protein loading control (N=3 biological replicates for KMS11 and N=2 biological replicates for BRZ R-R8226).

To address the reviewer's concern regarding western blot controls, we would like to clarify that all the western blots have been performed using the loading control, GAPDH, as well as H3 for all those with H3K27me3 immunoblot analysis. For clarity, we have now added the loading control details in the figure legends of the revised manuscript. Molecular weight for each protein is also mentioned in the revised manuscript figure files. Statistical information is also provided in the figure legends in more details in the revised manuscript.

Uncropped blots and WB quantification values are also provided in the Source Data file.

Comment B2

Line 42: The authors do not specify any lncRNAs that are currently being evaluated as drug targets in a clinical setting. The papers cited here refer to pre-clinical evaluations and does not at all reflect the current clinical evaluation of lncRNAs. This part of the paper should either be referenced correctly to original papers or omitted.

Response B2

Thank you for your valuable insight and comment. While going through the statement, we realised that the manuscript lines cited by you from line 52 rather than line 42. So accordingly, we have revised the introduction to provide a brief overview of the current clinical evaluation work for lncRNAs and added the requested citations as follows. "Several lncRNAs have been reported to promote chemoresistance in MM by regulating various factors including miRNAs and oncogenes, implicating their potential applications as biomarkers or drug targets for diagnosis, prognosis and therapy [17-19]. Although majority of these studies are still under pre-clinical investigation, the advancement in FDA approved RNA-based therapies such as antisense oligonucleotides (ASOs) for various diseases support the promising prospect of developing lncRNA based therapeutics to improve cancer treatment [20]. While there are currently no FDA approved lncRNA targeting drugs for cancer therapy, a number of clinical trials are ongoing to evaluate the use of lncRNAs as biomarkers for cancer detection and therapy, as well as oligonucleotide based therapeutics targeting oncogenic lncRNAs. Some reported lncRNAs which are in clinical trials showing successful targeting using oligonucleotide based therapeutics include lncRNA *TUG1* in glioblastoma, *H19* in pancreatic cancer, and non-coding mitochondrial RNA in several solid cancers [28-31](database: <http://clinicaltrials.gov>). Other trials also include detecting the expression of lncRNAs like *CCAT1* and *HOTAIR* in CRC patients [24], *HOTAIR* in thyroid cancer [25] and in general using lncRNAs as biomarkers for the detection/prognosis of several other cancer types including lung cancer (NCT03830619), ovarian cancer (NCT03738319) and triple negative breast cancer (NCT02641847). Such developments pave the way towards the discovery and characterization of mechanisms driving the activation and cellular functions of cancer associated lncRNAs. Our previous study showed that dysregulated κ B signalling can alter the myeloma epigenome, in turn regulating gene expression critical for MM progression [14]. Hence, investigating whether this pathway can drive the expression of lncRNAs associated with myeloma progression and resistance to current chemotherapeutic drugs may provide new avenues for the targeted detection and therapy of MM patients." (line 51 to 74 in the revised manuscript)

Comment B3

Line 60: There are recent papers describing the recruiting function of lncRNAs for PRC2 in MM in particular. These results should be cited these to get a proper overview of the field of interacting partners to the PRC2 complex.

Response B3

Thank you for your suggestions. The most recent and relevant publications have been cited in our revised manuscript accordingly. (line 84 to 86 in the revised manuscript)

Comment B4

Line 65: The authors suggests that the lncRNA PLUM might drive chemoresistance but also EZH2. What would be the advantage of targeting the lncRNA over targeting EZH2 directly? If anything the lncRNAs in general have many targets that are not acting on EZH2 or the UPR pathway genes.

Response B4

Considering the extensive list of targets regulated by *PLUM* and EZH2 in general, we designed steric blocking ASOs modified with 2'MOE so that RNase H is unable to degrade the target RNA (unlike ASO gapmers that trigger target degradation). Instead, these ASOs can bind *PLUM* and sterically block binding of EZH2 to *PLUM* (disrupting the *PLUM*-EZH2 RNP complex). Furthermore, we validated the efficiency of these ASOs in modulating the pro-proliferative and chemo-resistance role of *PLUM*-EZH2 complex in MMCLs and tumour xenograft mouse models. Hence, the use of steric ASOs provides an advantage of specificity in targeting the RNP complex (*PLUM*-EZH2) and its downstream pathway (UPR pathway). It further reduces the off-target inhibition of EZH2 and its downstream targets which might have some other unknown functions that are not explored in this study.

Comment B5

Line 106: The text refers wrongly to Figure 1a.

Response B5

Thank you for the correction. It has been changed to Figure 1b. (line 126 in the revised manuscript)

Comment B6

Line 113: In what way can the authors conclude that overexpression of PLUM compared to other cancers has any impact on MM development? The authors have not demonstrated any functional association between PLUM and MM pathogenesis, but rather characterized the expression in different molecular groups and the differential expression in other cancers, none of which suggests pathogenesis or tumor development. This statement should be underbuilt by data or modified.

Response B6

Thank you for raising the concern. We think that there could be some misunderstanding regarding the statement and have modified it to reflect the possible association of increased *PLUM* expression with high-risk MM subtypes carrying NF- κ B activation (NF- κ B+) and patients showing poorer response to VRd treatment regimen (line 135 to 139 in the revised manuscript).

Comment B7

Line 132: The authors claim that *PLUM* is novel to NF-kb/p52 activation, but when inhibiting *PLUM* the results show indication of increased proliferation in both NF-kb+ and NFkb- cell lines. There is also induction of apoptosis in 3 out of the 4 cell lines, so in what way can the authors argue and claim this to be NF-kb activation-specific or are the results independent of NF-kb activation? Did the authors try to use ASOs on the NF-kb- cell lines and if so - what was the results? (see comment Fig 2 Suppl).

Response B7

Thank you for the comment. We would like to clarify the misunderstanding of our findings in this part by the esteemed reviewer. Here, we are not claiming that inhibition of *PLUM* resulted in increased proliferation, instead our data indicated the reverse. Overexpression of *PLUM* increased the proliferation of both NF-kb+ and NFkb- cell lines (Figure 2a, supplementary figure 2b and 2d) whereas shRNA mediated knock down of *PLUM* induced apoptosis (Figure 2b and supplementary figure 2f).

As suggested, we have also performed an additional experiment: degradative ASO treatment of NF- κ B- cells (XG7, H929) followed by proliferation assay and drug sensitivity assay in our revised manuscript. Consistent with the oncogenic functions of *PLUM*, treatment with antisense oligonucleotides (ASOs) designed to degrade *PLUM* reversed the proliferative and drug resistance phenotype of NF- κ B+ MMCLs (KMS11 and LP1) but not that of NF- κ B- MMCLs (XG-7 and H929) (Supplementary figure 2h, 2i and supplementary figure 3c-3f). We have also shown earlier that the endogenous expression of *PLUM* is 5-10 fold lower in NF- κ B- cells (XG-7, H929 and MOLP8) compared to NF- κ B+ cells (Figure supp 1c), suggesting the dependency on increased *PLUM* expression during NF- κ B activation as well as the downstream functions of *PLUM* in promoting the proliferation of myeloma cells.

A

(Supplementary figure 2i)

B

(Supplementary figure 3e, 3f)

C

Figure 11: A: Proliferation rate of NF- κ B- MMCLs (XG-7 and H929) treated with negative control (NC) ASO and PLUM degradative ASOs (Aso-gIV and ASO-gV) for 36 h (N=3, OD mean \pm SEM is plotted, and statistical significance was determined by two-way ANOVA; ns: non-significant). **B, C:** Drug sensitivity IC₅₀ survival curve for NC-ASO and PLUM degradative ASO (ASO-gIV and ASO-gV) treated NF- κ B- MMCLs (XG-7 and H929) in response to BTZ treatment for 24h and Len treatment for 4 days (N=3, mean viability percentage \pm SEM is plotted, and statistical significance was determined by two-way ANOVA; ns: non-significant).

Fig 3h already contains GAPDH blot as a loading control for input samples. Since it is a western blot following RNA-protein protein pull down so, we can only have loading control for our input samples (GAPDH). For pulled elute samples, we cannot have any loading control as we are not aware which protein binds to PLUM positively in all conditions.

Additionally, the molecular weight for all the proteins are mentioned in the main and supplementary figures of the revised manuscript.

Comment B10

Line 178: Have the authors evaluated if the same results can be achieved by using an EZH2 inhibitor? The argument made prior that this is potential novel targeting needs to be certified in a way that it outperforms EZH2 inhibition. The use of EZH2i has already been evaluated in MM clinical trials with very limited successful outcomes.

Response B10

We thank the reviewer for the comment. To examine whether the steric ASOs have similar effects in EZH2 inhibition as known inhibitors, we treated two MM cell lines (KMS11 and BTZ resistant RPMI8226) with tazemetostat, a known EZH2 inhibitor used in clinical trials, and compared its effect with our steric-ASOs (ASO-NC, s-ASO-g4, s-ASO-g11 and s-ASO-g12) on the stability and activity of EZH2 via immunoblotting for p-EZH2, EZH2, H3K27me3 and H3 protein (supplementary figure 6d). We also performed drug sensitivity assay in the presence of tazemetostat and compared its efficiency with the steric ASOs in increasing the sensitivity of MM cells to BTZ and Len treatment (Supplementary figure 6g and 6h). Here, we observed that the effects of our sASO treatment are comparable to that of tazemetostat in disrupting EZH2 stability/activity as well as sensitizing myeloma cells to BTZ and Len treatment.

(Supplementary figure 6d- KMS11 and BTZ-R 8226)

A

B

(Supplementary figure 6g-KMS11)

C

(Supplementary figure 6h-BTZ-R 8226 and Len-R KMS11)

Figure 12: A. Immunoblot analysis of p-EZH2/EZH2, H3K27me3/H3 and GAPDH expression levels in steric ASOs (0.5 μ M) treated and EZH2 inhibitor (Tazemetostat) (0.5 μ M) treated cell lines (KMS11 and BTZ-R 8226). ASO-NC and DMSO is used as negative control for s-ASO and EZH2 inhibitor treatment respectively. GAPDH is used as loading control for all the samples and treatment was given for 3days. (N=3 biological replicates). B. Drug sensitivity IC₅₀ survival curve for steric ASOs and EZH2i (Tazemetostat) treated KMS11 cells in response to BTZ treatment for 24 h and Len treatment for 4 days respectively (N=3, mean \pm SEM is plotted, and statistical significance was determined by two-way ANOVA; **p<0.005, ***p<0.0005, ns: non-significant). C. Drug sensitivity IC₅₀ survival curve for steric ASOs and EZH2i (Tazemetostat) treated BTZ resistant R8226 cell line in response to BTZ treatment for 24 h and Len resistant KMS11 cell line in response to Len treatment for 4 days

respectively (N=3 mean \pm SEM is plotted, and statistical significance was determined by two-way ANOVA; *p<0.05, ns: non-significant).

Comment B11

Fig 1a: The article includes functional studies showing that PLUM interacts with the catalytic subunit EZH2 of the PRC2 to regulate histone methyltransferase activity and thus modulate the expression of tumour suppressor genes to activate the unfolded protein response (UPR). Why did the authors specifically select PLUM above other lncRNAs? In figure 1a it also looks like PLUM has a negative log2FC suggesting that it is downregulated as compared to control? Figure 1a is overall very difficult to understand and should be clarified.

Figure 1c: These differences referred to in the text (line 110) are not significant. This should be omitted or clarified. What does the results really show?

Response B11

Thank you for the comment. We have explained our prioritization strategy and provided our rationale for specifically selecting *PLUM* over other lncRNAs in response A3 of this rebuttal. In brief, although *lnc-NTMT1-2* was the most significantly regulated lncRNA following p52 KD, *PLUM* expression was most highly enriched in MM cell lines compared to other 51 cancer types from the CCLE dataset (Figure 1e). In contrast, *lnc-NTMT1-2* expression was not selectively enriched in MM cell lines relative to other cancer types. As our study focused on potential p52-regulated lncRNAs linked to MM progression, we prioritized our investigation of *PLUM* functions over *lnc-NTMT1-2*. Additionally, there is no significant difference in *lnc-NTMT1-2* expression levels between VRd responsive and non-responsive MM patients.

Figure 1a represent all the transcripts downregulated (Down DEGs) upon NF- κ B/p52 knock down. Hence, negative log2FC means *PLUM* expression is negatively correlated with p52 expression. For better clarity, we have revised our text and figure legend as follows: “To understand the role of ncNF- κ B regulated lncRNAs in MM progression, we performed RNA seq analysis of control versus NF- κ B/p52 knockdown (KD) MM cells, which identified several p52 regulated non-coding transcripts showing negative correlation with the expression of p52 (Blue circles in Figure 1a).” (line 109 to 112 in the revised manuscript)

Figure 1c is an integrated heatmap of control versus p52 KD RNA seq with high risk MAF and HRD low TP53 patient samples subdivided by NF- κ B activity (NF κ B⁺ versus NF κ B⁻) identified in DESeq2 (Wald, FDR \leq 0.1) analysis of CoMMpass dataset.

The plot shows all the p52 regulated lncRNAs which are either enriched in both the MAF and HRD low TP53 patient samples or enriched in one subtype and depleted in another subtype. Furthermore, those transcripts were subdivided by NF- κ B activity (NF κ B⁺ versus NF κ B⁻). For *PLUM*, it is downregulated upon p52 KD, and is enriched in both the MAF and HRD low TP53 patient samples harbouring high NF- κ B activity (NF- κ B⁺ samples).

Comment B12

Fig 1a Suppl.: The knockdown of the p52 looks very minor and not at all convincing on the western blot. In general evaluation of western blots are very difficult. The authors should provide uncropped images of the WB and add molecular weight indication.

Response B12

Thank you for the comment. All the uncropped versions of WBs are provided in the “Source Data file” in our revised submission. For Supplementary Fig1a, knock down was performed in biological triplicates in two cell lines for which we have also provided the densitometry values of the bands along with statistical analysis in our source data file.

Figure 13: Boxplots showing the densitometric values for p52 band in western blots following CRISPR KD of p52 in KMS11 and LP1 cell lines. Analysis is done using GelQuant.NET. The values for p52 band were normalised to GAPDH band and mean \pm SEM is plotted. Statistical significance was determined by unpaired student's t test.

Comment B13

Fig 1c Suppl: No statistics is presented. This should be added for proper evaluation.

Response B13

Thank you for the concern. We have included the statistics for Supplementary Fig 1c in our revised manuscript.

Comment B14

Fig 2 Suppl: Did the authors use ASOs on the NF-kb- cell lines and if so what was the results?

Response B14

Thank you for the comment. As suggested, we have performed the experiment using NF- κ B-cell lines (XG-7 and H929) and explained the results in our earlier “Response B7, Figure 11 of rebuttal”.

Comment B15

Fig 3: Exon deletion mutations: It is very difficult to see those lines to make a conclusion. The authors state that mutant deficient in exon 7 lose their oncogenic effects and BTZ resistance but binds EZH2, while the full length and exon 1 deficient does not. In what way does this drive resistance?

Response B15

Thank you for the comment.

For better interpretation of the data, we have provided separate plots for mutants that lose their oncogenic effects and for mutants that display similar effects in promoting myeloma proliferation and chemoresistance as FL-*PLUM* in the rebuttal letter.

A

B

Figure 14: **A.** Graph showing the proliferation rate of MMCLs (KMS11 and LP1) overexpressed with vector control (VC-black line), FL-*PLUM* (green line) and *PLUM* mutants that showed loss of proliferative phenotype as compared to FL-*PLUM*. The proliferation rate of the mutants that carried deletion in exon 1 and exon 7 (Δ Ex1, Δ Ex1-2, Δ Ex6-7, Δ Ex7, Δ Ex7-9) behaves similar to VC transduced cells. **B.** Graph showing the proliferation rate of MMCLs (KMS11 and LP1) overexpressed with vector control (VC-black line), FL-*PLUM* (green line) and *PLUM* mutants (Δ Ex2, Δ Ex5-6 and Δ Ex8-9) that showed enhanced proliferation as compared to VC. The proliferation rate of the mutants is similar to that of FL-*PLUM* overexpressed cells.

Figure 15: **A.** Graph showing the viability percentage of MMCLs (KMS11 and LP1) overexpressed with vector control (VC-black line), FL-*PLUM* (green line) and *PLUM* mutants that showed reduced viability percentage as compared to FL-*PLUM* upon treatment with BTZ for 24h. The viability percentage of the mutants that carried deletion of exon 1 and exon 7 (Δ Ex1, Δ Ex1-2, Δ Ex6-7, Δ Ex7, Δ Ex7-9) behaves similar to VC transduced cells. **B.** Graph showing the viability percentage of MMCLs (KMS11 and LP1) overexpressed with vector control (VC-black line), FL-*PLUM* (green line) and *PLUM* mutants (Δ Ex2, Δ Ex-5-6 and Δ Ex8-9) that showed enhanced viability percentage as compared to VC upon treatment with BTZ for 24h. The viability of the mutants is similar to that of FL-*PLUM* overexpressed cells.

Also, regarding binding data, we would like to clarify that there is a misunderstanding with our concluded inference. We instead state that *PLUM*- Δ exon7 mutant not only lose its oncogenic effect and BTZ resistance phenotype but also fails to bind EZH2 (Figure 2h). We have also validated similar phenotypic effects in an additional cell line (BTZ-R 8226) in our revised manuscript (Supplementary figure 4e). Hence, with experimental validation using steric ASOs, we show that *PLUM* drives resistance via interaction with EZH2. The use of steric ASOs that disrupt the binding between EZH2 and *PLUM* results in the loss of resistance to both BTZ and Len drug treatment in NF- κ B + MMCLs as well as acquired resistant cell lines (Figure 5j, 5k).

Supplementary figure 4e

Figure 16: Immunoblots showing differential binding of PRC2 complex proteins (EZH2, EED and SUZ12) in BTZ-R 8226 cell line with FL *PLUM*, Δ Exon1 and Δ Exon7 mutants of *PLUM*. IRE transcript is used as negative control and GAPDH is used as input protein loading control. (N=2 biological replicates)

Comment B16

Fig 3 f and g: The authors decided only to use one cell line here and only 2 replicates yet have statistics on the data! This should be clarified and corrected.

Response B16

Thank you for the comment. In our revised manuscript, we have added one more biological replicate of RIP-qPCR in KMS11 (total N=3) (Figure 3g) and two additional biological replicates of RIP-qPCR in LP1 cell line in supplementary (Supplementary figure 4d).

Comment B17

Fig 4: Was the co-immunoprecipitation (co-IP) only done once or in triplicates?

Response B17

Thank you for the question.

Yes, Co-IP of figure 4a was performed in triplicates. The western blots for all the reps are provided in the “Source Data file” in our revised submission.

Additionally, in our revised manuscript, we have added Co-IP data for two more cell lines (LP1 and BTZ-R 8226) with two biological replicates for each cell line (Supplementary figure 5a)

LP1 and BTZ-R 8226 cell line (Supplementary figure 5a)

Figure 17: Immunoblots showing levels of EED and SUZ12 protein co-immunoprecipitated (Co-IP) by EZH2 antibody in sh-scramble and sh2-*PLUM* KD MMCL (LP1) and BTZ-R 8226 cell line after two days of selection. Experiment was performed in the presence of MG132 (20µM) for all samples. IgG was taken as IP control, GAPDH was used as input protein loading control (N=2 biological replicates).

Comment B18

Fig 4g: This is only shown in one, very specific MM cell line which limits the impact of the functional experiment and validation. This is also true for many of the other experiments. The authors should include a few more MM cell lines to confirm that this is not only an artifact of 1 cell line.

Response B18

Thank you for the comment. For fig 4g, we have added data for one more cell line (BTZ-R 8226) in our revised manuscript (Supplementary figure 5d).

BTZ-R 8226 cell line (Supplementary figure 5d)

Figure 18: Immunoblots showing the binding level of EZH2 protein pulled with FL *PLUM* in CDK1 inhibitor (R03306) treated acquired BTZ resistant RPMI8226 cell line. DMSO is taken as vehicle control. IRE transcript is taken as negative control and GAPDH is used as input protein loading control. (N=2 biological replicates)

For other experiments, number of cell lines used for all the experiments are provided in “comment B1”.

Comment B19

Fig 4i: Results should be shown whether this causes reduction of H3K27me3 either globally or site specific.

Response B19

Thank you for the comment. We would like to clarify that these four sites of mutations were selected based on available literature and our in-silico docking data (Figure 4e). In particular, all the mutants have been reported to result in reduced H3K27me3 activity from previous publications [2-6]. Hence, we focused on examining how these mutations might alter the interaction of EZH2 with *PLUM* and PRC2 complex formation. Our data from Fig 4h and 4i suggest that the mutants of EZH2 (HA-Mut2, HA-Mut3 and HA-Mut4) which fail to bind *PLUM* (Figure 4h) are also defective in the formation of PRC2 complex with EED and SUZ12 (Figure 4i).

Comment B20

Fig 5g: This figure does not show any reduction of H3K27me3 so how does the authors know that this treatment reduced EZH2 activation?

Response B20

Thank you for the comment. There is a significant reduction in the phospho-EZH2 and H3K27me3 levels of sASO-g11 and sASO-g12 treated cells, from which we infer that sASO treatment disrupted *PLUM*-EZH2 interaction, further reducing EZH2 activation. For better inference, we have provided densitometric analysis of the western blots (N=5 biological replicates for KMS11) in this rebuttal submission. We have added western blot data in one more cell line BTZ-R 8226 for which densitometric analysis is provided (N=3 biological replicates for BTZ-R 8226). The quantification values are provided in Source data file (Figure 5g for KMS11 and Supplementary figure 6d for BTZ-R 8226).

Figure 19: A, B. Boxplots showing densitometric quantification of pEZH2/EZH2 and H3K27me3/H3 levels w.r.t. GAPDH in KMS11 and BTZ-R 8226 cell lines. (Analysis was done using GelQuant.NET). N=5 biological replicates for KMS11 and N=3 biological replicates for BTZ-R8226; mean \pm SEM is plotted; statistical significance was determined by unpaired student's t test).

Comment B21

Fig 5h and i: The changes here look very limited, what statistical test was used here? The results should also be validated in other MM cell lines.

Response B21

Thank you for the comment. For 5h, two-way ANOVA is used to determine the p-value. For 5i, student's t-test was used to determine p-value. For further validation, we performed proliferation assay in an additional cell line (BTZ-R 8226). Data is provided in supplementary figure 6e, 6f.

BTZ-R 8226 cell line (Supplementary figure 6e, 6f)

Figure 20: A. Proliferation rate of BTZ-R 8226 cells treated with NC-ASO, s-ASO-g4, s-ASO-g11 and s-ASO-g12 till day 4 post treatment with ASOs (24 h) (N=3, mean \pm SEM is plotted, and statistical significance was determined by two-way ANOVA; **p<0.005, ns: non-significant). **B.** Box plot

depicting the percentage of annexin + cells (Red colour) and annexin – cells (Green color) cells post treatment with steric ASOs. (Mean \pm SEM for annexin+ versus annexin- is plotted, and statistical significance was determined by multiple t-test; **p<0.005, ***p<0.0005).

Comment B22

Line 260 and Fig 5: Does PLUM or EZH2 contribute to the resistance here? With lacking information that PLUMi outperforms EZH2i in a clinical setting this statement becomes rather overexaggerated.

Response B22

Thank you for the comment. Our data suggests that both *PLUM* and EZH2 are important in conferring resistance through their interaction. As shown in figure 5, treatment of steric ASOs, which disrupted *PLUM*-EZH2 interaction, resulted in the significant reduction of IC50 values for both BTZ and lenalidomide treatment in three MMCLs. Hence, our data indicates the major role of *PLUM*-EZH2 complex in promoting drug resistance in MM.

Additionally, our new data using EZH2 inhibitor (tazemetostat) in two MM cell lines (KMS11 and BTZ-R 8226) suggest that the effect of our steric-ASOs (ASO-NC, s-ASO-g4, s-ASO-g11 and s-ASO-g12) treatment in sensitizing MMCLs to BTZ and Len exposure are comparable to that of EZH2 inhibitor (supplementary figure 5k,l). Refer to data shown in Response B10 (Rebuttal Figure 12).

Comment B23

Fig 6a: 9400 genes seem utterly a large number as this would almost affect 50% of the genome. Perhaps a cut/off is needed?

Response B23

We appreciate the reviewer's concern regarding the large number of differentially expressed genes identified in our original analysis. Following Reviewer A's suggestion addressing a similar point, we have revised our RNA-seq analysis to adopt Independent Hypothesis Weighting (IHW) for better false discovery rate control and a more stringent statistical threshold (≤ 0.05) in line with standard RNA-seq analysis practices, eliminating 1248 low confidence genes. Downstream analyses such as the UPR Gene Set Enrichment Analysis showed increased significance.

Figure 21: **A.** Heat map representing the differential gene expression (DEGs) of sh-scramble versus sh-*PLUM* KD KMS11 cells 3 days post transduction (adjusted p-value ≤ 0.05). **B.** Gene Set Enrichment Analysis (GSEA) showing the association of significantly downregulated genes from *PLUM* KD KMS11 cells with UPR pathway. NES: -1.68; FDR: 0.00353.

Comment B24

Fig 6c and d: This WB has to be explained better, what are the differences here?

Response B24

Thank you for the comment. Figure 6c shows the immunoblot analysis of UPR pathway markers in three NF- κ B+ MMCLs (KMS11, LP1 and MM.1.S) following sh-*PLUM* knock down compared to sh-scramble conditions whereas figure 6d shows the immunoblot analysis of UPR pathway markers in three NF- κ B+ MMCLs (KMS11, LP1 and MM.1.S) following *PLUM* overexpression compared to vector control (VC) overexpressed cells.

To improve clarity, we have provided additional labels in the figure for sh-*PLUM* KD and *PLUM* overexpression conditions used respectively. We have also provided more details in the figure legends of our revised manuscript.

Reviewer #3, ECR (Remarks to the Author):

Reviewer #4, expertise in in silico docking (Remarks to the Author):

I restrict my comments to the in-silico modelling work in this manuscript.

Comment D1

1. To produce molecular models for the PLUMB-EZH2 RNA-protein complex, the authors first generate models for the two independent components using a variety of structure prediction tools, filter the results for plausibility on the basis of scoring metrics, then use docking (HDock) to generate alternative models for the complexes. The issue with this approach is that a critical section of EZH2 (residues 480-515), is observed to be disordered in the available crystal structure (5HYN). This includes the highly basic (and so highly positively charged) RKKKR patch (residues 490-994) that unsurprisingly is highly implicated in RNA binding. The fact that the region is observed (in the context of the protein component alone) to be dynamic and disordered argues that it is interaction with the RNA component that is required to bring about its structural organisation. There is therefore a chicken-and-egg problem here: the protein-RNA docking method used here requires a complete structural model for each component, but an accurate model for the complete protein component requires a prediction of the RNA-protein interaction. With the recent release of AlphaFold3, the authors have an opportunity to evaluate the prediction of the complete complex ab initio, overcoming this issue, and this should be done.

Response D1

We would like to thank the reviewer for pointing this out and for their great suggestion on using AlphaFold3, which can predict the structure of the *PLUM*-EZH2 RNA-protein complex directly from the sequences. We predicted the structure of the complex (EZH2-*PLUM*) using AlphaFold3, which showed binding of EZH2 protein maximum to exon7 of *PLUM* with inclusion of exon3 and exon8 to some extent. We are providing the docked structure, Predicted Aligned error (PAE) figure, along with the interface for the interacting residues in the rebuttal (Figure 22) and revised manuscript (Supplementary figure 5c). Additionally, the docked model generated by AlphaFold3 also showed interaction of the disordered region residues on EZH2 (491 to 497) with atoms in the exon7 of *PLUM*.

From the pIDDT scores, it is clear that in the predicted structure of the complex the confidence score for most of the *PLUM* RNA structure and the loop regions of the EZH2 proteins is low (pIDDT < 50). The ipTM score of the interface region is also quite low (0.17). To address the point about the disordered region in the EZH2 protein, which can only be structurally organized after the complex formation with *PLUM*, we compared the structure of the EZH2 subunit from the EZH2-*PLUM* complex and EZH2 protein alone, both predicted by AlphaFold3, using structural alignment and reporting the RMSD between the complexed-EZH2 and independent-EZH2. The all-atom RMSD between the two structures was 21.71 Å, which was largely due to

the non-aligned region (shown in red for EZH2 from the *PLUM*-EZH2 complex and orange for the solo-EZH2 structure; Rebuttal Figure 24). Interestingly, the RMSD between the 480-515 region was only 1.6 Å.

As the confidence of the predicted structure is low and the limitations of these in-silico dockings (especially in case of disordered regions and lncRNA) are well known, we would like to point out that the in-silico modelling analysis along with information about the earlier confirmed interacting regions of EZH2 in previous literature is simply used as a guide to inform the design of the steric-ASO experiment. We looked at the interface of the predicted complex structure by AlphaFold3 and it confirmed an interaction between the disordered 490-497 region of EZH2 and Exon 7 of *PLUM*. We have listed the interacting residues at the interfaces below:

List showing the interacting residues on chain A (EZH2) and chain B (*PLUM*) in EZH2-*PLUM* complex generated by Alphafold3

Protein – EZH2 (Chain A):

PRO 12, VAL 13, ARG 16, LYS 20, SER 21, MET 24, ARG 25, GLN 28, ARG 31, ARG 78
THR 144, GLU 147, GLU 148, LYS 151, ARG 161, GLU 162, LYS 491, LYS 492, LYS 493
ARG 494, LYS 495, ARG 497, LYS 563, GLN 570, LYS 634, LYS 713, ARG 714, ALA 715
SER 729, GLN 730, ALA 731, LEU 734

RNA – *PLUM* (Chain B):

C 71, U 72, C 106, U 107, C 402, U 403, U 404, A 405, A 406, U 412 C 413, U 414, A 557, U
558, U 559, G 586, G 587, G 588, U 600, G 601 A 992, C 993, C 994, A1000, A1001, A1002,
U1003, U1011, A1013, G1270 A1271, A1272, C1273

Residue (Chain B)	Exon	Residue (Chain B)	Exon
C 71	Exon 1	G 587	Exon 4
U 72	Exon 1	G 588	Exon 4
C 106	Exon 1	U 600	Exon 4
U 107	Exon1	G 601	Exon 4
C 402	Exon 3	A 992	Exon 7
U 403	Exon 3	C 993	Exon 7
U 404	Exon 3	C 994	Exon 7
A 405	Exon 3	A 1000	Exon 7
A 406	Exon 3	A1001	Exon 7
U 412	Exon 3	A 1002	Exon 7
C 413	Exon 3	U 1003	Exon 7
U 414	Exon 3	U 1011	Exon 7
A 557	Exon 4	A 1013	Exon 7
U 558	Exon 4	G 1270	Exon 8
U 559	Exon 4	A 1271	Exon 8
G 586	Exon 4	A 1272	Exon 8

Table 1: Table showing the residues on chain B (*PLUM*) and its corresponding exon number that are found to interact with EZH2 in the EZH2-*PLUM* complex generated by Alphafold3.

List showing the interacting atoms from the RNA (*PLUM*) with 491 to 497 regions on EZH2. The data showed interaction to be with 994, 1000, 1001, 1002, 1003, 1010, 1011 atoms on PLUM which are part of its exon7.

LYS 491 A CA	U	1003 B OP1	4.84	7 1
LYS 491 A CA	U	1003 B O5'	5.37	7 2
LYS 491 A C	U	1003 B O5'	5.6	6 2
LYS 491 A CB	U	1003 B C6	6.21	4 5
LYS 491 A CD	U	1003 B C5	5.78	4 5
LYS 491 A CD	U	1003 B C6	5.54	4 5
LYS 491 A CE	U	1003 B OP1	5.69	7 1
LYS 491 A CE	U	1003 B O5'	5.69	7 2
LYS 491 A CE	U	1003 B O4'	5.87	7 2
LYS 491 A NZ	U	1003 B C6	6.04	3 5
LYS 492 A N	A	1002 B O3'	5.43	3 2
LYS 492 A N	U	1003 B OP1	5.03	3 1
LYS 492 A N	U	1003 B O5'	5.51	3 2
LYS 492 A C	U	1003 B O5'	5.57	6 2
LYS 492 A O	A	1002 B O2'	3.77	2 1
LYS 492 A CB	A	1002 B C4'	6.18	4 8
LYS 493 A N	U	1003 B O5'	5.86	3 2
LYS 493 A N	U	1003 B C5'	5.83	3 8
LYS 493 A CA	A	1002 B O2'	5.44	7 1
LYS 493 A CA	U	1003 B C5'	5.65	7 8
LYS 493 A CG	U	1003 B C5'	5.22	4 8
LYS 493 A CG	U	1003 B C5	4.06	4 5
LYS 493 A CG	U	1003 B C6	4.42	4 5
LYS 493 A CD	U	1003 B C5	3.58	4 5
LYS 493 A CE	U	1003 B O4	3.5	7 2
LYS 493 A CE	U	1003 B C5	2.45	7 5
LYS 493 A CE	U	1003 B C6	3.35	7 5
LYS 493 A NZ	U	1003 B C4	3.16	3 5
LYS 493 A NZ	U	1003 B O4	2.83	3 2
LYS 493 A NZ	U	1003 B C5	2.9	3 5
ARG 494 A N	A	1002 B O2'	5.16	3 1
ARG 494 A CD	A	1002 B O2'	5.26	7 1
ARG 494 A CD	A	1002 B C1'	6.05	7 6
ARG 494 A NE	A	1002 B O2'	5.68	3 1
ARG 494 A NE	A	1002 B C1'	6.13	3 6
ARG 494 A CZ	A	1001 B N3	5.52	6 3
ARG 494 A CZ	A	1002 B C1'	6.25	6 6
ARG 494 A NH1	A	1000 B C2	6.07	3 5
ARG 494 A NH1	A	1001 B C2	4.8	3 5
LYS 495 A CE	U	1003 B C4	5.76	7 5
LYS 495 A CE	U	1003 B O4	5.23	7 2
LYS 495 A CE	U	1003 B C5	5.58	7 5
LYS 495 A NZ	U	1003 B O4	5.06	3 2
ARG 497 A CZ	U	1010 B O2	6.06	6 2
ARG 497 A NH1	U	1011 B O2	5.31	3 2
ARG 497 A NH2	C	994 B OP1	5.6	3 1

ARG 497 A NH2 | U 1011 B C1' | 5.77 | 3 6
 ARG 497 A NH2 | U 1011 B O2 | 4.18 | 3 2

994, 1000, 1001, 1002, 1003, 1010, 1011 (all of these are residues on exon7) which have shown interaction with 491 to 497 regions on EZH2.

Figure 22: The structure of the *PLUM*-*EZH2* complex generated using AlphaFold3. The colour shading of the structure represents the pIDDT confidence score of the structure. The pTM score (predicted TM-score for the full structure) is 0.35 and the ipTM (predicted TM-score of the interfaces between subunits) is just 0.17.

Figure 23: Predicted Aligned error (PAE) showing the errors (in Angstroms) between the relative positions of residues within the predicted complex. It shows relatively higher confidence between the

residue pairs within the EZH2 protein, but the residue pairs within the PLUM and between PLUM-EZH2 have low confidence (large errors).

Figure 24: Superimposition of the EZH2 structure from the PLUM-EZH2 complex (in cyan) and solo-EZH2 structure (light grey), both predicted by AlphaFold3. The all-atom RMSD between the two structures is 21.7 Å. The non-aligned region in both the structures is shown in red for EZH2 from the PLUM-EZH2 complex and orange for solo-EZH2 model.

Comment D2

2. The in-silico structure prediction work falls into two sections. In the first it is assumed (on the basis of significant experimental data) that the key interaction is between EZH2 and exon7=PLUM, and so molecular models for this complex are generated (with the issues described above) in order to predict key RNA-interacting residues in EZH2. In the second section the assumption that exon7 is involved is put aside, and models for the complex of full-length PLUM with EZH2 are generated, in order to predict key protein-interacting regions in PLUMB. What is missing though is any presentation or discussion of the correlation between these two pieces of work. The authors must demonstrate that the structure predictions for FL-PLUMB-EZH2 recapitulate (in the exon-7 region) the structure prediction for exon-7 PLUMB-EZH2; so that e.g., prediction of key RNA-interacting regions of the protein are the same. Of course, if they do then one can question the value of the “exon 7-only” work; while if they disagree, this is a significant issue that must be addressed.

Response D2

We thank the reviewer for such an insightful comment and suggestion. To address the issue raised, we have used Alphfold3 to predict the *PLUM*-EZH2 docked model and discussed the results in response D1. To correlate between the two prediction models we used in the manuscript (Exon7-*PLUM*-EZH2 and FL-*PLUM*-EZH2), we are providing the interface file in one dimension for all the predicted models in “supplementary table 5” which suggest that FL-*PLUM*-EZH2 model 3 can recapitulate the Exon7-*PLUM*-EZH2 docked model to some extent.

For both models, the residues between 489-494 in EZH2 are common for the interaction with *PLUM*.

We also thank the reviewer for the critical question raised on the value of the “exon 7-only” work. To clarify the concern, we have considered all the exons to have potential functional roles in our initial experiments. However, site directed mutagenesis experiments of *PLUM* revealed the oncogenic effects of this lncRNA are mediated via exon 1 and exon 7 (Figure 3a, 3b). Subsequently, RNA protein pull-down and MS analysis further narrowed down the differential binding of *PLUM* with EZH2 to exon7 only (Figure 3d, 3e). Hence, we performed our first docking experiment with exon7-*PLUM* and EZH2 only to identify the possible interacting residues on EZH2 for downstream experimental validation.

For the steric ASO experiments, we have considered all the three predicted models for FL *PLUM*-EZH2 docking data (Figure 5a, 5b, 5c) and designed our steric ASOs against all the interacting regions between EZH2 and *PLUM* (inclusive of all exons): s-ASO-g2 against exon1, s-ASO-g3 against exon9, s-ASO-g4 against exon4 (Model 1-Figure 5a); s-ASO-g6 against exon4, s-ASO-g7 against exon5, s-ASO-g8 against exon5, s-ASO-g9 against exon9 (Model 2 – Figure 5b); s-ASO-g11 against exon6-exon7 junction, s-ASO-g12 against exon7 and s-ASO-g14 against exon8 (Model 3 – Figure 5c). Furthermore, all the steric ASOs were tested for their activity to validate the in-silico docking data using wet lab experiment (RIP-qPCR) (Supplementary figure 6a and figure 5e). The results showed s-ASO-g11 and s-ASO-g12 targeting exon6-exon7 junction and exon7 respectively are most effective in disrupting the *PLUM*-EZH2 complex. Hence, based on these validation work, we confirmed our experimental focus on “exon 7 only” to perform other downstream experiments.

A

FL PLUM -EZH2 docked model	Steric ASO	Targeting Exon on PLUM
Model1	s-ASO-g2	Exon1
Model1	s-ASO-g3	Exon9
Model1	s-ASO-g4	Exon4
Model2	s-ASO-g6	Exon4
Model2	s-ASO-g7	Exon5
Model2	s-ASO-g8	Exon5
Model2	s-ASO-g9	Exon9
Model3	s-ASO-g11	Exon6-exon7 junction
Model3	s-ASO-g12	Exon7
Model3	s-ASO-g14	Exon8

B

Figure 25: A. Table showing the designed steric ASOs and their respective targeting exons on *PLUM* based on predicted interacting regions between FL *PLUM*-EZH2 docked models (Model1, Model2 and Model3). **B.** RIP validation of *PLUM* with EZH2 antibody compared to IgG antibody in KMS11 post treated with NC-ASO and steric ASOs (s-ASO-g2, s-ASO-g3, s-ASO-g4, s-ASO-g6, s-ASO-g7, s-ASO-g8, s-ASO-g9, s-ASO-g11 and s-ASO-g12) for 3days (N=2, mean %input \pm SEM is plotted, and statistical significance was determined by unpaired student’s t test).

Comment D3

3. In any case I find the “exon-7 only” modelling section weaker, in that it does not drive any subsequent experimental work or provide significant additional insights. The prediction that the RKKKR patch is involved in RNA recognition is hardly surprising. What could have been more interesting was the prediction of two other RNA-recognising regions (562-575 and 586-602), but I do not see this discussed or followed up in any way. As an aside, the authors should clarify their wording on lines 199-200: the statement “the C-terminal disordered region of EZH2, ranging from aa residue 488 to 602 (Figure 4e)” could be misleading, as the crystal structure only shows residues 479-515 as disordered (depending slightly on which protein chain is being discussed). This means that these two extra patches correspond to well-structured regions of the protein, and so, ultimately, might be more attractive targets for the future development of therapeutic modalities.

Response D3

We appreciate the valuable comment provided by the reviewer on “exon-7-only” and validation work on RKKKR patch towards RNA binding in our downstream experiments. We have clarified our rationale for selecting exon-7 in the previous response D2.

Additionally, we moved ahead to work with 489-RKKKR-494 region based on previous published data on importance of disordered region of protein in RNA recognition and interaction [7-11]. For EZH2 also, this region was already shown to have RNA binding role in-vitro which we further attempted to validate in a cell line model [2, 5]. We agree that other two regions (562-575 and 586-602) could be of interest but due to the reason mentioned earlier, we concentrated more on the 489-RKKKR-494 region. Also, in FL-*PLUM*-EZH2 docked models, these two regions did not show any interaction with *PLUM* in either of the three models. Instead, our revised docking model using Alphafold3 (as suggested) also showed the involvement of disordered region from 491-497 on EZH2 in the interaction with exon7-*PLUM*, further suggesting the importance of this region on EZH2 towards its RNA binding affinity.

As suggested, we have also reframed our writing in the revised manuscript as “From the docking data, the most significant interaction was observed with the C-terminal region of EZH2, 488-498 residues within known disordered region of EZH2 (479-515) and two additional structured region (562-575 and 586-602) (Figure 4e)” (line 234 to 237 in the revised manuscript)

List showing the interacting atoms from the RNA (PLUM) with 491 to 497 regions on EZH2. The data showed interaction to be with 994, 1000, 1001, 1002, 1003, 1010, 1011 atoms on PLUM which are part of its exon7.

LYS	491	A	CA	U	1003	B	OP1	4.84	7	1
LYS	491	A	CA	U	1003	B	O5'	5.37	7	2
LYS	491	A	C	U	1003	B	O5'	5.6	6	2
LYS	491	A	CB	U	1003	B	C6	6.21	4	5
LYS	491	A	CD	U	1003	B	C5	5.78	4	5
LYS	491	A	CD	U	1003	B	C6	5.54	4	5

LYS	491	A	CE	U	1003	B	OP1	5.69		7	1
LYS	491	A	CE	U	1003	B	O5'	5.69		7	2
LYS	491	A	CE	U	1003	B	O4'	5.87		7	2
LYS	491	A	NZ	U	1003	B	C6	6.04		3	5
LYS	492	A	N	A	1002	B	O3'	5.43		3	2
LYS	492	A	N	U	1003	B	OP1	5.03		3	1
LYS	492	A	N	U	1003	B	O5'	5.51		3	2
LYS	492	A	C	U	1003	B	O5'	5.57		6	2
LYS	492	A	O	A	1002	B	O2'	3.77		2	1
LYS	492	A	CB	A	1002	B	C4'	6.18		4	8
LYS	493	A	N	U	1003	B	O5'	5.86		3	2
LYS	493	A	N	U	1003	B	C5'	5.83		3	8
LYS	493	A	CA	A	1002	B	O2'	5.44		7	1
LYS	493	A	CA	U	1003	B	C5'	5.65		7	8
LYS	493	A	CG	U	1003	B	C5'	5.22		4	8
LYS	493	A	CG	U	1003	B	C5	4.06		4	5
LYS	493	A	CG	U	1003	B	C6	4.42		4	5
LYS	493	A	CD	U	1003	B	C5	3.58		4	5
LYS	493	A	CE	U	1003	B	O4	3.5		7	2
LYS	493	A	CE	U	1003	B	C5	2.45		7	5
LYS	493	A	CE	U	1003	B	C6	3.35		7	5
LYS	493	A	NZ	U	1003	B	C4	3.16		3	5
LYS	493	A	NZ	U	1003	B	O4	2.83		3	2
LYS	493	A	NZ	U	1003	B	C5	2.9		3	5
ARG	494	A	N	A	1002	B	O2'	5.16		3	1
ARG	494	A	CD	A	1002	B	O2'	5.26		7	1
ARG	494	A	CD	A	1002	B	C1'	6.05		7	6
ARG	494	A	NE	A	1002	B	O2'	5.68		3	1
ARG	494	A	NE	A	1002	B	C1'	6.13		3	6
ARG	494	A	CZ	A	1001	B	N3	5.52		6	3
ARG	494	A	CZ	A	1002	B	C1'	6.25		6	6
ARG	494	A	NH1	A	1000	B	C2	6.07		3	5
ARG	494	A	NH1	A	1001	B	C2	4.8		3	5
LYS	495	A	CE	U	1003	B	C4	5.76		7	5
LYS	495	A	CE	U	1003	B	O4	5.23		7	2
LYS	495	A	CE	U	1003	B	C5	5.58		7	5
LYS	495	A	NZ	U	1003	B	O4	5.06		3	2
ARG	497	A	CZ	U	1010	B	O2	6.06		6	2
ARG	497	A	NH1	U	1011	B	O2	5.31		3	2
ARG	497	A	NH2	C	994	B	OP1	5.6		3	1
ARG	497	A	NH2	U	1011	B	C1'	5.77		3	6
ARG	497	A	NH2	U	1011	B	O2	4.18		3	2

994, 1000, 1001, 1002, 1003, 1010, 1011 (all of these are residues on exon7) which have shown interaction with 491 to 497 regions on EZH2.

Comment D4

4. The “FL-PLUMB” modelling section is much stronger, in that it results in three alternative candidate models for the protein-RNA complex being generated that then compete for validity

through the subsequent ASO studies. The authors assert that the results of these experiments align most closely with what would be expected if their “model 3” is correct (lines 240-242) however I cannot see the full evidence for this – Figures 5a-c are not insightful. To support their contention, the authors should show - probably in one-dimensional form – what regions of PLUMB are predicted to be interacting directly with the protein in each of the three models put forward, and what regions of PLUMB each of the ASOs target.

Response D4

We thank the reviewer for all the valuable comments and suggestions. In response to the comment above, we are providing the interface file in one dimension form for all the three models (Figure 5a-c) in Supplementary Table 5. Additionally, we are also providing a table showing the designed steric ASOs and their respective targeting exons on *PLUM* based on predicted interacting regions between FL *PLUM*-EZH2 docked models (Model1, Model2 and Model3).

FL PLUM -EZH2 docked model	Steric ASO	Targeting Exon on PLUM
Model1	s-ASO-g2	Exon1
Model1	s-ASO-g3	Exon9
Model1	s-ASO-g4	Exon4
Model2	s-ASO-g6	Exon4
Model2	s-ASO-g7	Exon5
Model2	s-ASO-g8	Exon5
Model2	s-ASO-g9	Exon9
Model3	s-ASO-g11	Exon6-exon7 junction
Model3	s-ASO-g12	Exon7
Model3	s-ASO-g14	Exon8

Table2: Table showing the designed steric ASOs and their respective targeting exons on *PLUM* based on predicted interacting regions between FL *PLUM*-EZH2 docked models (Model1, Model2 and Model3).

Reviewer #5, expertise in the UPR pathway (Remarks to the Author):

The manuscript by Deka et al. describes a new mechanism of chemoresistance in multiple myeloma (MM) involving a lncRNA named PLUM. PLUM regulates PRC2, leading to activation of the UPR and chemoresistance in MM. The manuscript presents a complete mechanistic story backed up by compelling evidence linking RNA, cell, and cancer biology, with well-chosen robust in vitro systems and metrics and substantiation of the findings in appropriate in vivo settings. However, it needs more discussion of previous research on the IRE1-XBP1 pathway in MM pathogenesis. Overall, the manuscript presents a significant advancement in understanding MM chemoresistance mechanisms and is suitable for publication in Nature Communications after addressing the comments below.

Main comment:

Comment E1

The IRE1-XBP1 pathway has been extensively studied for more than a decade as a contributor/driver of MM pathogenesis, and several studies show that blocking it produces desirable therapeutic outcomes in animal models (see, for example, PMIDS 17418411, 21081713, 24812669, 38748820, 31371506). Moreover, a recent comprehensive review of the UPR in disease highlights its many roles in cancer, including MM (PMID 39501044). These works should be cited and discussed accordingly.

Response E1

We appreciate the insightful suggestion provided by the reviewer regarding detailed studies already done on UPR pathway and its evaluation in MM pathogenesis. To enrich and support our findings, we have incorporated these previous finding details in result as well in discussion section more elaborately in our revised manuscript (line 399 to 408 in the revised manuscript).

Minor comments and suggestions for improvements:

Comment E2

1. The figures present an enormous amount of data to the reader. For that reason, the panels have been minimized considerably, which sometimes makes it challenging to read them and fully appreciate the data. Consider enlarging the panels and fonts.

Response E2

We thank the esteemed reviewer for your comment. We have now enlarged our figure panels as much as possible in the revised manuscript figure files.

Comment E3

2. The heat map in Fig. 1c is difficult to read; consider enlarging or presenting a crop with the most significant finding and place the complete heat map in the supplement.

Response E3

Thank you for your insightful suggestion. In our revised figure files, we have enlarged the heat map for fig 1c to the maximum extent possible.

Comment E4

3. The micrographs in Fig. 1g and 3f are hard to interpret as provided. Consider enlarging the images and providing close-ups (crops) in insets, showing representative data.

Response E4

We have provided enlarged close ups of the observed frames in insets for Figure 1g and Figure 3f of our revised manuscript.

Comment E5

4. In Fig. 3e, what are the identities of the two data points depicted in blue that also show significant upregulation?

Can add the name of the other two proteins

Response E5

We appreciate the question from the reviewer regarding other two upregulated proteins in our MS data of figure 3e, but as we intend to pursue those targets in future studies, we constrain ourselves from mentioning the name of the proteins in the figure. Although, we have provided the raw data file for our MS data in Supplementary table 4.

Comment E6

5. Fig. 3j/k suggests PLUM regulates EZH2 stability. Could the authors discuss or speculate how PLUM may protect EZH2 from degradation?

Response E6

Thank you for the comment on the probable mechanism underlying *PLUM* mediated protection of EZH2 from degradation. Our data suggests *PLUM* can mediate PRC2 complex formation through interaction with EZH2, which we speculate can in turn promote the stability of EZH2. As shown in Figures 4a and 5f, sh-*PLUM* KD or s-ASO targeted inhibition of *PLUM*-EZH2 interaction resulted in the disruption of PRC2 complex formation. Since previous studies have reported that the interaction of EZH2 with PRC2 complex subunits, EED and SUZ12, is critical for its protein stability [12-15], we speculate that one plausible mechanism (based on our current observations) could be *PLUM* facilitates formation of the PRC2 complex, thus promoting EZH2 stability and activity.

Comment E7

6. Can the unstructured region 32-42 of EZH2 be depicted in Fig. 4 to better situate the reader spatially on the structural model? The data in Fig. 4h suggest this region is essential.

For fig 4f: instead of line diagram, can add a tertiary structure of EZH2 with residues marked in different color.

Response E7

We thank the reviewer for the valuable suggestions. For first part of the comment, since unstructured region from 32-42 residues is not predicted in our docking data (*PLUM*-EZH2 docking in figure 4e), we did not show those residues in that spatial structural model. However, as suggested we have revised our figure 4f with an additional tertiary structure of EZH2 highlighting all the selected interacting residues in different color code in our revised manuscript (Figure 4f).

Figure 26: Tertiary structure of EZH2 protein with marked RNA interacting residues in different color code. **Green: F32-K39** (from literature and our experimental data), **Magenta: T345** (from literature-CDK1/2 phosphorylation site), **Red: P489-R494** (from literature and our in-silico docking data).

Comment E8

6. Consider softening the claim that s-ASO-g4 has an effect; the effect is modest or non-existent (Fig. 5).

Response E8

We thank the reviewer for the suggestion. In our response, we would like to clarify that we are not claiming s-ASO-g4 has an effect in reducing resistance or reduction in oncogenic functions. We only found it to disrupt *PLUM*-EZH2 interaction to some extent in our preliminary RIP-qPCR experiment compared to other s-ASOs tested (Supplementary figure 6a). Hence, we picked up top three s-ASOs (s-ASO-g4, s-ASO-g11 and s-ASO-g12) to perform our downstream confirmatory experiments. However, we later observed that the reduction in *PLUM*-EZH2 interaction in the presence of s-ASO-g4 might be due to overall increased degradation rate of *PLUM* in s-ASO-g4 treated cells (Supplementary figure 6c). We have revised the text to soften the claim as suggested (lines 282-284 of revised manuscript).

Comment E9

7. In Fig. 6h, did the authors inadvertently reverse the blot for P-eIF2a? These data are not consistent with the rest of the UPR markers.

Response E9

We appreciate the detailed observation and concerned raised by the reviewer. However, we did not inadvertently reverse the blot for P-eIF2a in Fig 6h. This result was found consistently for all three replicates in BTZ-S/R, Len S/R and Dexa S/R cells. We speculate that this phenomenon might be due to enhanced proliferative stress in all the resistant cell lines, which can be pursued in future as a separate study. Additionally, there are reports suggesting the

enhanced activation of eIF2 α phosphatases in BTZ resistant conditions can cause MM cells to enter the quiescent phase [16].

Comment E10

8. In Fig. 6c,d, PLUM KD reduces P-eIF2a while OE enhances it. In Fig. 7j,k, the KD of FOXO3 and ZFP36 also enhance P-eIF2a, which is in line with the authors' interpretation that FOXO3 and ZFP36 suppress UPR activity (not just IRE1). Accordingly, these findings and the possible roles of the other two UPR sensors, PERK and ATF6, should be discussed more thoroughly.

Response E10

We thank the reviewer for such insightful observation and comments on discussing the possible role of all sensors of UPR pathway (as observed in our immunoblotting and qPCR data) more elaborately. Accordingly, we have edited our main text for figure 6 and figure 7 results along with discussion part to include both our and previous findings on the role of all three UPR sensors (IRE1 α , PERK, ATF6 α) in MM pathogenesis and chemoresistance. We have also discussed to some extent how these sensors are exploited over time as therapeutic targets and how our findings on two more EZH2 regulated genes *FOXO3* and *ZFP36* might add some additional insight into UPR pathway based targeted therapy in MM (line 399 to 408 in the revised manuscript).

Comment E11

9. Are the blots shown for ATF6 for the full-length protein? Commercial ATF6 ~would be best to look at downstream transcriptional targets (e.g., BiP, HERPUD1, GRP94, PDIA4).

Response E11

We thank the reviewer for the technical question.

Yes, though antibody was shown to detect full length ATF6a in the website, in our western blots, we usually observe strong bands for cleaved ATF6a which is ~50 kDa in size. We have provided our uncropped blots in "Source Data file".

As suggested, we have also performed qPCR for BiP, HERPUD1, GRP94, PDIA4 in sh *PLUM* and *PLUM* OE samples. As per the qPCR data, all the ATF6 α downstream targets are downregulated upon sh-RNA mediated KD of *PLUM* and are upregulated in *PLUM* overexpressed MMCLs (KMS11 and LP1).

Figure 27: A, B. Expression level of ATF6 α target genes (*Bip*, *HERPUD1*, *GRP94* and *PDIA4*) in sh-*PLUM* knock down MMCLs (KMS11 and LP1) compared to scramble RNA transduced cells. (N=3, mean \pm SEM is plotted). **C, D.** Expression level of ATF6 α target genes (*Bip*, *HERPUD1*, *GRP94* and *PDIA4*) in *PLUM* overexpressed MMCLs (KMS11 and LP1) compared to scramble RNA transduced cells. (N=3, mean \pm SEM is plotted).

Comment E12

11. The manuscript would benefit from quantification of the immunoblots, especially those in Fig.6 and 7, as they provide the mechanistic links between PLUM/PRC2 and the regulation of the UPR, potentially via FOXO3 and ZPF36.

Response E12

We thank the reviewer for the valuable suggestion. Agreeing with it, we have added quantified values for immunoblots for UPR pathway analysis provided in figures 6 and 7 in “Source Data file”.

Reviewer #6, ECR (Remarks to the Author):

1. Sebastian, S., et al., *Intracellular Accumulation Of Light Chains Caused By Lenalidomide, and Mediated Via CRBN, Causes Major ER Stress and Is Implicated In The Synergistic Combination Of Imids and Proteasome Inhibitors*. Blood, The Journal of the American Society of Hematology, 2013. **122**(21): p. 4434.
2. Long, Y., et al., *Conserved RNA-binding specificity of polycomb repressive complex 2 is achieved by dispersed amino acid patches in EZH2*. Elife, 2017. **6**: p. e31558.
3. Szabó, C.L., et al., *The Disordered EZH2 Loop: Atomic Level Characterization by 1HN-and 1Hα-Detected NMR Approaches, Interaction with the Long Noncoding HOTAIR RNA*. International Journal of Molecular Sciences, 2022. **23**(11): p. 6150.
4. Davidovich, C., et al., *Promiscuous RNA binding by Polycomb repressive complex 2*. Nature structural & molecular biology, 2013. **20**(11): p. 1250-1257.
5. Gail, E.H., et al., *Inseparable RNA binding and chromatin modification activities of a nucleosome-interacting surface in EZH2*. Nature Genetics, 2024: p. 1-10.
6. Kaneko, S., et al., *Phosphorylation of the PRC2 component Ezh2 is cell cycle-regulated and up-regulates its binding to ncRNA*. Genes & development, 2010. **24**(23): p. 2615-2620.
7. Basu, S. and R.P. Bahadur, *A structural perspective of RNA recognition by intrinsically disordered proteins*. Cellular & molecular life sciences, 2016. **73**: p. 4075-4084.
8. Ottoz, D.S. and L.E. Berchowitz, *The role of disorder in RNA binding affinity and specificity*. Open Biology, 2020. **10**(12): p. 200328.
9. Calabretta, S. and S. Richard, *Emerging roles of disordered sequences in RNA-binding proteins*. Trends in biochemical sciences, 2015. **40**(11): p. 662-672.
10. Qiu, C., et al., *Intra-and inter-molecular regulation by intrinsically-disordered regions governs PUF protein RNA binding*. Nature Communications, 2023. **14**(1): p. 7323.
11. Zeke, A., et al., *Deep structural insights into RNA-binding disordered protein regions*. Wiley Interdisciplinary Reviews: RNA, 2022. **13**(5): p. e1714.

12. Wu, S.C. and Y. Zhang, *Cyclin-dependent kinase 1 (CDK1)-mediated phosphorylation of enhancer of zeste 2 (Ezh2) regulates its stability*. Journal of biological chemistry, 2011. **286**(32): p. 28511-28519.
13. Wan, L., et al., *Phosphorylation of EZH2 by AMPK suppresses PRC2 methyltransferase activity and oncogenic function*. Molecular cell, 2018. **69**(2): p. 279-291. e5.
14. Wei, Y., et al., *CDK1-dependent phosphorylation of EZH2 suppresses methylation of H3K27 and promotes osteogenic differentiation of human mesenchymal stem cells*. Nature cell biology, 2011. **13**(1): p. 87-94.
15. Guo, Y., et al., *Regulation of EZH2 protein stability: new mechanisms, roles in tumorigenesis, and roads to the clinic*. EBioMedicine, 2024. **100**.
16. Schewe, D.M. and J.A. Aguirre-Ghiso, *Inhibition of eIF2 α dephosphorylation maximizes bortezomib efficiency and eliminates quiescent multiple myeloma cells surviving proteasome inhibitor therapy*. Cancer research, 2009. **69**(4): p. 1545-1552.

Reviewer #1 (Remarks to the Author):

The authors have satisfactorily address all of my concerns.

Reviewer #2 (Remarks to the Author):

General major comments:

Could the authors clarify, for example in the abstract (lines 32–33: “Hence, targeting PLUM–EZH2 interactions may represent a clinically potent strategy for the treatment of relapsed, refractory MM”) and in lines 104–106 (“Our data further suggest that PLUM may be a potential therapeutic target for high-risk MM patients displaying enhanced EZH2 levels and resistance to current treatments”), how targeting PLUM–EZH2 interactions would yield a distinct clinical advantage compared to directly inhibiting EZH2 alone? Prior studies, including some cited by the authors, have demonstrated that EZH2 inhibition can sensitize multiple myeloma (MM) cells to proteasome inhibitors, primarily through reductions in EZH2 activity, loss of H3K27me3, and de-repression of tumor suppressors. If PLUM’s therapeutic relevance hinges on modulating these same downstream effects, it remains unclear what additional benefit PLUM targeting would offer. Furthermore, since EZH2 inhibitors are already FDA-approved and generally well-tolerated, the potential therapeutic gain from targeting a lncRNA like PLUM—which may have pleiotropic roles—must be carefully justified. Could the authors elaborate on whether PLUM targeting is expected to achieve mechanistically distinct or superior outcomes compared to existing EZH2 inhibitors?

Response: We thank the reviewer for commenting on the advantage of targeting *PLUM*-EZH2 interactions compared to directly inhibiting EZH2 alone. In a similar comment by another reviewer in our previous submission, we have clarified how targeting *PLUM*-EZH2 interactions would yield a distinct clinical advantage compared to directly inhibiting EZH2 alone (**Discussion section – line 453-478** in the revised manuscript). We would like to provide a more elaborate response on the key areas below for more clarity:

1. The tumorigenic and chemoresistance functions of EZH2 in MM have prompted several pre-clinical studies on the use of EZH2 catalytic inhibitors as a treatment option [1-4]. However, unlike other cancers, EZH2 expression in MM is mostly regulated by epigenetic mechanisms rather than gain-of-function mutations [5-9]. Though EZH2 overexpression is reported in MM, recurrent mutations of EZH2 have not been observed in MM [7]. Thus, exploring EZH2 as a potential therapeutic target in MM demands detailed investigation of its regulatory factors beyond its catalytic inhibitors.

2. Given the unknown role of PLUM in normal physiological processes and key roles of EZH2/PRC2 complex in normal immunoglobulin VDJ recombination in pre-B-cells and formation of germinal centre (GC) in B cells [10, 11], our strategy of using mixmer ASOs to interrupt specific PLUM-EZH2 interaction and its downstream oncogenic functions could be more beneficial compared to the use of global inhibitors of EZH2 like Tazemetostat. Our data also suggest non-mutant MMCLs (where PLUM expression is low) are not affected when treated with ASOs (Supplementary figure 2i and 2j).
3. There are many clinical trials ongoing involving EZH2 inhibitors in haematological malignancies (NCT02220842, NCT03460977, NCT02395601, NCT02900651) [12-14] but it appears that the drug(s) are not evaluated in MM patients yet [12]. Currently, EZH2 inhibitors are not FDA approved for use in MM patients.
4. Though initial pre-clinical results on the use of EZH2 inhibitors in MM are encouraging, some reports also demonstrate that MM cells can develop resistance to such inhibitors like tazemetostat (EPZ-6438) through epigenetic mechanisms [15]. In other haematological malignancies like DLBCL, these cancer cells have also been found to acquire genetic mutations to overcome the sensitivity to EZH2 inhibitors [16].

Hence, our steric-ASOs targeting PLUM-EZH2 interaction specifically represents a potential alternative treatment strategy to overcome some of the above-mentioned consequences or limited success of EZH2 inhibitors in MM. Moreover, the similar effects of PLUM-EZH2 targeting s-ASOs with that of EZH2 inhibitor (tazemetostat) in the destabilization of EZH2 activity and abrogation of chemoresistance support the possible use of PLUM as an alternative target to overcome the resistance mechanisms associated with known EZH2 catalytic inhibitors in MM. However, a competitive evaluation of these steric ASOs with available EZH2 inhibitors requires further studies.

Major comments:

1. Inconsistencies and Concerns Regarding Western Blot (WB) Source Data and Replicates

Comment 2A: Several issues were observed in the Western blot data that should be clarified:

In Supplementary Figure 1a, the provided source data shows contrasting image qualities compared to the published figure. Notably, GAPDH in replicate 1 of LP-1 appears different between the figure and the source data. Additionally, replicate 3 of KMS11 shows an inverted isoform pattern (less isoform 1, more isoform 2), inconsistent with the other replicates. This dataset should be repeated for clarity and consistency.

Response 2A:

Supplementary figure 1a: For LP1, the GAPDH blot shown in the main figure and source data file for rep1 belongs to the same blot. Previously, we have inserted the images from the same blot but with different intensities. Hence, the reviewer might have misunderstood them to be from two different blots. In the revised source data file, we have provided the GAPDH images at the same intensity (both separately).

The difference in the p52 band pattern in our western blots arises due to differences in the lot number of the same antibody. Hence, to address this issue, we usually perform densitometry considering both the bands together.

Additionally, we have added western blots for two more replicates in KMS11 and this has been updated in the current densitometric analysis as well as *PLUM* expression dataset (supplementary figure 1b).

Comment 2B: The GAPDH blots for replicate 1–2 in KMS11 and those in LP-1 (replicates 2–3) differ significantly in appearance or cropping compared to their respective KD blots. Can the authors confirm that these are the correct corresponding membranes?

Response 2B:

We confirm that these GAPDH blots are from the correct corresponding membranes. There might be some differences in the dimension of these blots after cropping.

Comment 2C:

In several instances (e.g., Figures 3H, 3I, 3K, 3L, 4H, Supplementary Figures 4E, 4H, 5A, 5B), GAPDH blots appear inconsistent or missing altogether. For example:
Figure 3H: Replicate 1 suggests EZH2 and EED were run on the same membrane, but SUZ12 and GAPDH appear on a different one.

Response 2C: We would like to clarify that we have shown corresponding GAPDH blots for all the western blots and RNA-protein pull down experiments described in our manuscript. We provide details of the listed figure panels below.

Figure 3H: EZH2, EED and GAPDH are run on same blot. SUZ12 blot was run separately. Yes, at times we need to run two blots for the same samples due to similar band size for multiple targets and in experiments like pull down, the band size does not come at the exact expected size due to different buffer system used during elution.

Since 3H is a RNA-protein pull down experiment, we expect to have GAPDH band only in the input samples. Eluted samples are not expected to display GAPDH band since lncRNA *PLUM* is not known or found to bind GAPDH transcript. We have run GAPDH for input samples to confirm that we have taken equal amount of protein to perform pull

down and the difference in binding we observed for different fragment of *PLUM* is not because of different loading volume (Red box in the image below). Usually, we keep 1% of total lysate as input samples (10 μ g as input if we take 1mg total lysate to perform pull down).

Figure 3I: For Rep1, GAPDH blot is from same blot as EZH2 blot. For Rep2 and Rep3, GAPDH blot is from the same blot as SUZ12 blot.

Figure 3K: Rep1 EZH2 blot was only later cut and re-probed for GAPDH. As you can see there was some ladder degradation in the EZH2 blot towards the high molecular weight size so I probed the whole blot for EZH2 first (to confirm that its only degradation of ladder and not my samples) and then cut it for re-probing with GAPDH. Rep2 GAPDH is again re-probed in the same EZH2 blot.

Figure 3L: GAPDH have been probed for all the reps.

Fig 4H: 4H is also a RNA-protein pull-down experiment so we are not expected to have GAPDH in our eluted samples. GAPDH in input samples are taken as loading and experimental control to be sure that we have taken equal amount of input lysate to perform our pull down with RNA.

For rep2 and rep3, both input and elute were loaded on the same blot but during taking of the chemiluminescence image, we need to cut the blot in between to take image for input and elute separately. This is due to limitation of the chemidoc machine where once it captures the saturated band, rest of the bands cannot be captured efficiently.

Supp fig 4E: It is also a RNA-protein pull-down experiment so we are not expected to have GAPDH in our eluted samples. GAPDH in input samples are taken as loading and experimental control to be sure that we have taken equal amount of input lysate to perform our pull down with RNA.

Supp fig 5A: It is a Co-IP experiment, so GAPDH band is not expected to be there for IPed samples. There are GAPDH blot for all our reps where we can see bands in the

input samples but not for IPed samples. GAPDH should not bind antibodies used for IP of some other proteins (In our case, we did IP for EZH2 Ab).

Supp 5B: All the blots have their respective GAPDH bots and are run on the same blot for all.

Comment 2D: Figure 3I: Replicate 1 shows a GAPDH with a visible ladder not present in the other protein blots.

Response 2D: Figure 3I, Rep1, GAPDH blot is cut from the same blot as EZH2 blot where the ladder pattern is same as that of the GAPDH blot.

General comment 2E: Total H3 should not be considered a loading control but rather to ensure that the effects seen are not due to a degradation of total histone H3.

Response 2E: Yes, we agree with the reviewer and in all our experiments, we would like to clarify that H3 was not considered as a loading control. Rather, GAPDH has been used and this information has been indicated in the figure legends as well.

Comment 2F: Figure 3L: Replicates show varying results for H3K27me3 and pEZH2, but loading controls are inconsistent or unclear.

Response 2F: In all our reps, we can confer that H3K27me3 and pEZH2 levels are downregulated upon treatment with degradative ASO-gIV and ASO-gV even in the presence of over-expressed EZH2. We provided the blot images herein, red box showing bands for pEZH2 and blue box showing bands for H3K27me3. And for all the sets, GAPDH is used as a loading control and this information has been indicated in the figure legends.

Comment 2G: Figure 4H: Input and pull-down appear to be from different membranes, yet a single GAPDH is shown.

Response 2G: For rep2 and rep3, both input and elute samples were loaded on the same blot but during taking of the chemiluminescence image, we need to cut the blot in between to take image for input and elute separately. This is due to a limitation of the chemi-doc machine where once it captures the saturated band, the remaining bands cannot be captured efficiently.

Also, it is a RNA-protein pull-down experiment so we are not expected to detect GAPDH in our eluted samples. GAPDH in the input samples are taken as loading and experimental control to be sure that we have taken equal amount of input lysates to perform our pull down with RNA.

Comment 2H: Supplementary Figure 5B appears to be missing GAPDH controls for sh-RNA SUZ12 replicates, and the number of replicates for EED (n=3) differs from EZH2 and SUZ12 (n=2).

Response 2H: Our raw data file already contains the GAPDH blot for sh-RNA SUZ12. It might be some technical error that during download or due to change in format during uploading of files in the portal that it is not visible in your end. For easy reference, we have added in the western blots for the same here (Supplementary Fig 5B – SUZ12 Rep1 Rep2). Additionally, we appreciate the reviewers concern for more replicates for EZH2 and SUZ12. We have now added one more replicate for both in our revised source data file.

Clarification is needed on whether membranes were stripped/reprobed or if these are distinct blots. If stripped, it should be noted; if distinct, each membrane should have its own corresponding loading control.

Response 2I: Thank you for the comment. We would like to clarify that most of our western blots are either stripped/reprobed for GAPDH or wherever possible, we cut out the GAPDH blot from the main blot (If we are not probing any protein near to GAPDH – 37kDa) band.

For some experiments like RNA-protein pull down experiment, we ran the same samples twice as stripping of blots from such experiments might result in false positive or false negative results. For such experiments, the same input and elute samples were divided into two parts and run in two different blots.

Comment 2J: For Supplementary Figures 1F, 4H, and possibly others, the uncropped WB source data is missing and should be included.

Response 2J:

Supp fig 1F: The uncropped WB was previously provided but we have now adjusted the cropping further and updated it for better clarity.

Supp fig 4H: We thank the reviewer for pointing this out. Yes, we have previously missed out the raw data file for this figure and have now added it in our revised source data file.

2. Questions About Normalization and Replication in RT-qPCR and Functional Assays

Comment 2K:

In Supplementary Figure 1G, the normalization across cytoplasmic and nuclear fractions differs between cell lines (KMS11 and LP-1), despite supposedly identical reference transcripts (7SK and cyto-C). However, MM.1S shows the same values for both compartments. Please clarify.

The figure legends claim biological duplicates were used, but based on the source data (Supplementary Figures 1G, 1C, 2A), it appears only a single technical replicate was used for many assays. Can the authors confirm whether qPCR assays were performed in technical triplicates for each gene and sample?

Response 2K:

Supp Fig 1G: We thank the reviewer for pointing out an error which arose while adding the raw data to source data file. This has now been rectified in the revised source data file. Additionally, the experiment comprises of two biological replicates but due to the enormous amount of raw data file, the average Ct value for two replicates is being provided in the source data file (unlike supp fig 1C, 2A where sample number is smaller so all the replicate Ct values are provided here).

Comment 2L:

Supplementary Figure 6C is unclear—were these results from technical triplicates?

For Figure 7C, are data presented from biological duplicates with only one technical replicate, or from full technical triplicates?

Response 2L:

Supp Fig 6C: The experiment for two technical replicates for each sample and 3 biological replicates has been presented in the overall analysis. The same is being highlighted more clearly in the revised source data file.

Figure 7C: The figure represent data from ChIP seq experiment performed as biological triplicates with only one technical replicate.

3. Statistical and Data Presentation Concerns

Comment 2M: The authors refer to statistically significant upregulation in Figure 1D (lines 133–135), though this is not supported by the visual or statistical output. Similarly, Supplementary Figure 7C lacks statistical testing, despite claims of significance.

Response 2M: We thank the reviewer for raising the concern on presentation of data. In response, we would like to justify the results shown as follows:

Figure 1D: When performing differential testing with DESeq2, we typically use an FDR cut off of 0.1 – a threshold commonly used in RNA-seq studies to balance sensitivity and specificity. *PLUM* has an FDR of 0.0747 and was therefore considered significant and included in the plot. *PLUM* expression shows a biologically relevant ~2-fold increase in the VRD Non-Responsive group (log₂ fold change = 0.98) compared to the Responsive group. This corresponds to the upwards shift in the count distribution for the former group observed in the plot. We understand this may not have been clear as the y-axis was mislabelled as “normalised counts”. The plot displays variance-stabilized transformed (VST; similar to log transformation) values and has now been corrected. Additionally, the group-wise separation is strongly supported by a Wilcoxon rank-sum test ($p = 0.0018$), applied directly to the transformed expression values shown in the figure. To better align the statistical evidence with the visual presentation, we have updated Figure 1D to display the Wilcoxon p-value, which more directly reflects the group-wise comparison of the plotted data.

Figure 1: *PLUM* expression in VRd non-responsive and responsive patients Violin plot showing *PLUM* expression in VRd non-responsive and responsive patients (MMRF CoMMpass). The p-value is from a Wilcoxon rank-sum test applied to variance-stabilized transformed (VST) expression values. The adjusted p-value (FDR = 0.0747) is from DESeq2 differential expression analysis using the Wald test.

Supp fig 7C: The number of biological replicates was 2 for this experiment so we did not perform any statistics for the experiment. But it is visibly clear that non-mutant cells (red boxplots) display lower *PLUM* expression compared to mutant MMCLs (Black grey boxplots). However, for better clarification of our claim, we have added one more replicate for the experiment and statistical test has been performed in our revised manuscript. All the p values are provided and source data file as well as mentioned in detail in the figure legend (Supplementary figure 7c).

Figure 2: Box plots showing the expression level of UPR target genes (EDEM, CHOP, GRP94, ATF4 and Bip) in NF- κ B⁺ and NF- κ B⁻ MMCLs. All the fold change is being normalised to MM.1.S sample (N=3, mean \pm SEM is plotted, and statistical significance was determined by unpaired student's t test; *: p value for mutant MMCLs versus XG-7, #: p value for mutant MMCLs versus H929; */# p<0.05, **/# p<0.005, ***/### p<0.0005, ns-non-significant).

Comment 2N: Apoptosis assays in Figure 5I appear to be based on only two biological replicates, which limits statistical confidence. Can the authors justify the conclusions based on this sample size?

Response 2N: We understand the reviewer's concern on the number of replicates for the following experiment. Hence, we have added two more replicates of the experiment in our revised manuscript. Additionally, the source data file has been updated accordingly.

Figure 3: Box plot depicting the percentage of annexin + cells (Red colour) and annexin – cells (Green color) cells post treatment with steric ASOs. (N=4, mean \pm SEM for annexin+ versus annexin- is plotted, and statistical significance was determined by multiple t-test; * $p < 0.05$, ** $p < 0.005$, *** $p < 0.0005$, ns-non significant).

General concern: Is statistical testing based on $n=2$ reliable across multiple figures (e.g., Figures 3H, 3I, 5I, 6C–D)? Please explain the rationale for using such low replicate numbers and whether any statistical adjustments were made.

The issue with only two replicates and statistical assessments:

No estimate of variability: With only two replicates, you can't accurately estimate variance (standard deviation), which is essential for most statistical tests like t-tests or ANOVAs.

No degrees of freedom: In a t-test, degrees of freedom ($n_1 + n_2 - 2$) become very small (e.g., $df = 2$), leading to: Wide confidence intervals. Low statistical power. High risk of both Type I (false positive) and Type II (false negative) errors.

Outlier sensitivity: One outlier can completely skew your result because there's no buffer from additional data.

Response: We appreciate the reviewer's concern on $n=2$ for some of the experiments, for which we have added more biological replicates in our revised manuscript.

Figure 3H: Since no statistical testing has been applied here, and we have only verified the binding results from our mass spectrometry data (where n=4 biological replicates) in two different cell lines, we find the number of replicates (N=2 for KMS11 and N=2 for BTZ-R 8226 – supp fig 4e) sufficient to support the finding.

Figure 3I: This was already done in 3 biological replicates in our previous submission.

Figure 5I: We have added two more biological replicates for the experiment in our revised manuscript. 5I is done in KMS11 and we have also shown the same experiment already in another cell line BTZ-R8226 with 3 biological replicates in supplementary figure 6e. This has been clarified in the figure legends.

Figure 6C: This was already performed in 3 biological replicates in our previous submission.

Figure 6D: We have added one more replicate for the experiment in our revised manuscript and updated our source data file along with quantification values accordingly.

4. Experimental Design and Data Interpretation Issues

Figure 1E: The differences between MM and other cancers are difficult to distinguish. Consider improving contrast or data representation.

Response: We have changed the representation of the same data by adjusting the range (Taking log₂ TPM) in our revised manuscript.

Figure 4: Graph showing the expression level (log₂ TPM) of top p52 regulated lncRNAs (*lnc-NTMT1*, *MALT1-AS1*, *NR2F2-AS1* and *PLUM*) in 51 pan-cancer cell types from CCLE datasets (O: non-MM, n = 638; M: MM, n = 20). Pair wise and global p-values obtained using Wilcoxon and Kruskal–Wallis respectively.

Supplementary Figure 2B: The source data lack column headers, making interpretation of the dataset difficult.

Response: We have added the column headers in the revised source data file.

Supplementary Figure 3B and Figures 2C–F: There are inconsistencies in how the data are processed—for example, 0nM is normalized to 100 in some cases (KMS11) but not others (LP-1). Please clarify the normalization method and rationale.

Response: In all the samples, 0nM is normalised to 100. For LP1(Figure 2C), the blank of each technical replicate was subtracted from the individual reading of each technical replicate separately (instead of taking the average blank value for all three technical replicates). Hence, the values were slightly different instead of 100 % for 0nM. In the revised raw data file, we have provided the LP1 calculation with 0nm as 100%.

Supp Fig 3B: Raw data file has also been updated accordingly.

Comment 20: The logic of Figure 5E vs Supplementary Figure 6A is unclear. One shows ASO treatment vs IgG, while the other should be treated vs untreated. Are these from different experiments?

Response 20: Thank you for the comment. We think the reviewer has misinterpreted our results. Both of them are RIP-pPCR experiments. Supplementary figure 6A shows RIP qPCR data for EZH2 antibody and IgG antibody (control) for all the 10 steric ASOs designed. Supp figure 6A is done in 2 biological replicates. From that data we picked up 3 top most steric ASOs (g4, g11 and g12) whose treatment showed reduced binding of PLUM with EZH2 and performed one more replicate of RIP-qPCR using those selected sASOs (g4, g11, g12), for which data is shown in Figure 5E.

Figure 5G: H3K27me3 is run on two membranes, but histone H3 loading control appears on only one. Why were the blots not run on the same membrane?

Response: We thank the reviewer for the observation. To clarify, we would like to mention that for this blot, we also faced the same problem as figure 4H comment. H3K27me3 was run on one membrane only for both the cell lines (KMS11 – left and BTZ-R8226 -right) but during the taking of chemiluminescence image, we need to cut the blot in between to take image for both the cell lines separately (due to the limitation of chemi-doc machine where once it captures the saturated band, the remaining bands

cannot be captured efficiently and difference in expression level cannot be viewed properly).

5. Interpretation of Functional Outcomes and Therapeutic Relevance

Comment 2P: The claim that PLUMi reduces toxicity compared to other drugs is questionable. The authors report >8000 DEGs following shRNA-mediated PLUM knockdown, which could indicate extensive off-target effects. How do the authors reconcile this with the suggestion of reduced toxicity?

Response 2P: We thank the reviewer for the question. We are not claiming that PLUMi reduces toxicity compared to other drugs. However, taking in consideration the change in around 8000 DEGs (both up DEGs and down DEGs together), we performed gene ontology analysis (GO analysis) for both UP DEGs and DOWN DEGs in our previous submission to check whether the *PLUM* regulated genes are involved in pathways related to toxicity (Supplementary figure 7a, 7b). In our analysis, UP DEGs are mostly found to be associated with terms like actin filament organisation, organelle transportation, cell growth, positive regulation of apoptotic process, protein localization to nucleus, etc). And DOWN DEGs are found to be associated with terms like ER stress, cytoplasmic translation, ATP metabolism, metabolic processes related to biomolecules, etc. We did not observe any enrichment of terms related to cellular toxicity. Additionally, ASO-mediated *PLUM* knockdown in XG-7 and H929 cells (NF- κ B non mutant MMCLs) did not lead to increased apoptosis or loss of cell proliferation (Supp Fig 2i-j).

Figure 5: Enriched biological processes among upregulated DEGs and down regulated DEGs upon *PLUM* knockdown in KMS11 cell line.

Comment 2Q: It is also unclear why *PLUM* overexpression affects proliferation but only induces treatment resistance in MM cell lines with NF- κ B mutations. Can the authors provide mechanistic insights or supporting evidence?

Response 2Q: We thank the reviewer for the comment. We feel the reviewer has misinterpreted our data in context to *PLUM* overexpression and treatment resistance. We have shown that *PLUM* overexpression induces treatment resistance in both NF- κ B mutant and NF- κ B non mutant MMCLs (Figure 2c-f).

Also, we would like to clarify that MM cell lines with NF- κ B mutations harbour enhanced expression of p52, which in turn regulates the expression of *PLUM* (Supplementary figure 1a, 1b). This is also evident from the expression profiles of *PLUM* in NF- κ B mutant and NF- κ B versus non mutant MMCLs (Supplementary figure 1c), where mutant cell lines showed higher expression of *PLUM* compared to non-mutant cell lines. Yet, when we overexpress *PLUM* in non-mutant cell lines (XG-7 and H929), we observed the increase in IC50 value in overexpressed cells compared to vector control transduced cells, confirming that treatment resistance is regulated by the expression levels of *PLUM* in MM cell lines.

(Figure 2c-2f)

(Supplementary Figure 1a-1b)

(Supplementary Figure 1c)

6. Animal Study Clarifications

Comment 2R: The manuscript states that 20 mice were used (10 vector controls, 10 treated with DMSO or BTZ), implying 5 mice per group. However, the source data lists 6 mice in the lncOE + DMSO group and 4 in the lncOE + BTZ group. Please clarify this discrepancy and update the source data as needed.

Response 2R: We thank the reviewer for the comment. The source data file remains unchanged. However, while marking the boundary in excel sheet, 6 mice were mismarked for lncOE + DMSO group and 4 mice for lncOE + BTZ group. In the revised source data file, marking is being rectified to avoid any further confusions.

Reviewer #3 (Remarks to the Author):

I co-reviewed this manuscript with one of the reviewers who provided the listed reports. This is part of the Nature Communications initiative to facilitate training in peer review and to provide appropriate recognition for Early Career Researchers who co-review

manuscripts.

Reviewer #4 (Remarks to the Author):

Comment 4A

The authors have addressed my questions, comments and suggestions on their original manuscript in a comprehensive and positive manner in their rebuttal letter; though it's slightly disappointing that the proportion of this that has actually been incorporated into the revised manuscript and supplementary material seems to be the bare minimum. I have no new issues to raise.

Response 4A

We appreciate the reviewer's valuable comment in the previous revision which helped us to strengthen our claim on our *in-silico* data. We at the same time apologise for not adding all the data provided in the rebuttal file to our main text or supplementary data. However, in the current revised manuscript, we have incorporated as much of the material as possible.

We mentioned about the AlphaFold3 data in the result section 4 emphasizing the similar finding with our previous prediction where disordered region of EZH2 is found to show interaction with residues present in exon7 of PLUM. The text is changed as "Consistent with this, structural predictions of full-length PLUM-EZH2 complex using Alphafold3 [41] similarly revealed interaction of the disordered region of EZH2 protein (491-497 residues) with atoms in the exon 7 of PLUM (Supplementary figure 5c and supplementary table 4, 8). Specifically, the prediction model data showed interaction of EZH2 with 994, 1000, 1001, 1002, 1003, 1010, 1011 atoms on PLUM which are part of its exon7 (Supplementary table 5). Previous studies have also indicated some potential RNA binding sites on mammalian PRC2 subunits, including residues 342-368 in an unstructured region of mouse EZH2 (phospho-mimic at residue 345), N-terminal residues from 32-42, and C terminal unstructured region from 489-494 in human EZH2 (in vitro binding data with G- quadruplex RNA) [42-45] (figure 4f). Additionally, earlier reports have suggested the disordered region of EZH2 to be important in RNA recognition and interaction [46-51]. Hence, we proceeded to validate the region on EZH2 protein having maximum potential to interact with PLUM." (Line 239-251 in the revised manuscript)

Also, regarding the ASO part we incorporated more details in result section 5. The revised text is changed as "Ten steric ASOs (s-ASOs) were designed against all the interacting regions between EZH2 and PLUM (inclusive of all exons): s-ASO-g2 against exon1, s-ASO-g3 against exon9, s-ASO-g4 against exon4 (Model 1-Figure 5a); s-ASO-g6 against exon4, s-ASO-g7 against exon5, s-ASO-g8 against exon5, s-ASO-g9 against exon9 (Model 2 – Figure 5b); s-ASO-g11 against exon6-exon7 junction, s-ASO-g12

against exon7 and s-ASO-g14 against exon8 (Model 3 – Figure 5c). All our steric ASOs were modified with 2'-O-Methoxyethyl/locked nucleic acid (2'MOE-LNA+) chemistry (Figure 5d). Furthermore, all the steric ASOs were tested alongside negative control (NC-ASO) for their activity to validate our in-silico docking data through RIP-qPCR using EZH2 antibody (Supplementary figure 6a and figure 5e). The results showed s-ASO-g11 and s-ASO-g12 targeting exon6-exon7 junction and exon7 respectively are most effective in disrupting the PLUM-EZH2 complex, whose treatment demonstrated maximum decline in fold enrichment for PLUM in EZH2 RIP-qPCR results compared to other s-ASOs (Figure 5e and supplementary figure 6a). The targeting region for s-ASO-g11 and s-ASO-g12 matched the docked model 3 of PLUM-EZH2 interaction, which forms complementary pairing with fragments of exon6-exon7 junction and exon of PLUM (Figure 5c and supplementary table 6). FL-PLUM-EZH2 model 3 could also recapitulate the Exon7-PLUM-EZH2 docked model (Figure 4e). For both the models, the residues between 489-494 in EZH2 interact with similar atoms on exon 7 of PLUM (Interface file provided in supplementary table 8).” (line 284-302 in the revised manuscript)

Reviewer #5 (Remarks to the Author):

I want to thank the Authors for addressing each one of the points I raised in my initial review of the work. All comments have been addressed satisfactorily.

However, to improve the manuscript, I would like to suggest the authors consider the following:

Comment 5A

1. Content: Please consider discussing the P-eIF2a results presented in Fig. 6 in the manuscript. The lowered P-eIF2a levels in resistant cells are unexpected given the high levels of IRE1 activity (measured by XBP1 splicing). The apparent discrepancy is explained in the rebuttal, but it would benefit the reader to include it in the manuscript. I also encourage the authors to consider adding their qPCR data on ATF6 targets (in response to one of my comments) as a supplementary figure, as it strengthens their conclusions.

Response 5A

We thank the reviewer for these additional suggestions to add in to the main manuscript. Accordingly, we have incorporated the explanation for p-EIF2a data for figure 6 in the result section (Line 351-355 in the revised manuscript).

We have also added the qPCR data for ATF6a targets as supplementary figure (Supplementary figure 7d in the revised manuscript).

Comment 5B

2. Writing: Line 400, consider the following edit: "...chemoresistance via the activation of the UPR pathway, in addition to several targeted therapy..."

Response 5B

We have edited the text accordingly.

Comment 5C

3. Figures: Revise the Western Blots in Figs. 6c and 6d as they appear distorted (squished); maybe there is a problem with document format conversions.

Response 5C

Thank you for pointing this out. We have looked into the dimension of both the figures and corrected them accordingly.

Reviewer #6 (Remarks to the Author):

References

1. Croonquist, P.A. and B. Van Ness, *The polycomb group protein enhancer of zeste homolog 2 (EZH2) is an oncogene that influences myeloma cell growth and the mutant ras phenotype*. *Oncogene*, 2005. **24**(41): p. 6269-6280.
2. Pawlyn, C., et al., *Overexpression of EZH2 in multiple myeloma is associated with poor prognosis and dysregulation of cell cycle control*. *Blood cancer journal*, 2017. **7**(3): p. e549-e549.
3. Rizq, O., et al., *Dual inhibition of EZH2 and EZH1 sensitizes PRC2-dependent tumors to proteasome inhibition*. *Clinical Cancer Research*, 2017. **23**(16): p. 4817-4830.
4. Dimopoulos, K., et al., *Dual inhibition of DNMTs and EZH2 can overcome both intrinsic and acquired resistance of myeloma cells to IMiDs in a cereblon-independent manner*. *Molecular Oncology*, 2018. **12**(2): p. 180-195.
5. Walker, B.A., et al., *Mutational spectrum, copy number changes, and outcome: results of a sequencing study of patients with newly diagnosed myeloma*. *Journal of clinical oncology*, 2015. **33**(33): p. 3911-3920.
6. Chapman, M.A., et al., *Initial genome sequencing and analysis of multiple myeloma*. *Nature*, 2011. **471**(7339): p. 467-472.
7. Rastgoo, N., et al., *Role of epigenetics-microRNA axis in drug resistance of multiple myeloma*. *Journal of hematology & oncology*, 2017. **10**: p. 1-10.
8. Rastgoo, N., et al., *Dysregulation of EZH2/miR-138 axis contributes to drug resistance in multiple myeloma by downregulating RBPMS*. *Leukemia*, 2018. **32**(11): p. 2471-2482.
9. Tremblay-LeMay, R., et al., *EZH2 as a therapeutic target for multiple myeloma and other haematological malignancies*. *Biomarker research*, 2018. **6**: p. 1-10.
10. Su, I.-h., et al., *Ezh2 controls B cell development through histone H3 methylation and Igh rearrangement*. *Nature immunology*, 2003. **4**(2): p. 124-131.
11. Béguelin, W., et al., *EZH2 is required for germinal center formation and somatic EZH2 mutations promote lymphoid transformation*. *Cancer cell*, 2013. **23**(5): p. 677-692.
12. *ClinicalTrials.gov. U.S. National Library of Medicine [https://clinicaltrials.gov/]. Accessed 12 June 2018.*
13. Morschhauser, F., et al., *Interim report from a phase 2 multicenter study of tazemetostat, an EZH2 inhibitor, in patients with relapsed or refractory B-cell non-Hodgkin Lymphomas*. *Hematological Oncology*, 2017. **35**: p. 24-25.
14. Alexander, W., *23rd Congress of the European Hematology Association*. *Pharmacy & Therapeutics*, 2018. **43**(9): p. 562.
15. Herviou, L., et al., *PRC2 targeting is a therapeutic strategy for EZ score defined high-risk multiple myeloma patients and overcome resistance to IMiDs*. *Clinical epigenetics*, 2018. **10**: p. 1-18.
16. Gibaja, V., et al., *Development of secondary mutations in wild-type and mutant EZH2 alleles cooperates to confer resistance to EZH2 inhibitors*. *Oncogene*, 2016. **35**(5): p. 558-566.